# Phenomenology of Double Descent in Finite-Width Neural Networks

Sidak Pal Singh[*][a,c], Aurelien Lucchi[b], Thomas Hofmann[a] and Bernhard Schölkopf[c]

[a]ETH Zürich, Switzerland
[b]Department of Mathematics and Computer Science, University of Basel
[c]MPI for Intelligent Systems, Tübingen, Germany

## Abstract

'Double descent' delineates the generalization behaviour of models depending on the regime they belong to: under- or over-parameterized. The current theoretical understanding behind the occurrence of this phenomenon is primarily based on linear and kernel regression models — with informal parallels to neural networks via the Neural Tangent Kernel. Therefore such analyses do not adequately capture the mechanisms behind double descent in finite-width neural networks, as well as, disregard crucial components — such as the choice of the loss function. We address these shortcomings by leveraging influence functions in order to derive suitable expressions of the population loss and its lower bound, while imposing minimal assumptions on the form of the parametric model. Our derived bounds bear an intimate connection with the spectrum of the Hessian at the optimum, and importantly, exhibit a double descent behaviour at the interpolation threshold. Building on our analysis, we further investigate how the loss function affects double descent — and thus uncover interesting properties of neural networks and their Hessian spectra near the interpolation threshold.

## 1 Introduction

Double-descent (DD) (Belkin et al., 2019) refers to the phenomenon of population (or test) loss exhibiting a second descent when the model is over-parameterized beyond a certain threshold (dubbed as the interpolation threshold). While such a behaviour has been originally observed in multiple contexts before (Loog et al., 2020), this terminology has been made commonplace by the influential work of (Belkin et al., 2019), who posited DD as a way to reconcile the traditional statistical wisdom with modern-day usage in machine learning. In particular, when models such as deep neural networks are employed in the over-parameterized regime, this second descent in test loss is seen as an extension of the usual U-shaped bias-variance curve — thus providing a basis to reason about the amazing generalization abilities of neural networks despite their extreme surplus of parameters.

As a result, there have been a vast number of studies that investigate DD further and others that seek to explain the underlying mechanisms. The former category of works (Nakkiran et al., 2019) has served to make this phenomenon conspicuous even in the context of frequently employed deep networks and shown the existence of such DD like behaviour based on other axes such as the amount of data or number of epochs (Nakkiran, 2019; Nakkiran et al., 2019); while the latter category of studies (Hastie et al., 2019; Mei & Montanari, 2019; Advani et al., 2020; Bartlett et al., 2020) has provided the theoretical grounds for the occurrence of such a phenomenon, but almost always, in linear or kernel regression. Although the connection of (infinite-width) neural networks to Neural Tangent Kernel (Jacot et al., 2018) provides interesting parallels, its applicability to finite-width neural networks — i.e., models of actual practical significance — is unclear.

Hence, our aim is to go beyond such informal correspondences, and instead, thoroughly understand and characterize the phenomenology of double descent *in finite-width neural networks*. In other words, we develop a theoretical analysis that can explain the source of double descent in the class of models that originally brought this phenomenon to the limelight, as well as study the impact of crucial training-related aspects, such as the choice of the loss function. To this end, we derive

---

[*]Correspondence to `sidak.singh@inf.ethz.ch`. The most recent version of our paper can be found on arXiv and the code for the experiments is available on GitHub.

an expression for the population risk by considering the change in the loss at a particular sample — when excluded from the training set and when included. We do so by utilizing the notion of influence functions from robust statistics, which provides us with a closed-form estimate of the change in parameters. Subsequently, we establish a lower bound to the population risk and leverage results from Random Matrix Theory to show that it diverges at the interpolation threshold — thereby demonstrating the presence of double descent.

The use of influence functions (Hampel et al., 1986; Efron & Stein, 1981) makes our analysis applicable to a broad class of parametric estimators such as maximum-likelihood-type — which includes the class of neural networks but also applies to other settings such as linear and kernel regression as a special case (see Section A.4). As a result, this approach allows us to probe the effect of the loss functions used to train neural networks — which, as we will see, clearly influences the nature of DD — and reveals novel insights into the properties of neural networks near the interpolation threshold.

**Contributions.** (a) Section 3: We derive a generalization expression based on a 'add-one-in' (akin to 'leave-one-out') procedure that requires minimal assumptions and, in principle, allows us to analyze the population risk of any neural network or linear/kernel regression model. (b) Section 4: We show how this yields a suitable asymptotic lower-bound, which diverges at the interpolation threshold and helps explain double descent for neural networks via the Hessian spectra at the optimum. We additionally support the theoretical arguments by an empirical investigation of the involved quantities, which also suggests its applicability in the non-asymptotic setting. (c) Section 5: We thoroughly study the effect of the loss function by exploring in detail the double descent behaviour for cross-entropy loss, such as the location of the interpolation threshold, Hessian spectra near interpolation, as well as discuss novel insights implied by these observations. (d) Section 6: Lastly, as a by-product of our influence functions based approach, we derive a generalized closed-form expression for leave-one-out loss that applies to finite-width neural networks.

## 2 SETUP AND BACKGROUND

Let us consider the general setting of statistical estimation. Assume that the input samples $z \in \mathcal{Z}$ are drawn i.i.d. from some (unknown) distribution $\mathcal{D}$. A parametric model can be defined as a family of probability distributions $\mathcal{D}_{\boldsymbol{\theta}}$ on the sample space $\mathcal{Z}$, where $\boldsymbol{\theta} \in \boldsymbol{\Theta}$. The main task is to find an estimate $\widehat{\boldsymbol{\theta}}$ of the parameters such that "$\mathcal{D}_{\widehat{\boldsymbol{\theta}}} \approx \mathcal{D}$", but given a finite set of samples $S = \{z_1, \cdots, z_n\}$ drawn i.i.d from $\mathcal{D}$. The parameter estimator, $\widehat{\boldsymbol{\theta}}$, is considered to be provided by some statistic $T(\boldsymbol{z}_1, \cdots, \boldsymbol{z}_n)$. Alternatively, if we denote the empirical distribution of the dataset $S$ by $\mathcal{D}_n$, then the parameter estimator can be written as a functional, $\widehat{\boldsymbol{\theta}}_{\mathcal{D}_n} := \widehat{\boldsymbol{\theta}}(\mathcal{D}_n) = T(\mathcal{D}_n)$.

### 2.1 INFLUENCE FUNCTIONS: A QUICK PRIMER

Quite often we are interested in analyzing how a slight contamination in the distribution affects, say, the estimated parameters $\widehat{\boldsymbol{\theta}}$, or some other statistic $T$. Specifically, when this contamination can be expressed in the form of a Dirac distribution $\delta_{\boldsymbol{z}}$, the (standardized) change in the statistic $T$ can be obtained via the concept of influence functions.

**Definition 1.** *(Hampel et al. (1986)) The influence function* IF *of a statistic $T$ based on some distribution $\mathcal{D}$, is given by,* $\quad \mathrm{IF}(\boldsymbol{z}; T, \mathcal{D}) = \lim_{\epsilon \to 0} \dfrac{T\left((1-\epsilon)\mathcal{D} + \epsilon\delta_{\boldsymbol{z}}\right) - T(\mathcal{D})}{\epsilon}$ *, evaluated on a point $\boldsymbol{z} \in \mathcal{Z}$ where such a limit exists.*

Essentially, the influence function (IF) involves a directional derivative of $T$ at $\mathcal{D}$ along the direction of $\delta_{\boldsymbol{z}}$. More generally, one can view influence functions from the perspective of a Taylor series expansion (or what is referred to in this context as the first-order von Mises expansion) of $T$ at $\mathcal{D}$, evaluated on some distribution $\widetilde{\mathcal{D}}$ close to $\mathcal{D}$: $T(\widetilde{\mathcal{D}}) = T(\mathcal{D}) + \int \mathrm{IF}(\boldsymbol{z}; T, \mathcal{D}) \, d(\widetilde{D} - \mathcal{D})(\boldsymbol{z}) + \mathcal{O}(\epsilon^2)$.

So, as evident from this, it is also possible to utilize higher-order influence functions as considered in Debruyne et al. (2008). However, the first-order term is usually the dominating term and to ensure tractability when we later consider neural networks — we will restrict our attention to first-order influence functions hereafter. Influence functions have a long history as an important tool, particularly in robust statistics (Hampel et al., 1986), but lately also in deep learning (Koh & Liang, 2017). Not only do they have useful properties (see Appendix B.1 for a detailed background), but they form a natural tool to analyze changes in distribution (e.g., leave-one-out estimation).

## 2.2 Influence function for Maximum likelihood type Estimators

In general, we do not explicitly have the analytic form of the estimator $\widehat{\boldsymbol{\theta}}$ — but implicitly as the solution to an optimization problem (e.g., parameters of neural networks obtained via training). So, let us consider influence functions for the class of 'maximum likelihood type' estimators (or M-estimators), i.e. $\widehat{\boldsymbol{\theta}}$ satisfying the implicit equation, $\mathbb{E}_{\boldsymbol{z} \sim \mathcal{D}} \left[ \boldsymbol{\psi}(\boldsymbol{z}; \widehat{\boldsymbol{\theta}}) \right] = 0$. Let us assume that the function $\boldsymbol{\psi}(\boldsymbol{z}; \boldsymbol{\theta}) := \nabla_{\boldsymbol{\theta}} \ell(\boldsymbol{z}, \boldsymbol{\theta})$, where the (loss) function $\ell : \mathcal{Z} \times \boldsymbol{\Theta} \mapsto \mathbb{R}$ is twice-differentiable in the parameters $\boldsymbol{\theta}$. When the loss $\ell$ is the negative log-likelihood, we recover the usual maximum likelihood estimator, $\widehat{\boldsymbol{\theta}} = \arg\min_{\boldsymbol{\theta} \in \boldsymbol{\Theta}} \mathbb{E}_{\boldsymbol{z} \sim \mathcal{D}} [\ell(\boldsymbol{z}, \boldsymbol{\theta})]$. The following proposition describes the influence function for such an estimator $\widehat{\boldsymbol{\theta}}$ (all the omitted proofs can be found in Appendix A).

**Proposition 2.** *The (first-order) influence function IF of the M-estimator $\widehat{\boldsymbol{\theta}}_{\mathcal{D}}$ based on the distribution $\mathcal{D}$, evaluated at a point $\boldsymbol{z}$, takes the following form:* $\mathrm{IF}(\boldsymbol{z}; \widehat{\boldsymbol{\theta}}_{\mathcal{D}}, \mathcal{D}) = -\left[ \mathbf{H}_{\mathcal{L}}(\widehat{\boldsymbol{\theta}}_{\mathcal{D}}) \right]^{-1} \nabla_{\boldsymbol{\theta}} \ell(\boldsymbol{z}, \widehat{\boldsymbol{\theta}}_{\mathcal{D}})$, *where, the Hessian matrix* $\mathbf{H}_{\mathcal{L}}(\widehat{\boldsymbol{\theta}}_{\mathcal{D}}) := \nabla_{\boldsymbol{\theta}}^2 \mathcal{L}(\widehat{\boldsymbol{\theta}}_{\mathcal{D}})$ *contains the second derivative of the loss* $\mathcal{L}(\boldsymbol{\theta}) := \mathbb{E}_{\boldsymbol{z} \sim \mathcal{D}} [\ell(\boldsymbol{z}, \boldsymbol{\theta})]$ *with respect to the parameters* $\boldsymbol{\theta}$.

**Remark.** The Hessian in neural networks is typically rank deficient (Sagun et al., 2017; Singh et al., 2021), so for the sake of our analysis we will consider an additive regularization term $\lambda \mathbf{I}$, $\lambda > 0$, alongside the $\mathbf{H}_{\mathcal{L}}(\widehat{\boldsymbol{\theta}}_{\mathcal{D}})$ term in the above-mentioned IF formula and call this modification $\mathrm{IF}_{\lambda}$. Later, we will take the limit $\lambda \to 0$.

## 3 Expression of the population risk

As we lack access to the true distribution, we usually take the route of empirical risk minimization and consider $\widehat{\boldsymbol{\theta}}_S := \arg\min_{\boldsymbol{\theta} \in \boldsymbol{\Theta}} \mathcal{L}_S(\boldsymbol{\theta})$ with $\mathcal{L}_S(\boldsymbol{\theta}) := \frac{1}{n} \sum_{i=1}^{n} \ell(\boldsymbol{z}_i, \boldsymbol{\theta})$ and recall, $S = \{\boldsymbol{z}_i\}_{i=1}^{n}$ is the training set of size $n$. However, in regards to performance, our actual concern is the population risk $\mathcal{L}(\boldsymbol{\theta}) := \mathbb{E}_{\boldsymbol{z} \sim D} [\ell(\boldsymbol{z}, \boldsymbol{\theta})]$, which is measured empirically via the loss $\mathcal{L}_{S'}(\boldsymbol{\theta})$ on some unseen (test) set $S'$. To analyze double descent (Belkin et al., 2019) — i.e., with increasing model capacity, the population risk exhibits a peak before descending again (besides the usual first descent) — we first derive a suitable expression of the population risk that will also apply to neural networks.

**'Add-one-in' procedure.** Let us consider how the parameter estimate $\widehat{\boldsymbol{\theta}}_S$ changes when an additional sample $\boldsymbol{z}' \sim D$ is included in the training set $S$. We can cast this as a contamination of the corresponding empirical distribution $\mathcal{D}_n$ to yield an altered distribution $\mathcal{D}'_{n+1} = (1 - \epsilon)\mathcal{D}_n + \epsilon \, \delta_{\boldsymbol{z}'}$, with the contamination amount $\epsilon = \frac{1}{n+1}$. Thanks to the influence function of the parameter estimate (Proposition 2 for $\mathcal{D} = \mathcal{D}_n$), we do not have to retrain and explicitly measure the change in final parameters, but have an analytic expression given as follows, $\mathrm{IF}_{\lambda}(\boldsymbol{z}'; \widehat{\boldsymbol{\theta}}_S, \mathcal{D}_n) = -\left[ \mathbf{H}_{\mathcal{L}}^{S}(\widehat{\boldsymbol{\theta}}_S) + \lambda \mathbf{I} \right]^{-1} \nabla_{\boldsymbol{\theta}} \ell(\boldsymbol{z}', \widehat{\boldsymbol{\theta}}_S)$, where the superscript $S$ in $\mathbf{H}_{\mathcal{L}}^{S}$ denotes the computation of the Hessian on the set $S$. Now using the chain rule of influence functions, we can express the influence on $\ell_{\boldsymbol{z}'} := \ell(\boldsymbol{z}', \boldsymbol{\theta})$ as, $\mathrm{IF}_{\lambda}(\ell_{\boldsymbol{z}'}; \widehat{\boldsymbol{\theta}}_S, \mathcal{D}_n) = -\nabla_{\boldsymbol{\theta}} \ell(\boldsymbol{z}', \widehat{\boldsymbol{\theta}}_S)^{\top} \left[ \mathbf{H}_{\mathcal{L}}^{S}(\widehat{\boldsymbol{\theta}}_S) + \lambda \mathbf{I} \right]^{-1} \nabla_{\boldsymbol{\theta}} \ell(\boldsymbol{z}', \widehat{\boldsymbol{\theta}}_S)$. When $|S| = n$ is large enough to ignore $\mathcal{O}(n^{-2})$ terms, the (infinitesimal) definition of influence function is equivalent to using the finite-difference form. This allows us to directly express the change in loss which we leverage to derive an expression of the test loss, in Theorem 3 below.

**Theorem 3.** *Consider the parameter estimator $\widehat{\boldsymbol{\theta}}_S$ based on the set of input samples $S$ of $|S| = n$. Then the population risk $\mathcal{L}(\widehat{\boldsymbol{\theta}}_S) := \mathbb{E}_{\boldsymbol{z} \sim D} \left[ \ell(\boldsymbol{z}, \widehat{\boldsymbol{\theta}}_S) \right])$ takes the following form,*

$$\mathcal{L}(\widehat{\boldsymbol{\theta}}_S) = \widetilde{\mathcal{L}}_S(\widehat{\boldsymbol{\theta}}_S) + \frac{1}{n+1} Tr\left( \left[ \mathbf{H}_{\mathcal{L}}^{S}(\widehat{\boldsymbol{\theta}}_S) + \lambda \mathbf{I} \right]^{-1} \mathbf{C}_{\mathcal{L}}^{\mathcal{D}}(\widehat{\boldsymbol{\theta}}_S) \right) + \mathcal{O}\left( \frac{1}{n^2} \right), \quad (1)$$

*where $\widetilde{\mathcal{L}}_S(\widehat{\boldsymbol{\theta}}_S) := \mathbb{E}_{\boldsymbol{z}' \sim \mathcal{D}} \left[ \ell\left( \boldsymbol{z}', \widehat{\boldsymbol{\theta}}_{S \cup \{\boldsymbol{z}'\}} \right) \right]$ is the expectation of 'one-sample training loss' and $\mathbf{C}_{\mathcal{L}}^{\mathcal{D}}(\widehat{\boldsymbol{\theta}}_S) := \mathbb{E}_{\boldsymbol{z}' \sim \mathcal{D}} \left[ \nabla_{\boldsymbol{\theta}} \ell(\boldsymbol{z}', \widehat{\boldsymbol{\theta}}_S) \nabla_{\boldsymbol{\theta}} \ell(\boldsymbol{z}', \widehat{\boldsymbol{\theta}}_S)^{\top} \right]$ is the (uncentered) covariance of loss gradients.*

**Remark.**    Note, the first term in the right-hand side of eq. 1 is not exactly the training loss, but deviates by a negligible quantity related to the expected difference between the loss of a training sample and the average loss on the remaining training set (as discussed in Section A.1.2). When trained sufficiently long so that the loss on individual samples is close to zero (i.e., the interpolation setting), this quantity becomes inconsequential for the purpose of analyzing double descent.

**Related work.**    The more interesting quantity in eq. 1 is the second term on the right, which is reminiscent of the Takeuchi Information Criterion (Takeuchi, 1976; Stone, 1977) and a similar term appears in several works on neural networks (Murata et al., 1994; Thomas et al., 2019) as well as in the analyses of least-squares regression (Flammarion & Bach, 2015; Défossez & Bach, 2015; Pillaud-Vivien et al., 2018). However, an important difference is that in our expression the Hessian $\mathbf{H}_{\mathcal{L}}^S$ evaluated on the training set appears, whereas it is based on the entire true data distribution in prior works. This seemingly minor difference is in fact crucial, since studying double descent necessarily involves analyzing the relation between the training set size and the model capacity (such as the number of parameters). Also, directly taking the results from previous works and merely approximating the involved quantities based on the training set, such as the Hessian and the covariance $\mathbf{C}_{\mathcal{L}}^{\mathcal{D}}$, does not work either. Since in such a scenario, when the limit of the regularization strength $\lambda \to 0$, results from prior works reduce to something ineffective for further analysis. E.g., for mean-squared loss, to the rank of Hessian at the optimum (see Section A.2.2 for more details).

## 4    LOWER BOUND ON THE POPULATION RISK

As a brief outline, our strategy to theoretically illustrate the double descent behaviour will be to show that the lower bound of the population risk diverges around a certain threshold. Before proceeding further, we would like to emphasize that so far *we have not employed any assumptions on the structural form of the model or the neural network*, as well as neither on the data distribution.

So let us now introduce some relevant notations to describe the precise setting for our upcoming result. We assume that the samples $z$ are tuples $(x, y)$, where the input $x \in \mathbb{R}^d$ has dimension $d$ and the targets $y \in \mathbb{R}^K$ are of dimension $K$. Let us consider the neural network function is $f_{\theta}(x) := f(x, \theta) : \mathbb{R}^d \times \mathbb{R}^p \mapsto \mathbb{R}^K$, where we have taken $\theta \in \mathbb{R}^p$. The parameter estimator, $\widehat{\theta}_S$, in this context refers to the parameters obtained by training to convergence using the training set $S$, and which we will henceforth denote by $\theta^\star := \widehat{\theta}_S$ (to emphasize the fact that we are at the local optimum). Also, the loss function $\ell(z, \theta)$, with a slight abuse of notation, refers to $\ell(f_{\theta}(x), y)$ in this setting. Let us additionally define a shorthand $\ell_i := \ell(f_{\theta}(x_i), y_i)$. Next, let us discuss in more detail the two matrices that appear in Theorem 3, before we introduce the assumptions we require.

First, the *Hessian matrix of the loss*, can in general be decomposed as a sum of two other matrices (Schraudolph, 2002): outer-product Hessian $\mathbf{H}_o^S(\theta)$ and functional Hessian $\mathbf{H}_f^S(\theta)$, i.e.,

$$\mathbf{H}_{\mathcal{L}}^S(\theta) = \mathbf{H}_o^S(\theta) + \mathbf{H}_f^S(\theta) = \frac{1}{n} \sum_{i=1}^{n} \nabla_{\theta} f_{\theta}(x_i) \left[ \nabla_f^2 \ell_i \right] \nabla_{\theta} f_{\theta}(x_i)^\top + \frac{1}{n} \sum_{i=1}^{n} \sum_{k=1}^{K} [\nabla_f \ell_i]_k \, \nabla_{\theta}^2 f_{\theta}^k(x_i)$$

(2)

where, $\nabla_{\theta} f_{\theta} \in \mathbb{R}^{p \times K}$ is the Jacobian of the function and $\nabla_f^2 \ell \in \mathbb{R}^{K \times K}$ is the Hessian of the loss with respect to the function. Next, the other matrix that appears in eq. 1 is the (uncentered) *covariance of loss gradients*, $\mathbf{C}_{\mathcal{L}}^{\mathcal{D}}(\theta) = \mathbb{E}_{(x,y) \sim \mathcal{D}} \left[ \nabla_{\theta} \ell(f_{\theta}(x), y) \, \nabla_{\theta} \ell(f_{\theta}(x), y)^\top \right]$, with $\nabla_{\theta} \ell(f_{\theta}(x), y) = \nabla_{\theta} f_{\theta}(x) \, \nabla_f \ell(f_{\theta}(x), y)$ where $\nabla_f \ell \in \mathbb{R}^K$ is the gradient of the loss with respect to the function. Similar to the covariance of loss gradients, let us define the covariance of function Jacobians, which we will denote by $\mathbf{C}_f^S$ when computed over the samples in set $S$ and which can be expressed as $\mathbf{C}_f^S(\theta) = \frac{1}{|S|} \mathbf{Z}_S(\theta) \mathbf{Z}_S(\theta)^\top$, with $\mathbf{Z}_S := [\nabla_{\theta} f_{\theta}(x_1) \cdots \nabla_{\theta} f_{\theta}(x_{|S|})]^\top \in \mathbb{R}^{p \times K|S|}$.

### 4.1    LOWER BOUND

We employ the following assumption to obtain an appropriate lower bound of the population risk:

**Assumption A1.** *There exists a sample* $(x, y) \sim \mathcal{D}$ *with non-zero probability* $\alpha$ *such that* $\sigma_{(x,y)}^2 := \|\nabla_f \ell(f_{\theta^\star}(x), y)\|^2 > 0$.

Finally, we are in a position to state our main theorem (all the proofs are in Appendix A).

**Theorem 4.** *Under the assumption A1 and taking the limit of external regularization $\lambda \to 0$ and $K = 1$, we obtain the following lower bound on the population risk (eq. 1) at the minimum $\boldsymbol{\theta}^\star$*

$$\mathcal{L}(\boldsymbol{\theta}^\star) \ \geq \ \widetilde{\mathcal{L}}_S(\boldsymbol{\theta}^\star) + \frac{1}{n+1} \frac{\sigma_{min}^2 \, \alpha \, \lambda_{min}\left(\mathbf{C}_{\boldsymbol{f}}^{\widetilde{D}}(\boldsymbol{\theta}^\star)\right)}{\lambda_r\left(\mathbf{H}_{\mathcal{L}}^S(\boldsymbol{\theta}^\star)\right)} \ . \tag{3}$$

*where we consider the convention $\lambda_1 \geq \cdots \geq \lambda_r$ for the eigenvalues with $r := \mathrm{rank}(\mathbf{H}_{\mathcal{L}}^S(\boldsymbol{\theta}^\star))$, and $\sigma_{min}^2$ denotes the minimum $\sigma_{(\boldsymbol{x},\boldsymbol{y})}^2 = \|\nabla_{\boldsymbol{f}}\ell\big(\boldsymbol{f}_{\boldsymbol{\theta}^\star}(\boldsymbol{x}), \boldsymbol{y}\big)\|^2$ over $\widetilde{D}$, i.e., $(\boldsymbol{x}, \boldsymbol{y}) \sim \mathcal{D} : \sigma_{(\boldsymbol{x},\boldsymbol{y})}^2 > 0.$*

The key takeaway of this theorem is that the lower bound on the population risk is inversely proportional to the minimum non-zero eigenvalue $\lambda_r$ of the Hessian at the optimum $\mathbf{H}_{\mathcal{L}}^S(\boldsymbol{\theta}^\star)$, which — as we will see shortly — largely characterizes the double descent like the behaviour of the population risk. Besides, we would like to emphasize that the primary purpose of the lower bounds is to isolate the source of double descent, and thus the practical applicability of the lower bounds — which is in itself an open research area — is of secondary concern. As a result, in our analysis we lower bound the quantity $\sum_{i=1}^{r} \frac{1}{\lambda_r\left(\mathbf{H}_{\mathcal{L}}^S(\boldsymbol{\theta}^\star)\right)}$ by $\frac{1}{\lambda_r\left(\mathbf{H}_{\mathcal{L}}^S(\boldsymbol{\theta}^\star)\right)}$, although it might be of interest to keep the original quantity in a different context.

**Remark.** The assumption A1 just requires the existence of such a point from the true distribution $\mathcal{D}$ with non-zero probability. Notice, otherwise, we would have zero population risk, which is obviously of no interest. The benefit of this assumption is that we can analyze $\mathbf{C}_{\boldsymbol{f}}^{\widetilde{D}}(\boldsymbol{\theta}^\star)$ instead of $\mathbf{C}_{\mathcal{L}}^{\mathcal{D}}(\boldsymbol{\theta}^\star)$ by taking the minimum non-zero $\sigma_{\min}^2$ outside of the corresponding expression.

## 4.2 DOUBLE DESCENT BEHAVIOUR

Having established this lower bound, we will utilize it to demonstrate the existence of the double descent behaviour. Let us consider the case of mean-squared error (MSE) loss, $\ell(\boldsymbol{f}_{\boldsymbol{\theta}}(\boldsymbol{x}), \boldsymbol{y}) = \frac{1}{2}\|\boldsymbol{y} - \boldsymbol{f}_{\boldsymbol{\theta}}(\boldsymbol{x})\|^2$, and where the Hessian of the loss with respect to the function is just the identity, i.e., $\nabla_{\boldsymbol{f}}^2 \ell = \mathbf{I}$. We will employ the following additional assumptions:

**Assumption A2.** *The functional Hessian at the optimum $\boldsymbol{\theta}^\star$ is zero, i.e., $\mathbf{H}_{\boldsymbol{f}}^S(\boldsymbol{\theta}^\star) = \mathbf{0}$ .*

**Assumption A3.** *The minimum non-zero eigenvalue $\lambda_{min}$ of covariance of function Jacobians at optimum $\boldsymbol{\theta}^\star$ is bounded by the corresponding one at initialization $\boldsymbol{\theta}^0$ over some common set $S$, i.e., $A \, \lambda_{min}(\mathbf{C}_{\boldsymbol{f}}^S(\boldsymbol{\theta}^0)) \leq \lambda_{min}(\mathbf{C}_{\boldsymbol{f}}^S(\boldsymbol{\theta}^\star)) \leq B \, \lambda_{min}(\mathbf{C}_{\boldsymbol{f}}^S(\boldsymbol{\theta}^0))$, with constants $0 < A, B < \infty$ .*

**Assumption A4.** *The columns of $\mathbf{Z}_S(\boldsymbol{\theta}^0)$ are sub-Gaussian independent random vectors at initialization $\boldsymbol{\theta}^0$.*

**Note on the assumptions.** Assumption A2 is known from prior works (Sagun et al., 2017; Singh et al., 2021) to hold empirically — in particular, c.f. Figures 5, S2 of Singh et al. (2021) where it is shown that the rank of the functional Hessian converges to 0 when trained sufficiently (besides, c.f. Appendix C.8). Also, in the setting of double descent, the individual losses (and their gradients) are themselves close to zero near the interpolation threshold, thereby making the functional Hessian vanish at the optimum (recall eq. 2). Next, the assumption A3 essentially guarantees that, at the optimum, the minimum non-zero eigenvalue of the covariance of function gradients does not change much relative to that at initialization. Importantly, this is purely for the purposes of the lower bound — *not* something that we impose as a constraint during training (vis-à-vis the NTK regime). The existence of the constants mentioned in this assumption A3 can be theoretically justified by the fact that the map $\mathbf{A} \mapsto \lambda_i(\mathbf{A})$ is Lipschitz-continuous on the space of Hermitian matrices, which follows from Weyl's inequality (Tao, 2012, p. 56). Besides, in Figure 2a, we empirically justify this assumption, and in the adjoining Figure 2b we also validate our lower bound to the population risk throughout the double descent curve (additional details of which can be found in Appendix C.10).

The last assumption may seem more demanding but is mild in comparison to that in prior work (Pennington & Bahri, 2017), where all entries of the covariance of function gradients at the optimum are considered to be independent and identically distributed. In contrast, our assumption just requires sub-gaussianity *only at initialization*. Similar sub-gaussianity assumptions are also common in the regression-based analyses of double descent (Muthukumar et al., 2019; Bartlett et al., 2020). The

benefit of such an assumption is that it allows us to precisely characterize the behaviour of the minimum eigenvalue that appears in Theorem 4 using results from Random Matrix Theory (Vershynin, 2010), and thereby that of double descent, as described in the upcoming Theorem 5. Lastly, before proceeding ahead, let us mention that the Appendix A.4 discusses concrete settings where all the above assumptions hold simultaneously for finite-width neural networks.

**Theorem 5.** *For the MSE loss, under the setting of Theorem 4 and the assumptions A2, A3, A4, the population risk takes the following form and diverges almost surely to $\infty$ at $p = \frac{1}{\sqrt{c}}n$,*

$$\mathcal{L}(\boldsymbol{\theta}^{\star}) \geq \widetilde{\mathcal{L}}_S(\boldsymbol{\theta}^{\star}) + \frac{\sigma_{min}^2 \, \alpha \, A \, \lambda_{min}\left(\mathbf{C}_{\boldsymbol{f}}^{\widetilde{\mathcal{D}}}(\boldsymbol{\theta}^0)\right)}{B \, \|\mathbf{C}_{\boldsymbol{f}}^{\mathcal{D}}(\boldsymbol{\theta}^0)\|_2 \, \left(\sqrt{n} - c\sqrt{p}\right)^2} \ , \ \text{in the asymptotic regime of } p, n \rightarrow \infty \text{ but}$$

*their ratio is a fixed constant, and where $c > 0$ is a constant that depends only on the sub-Gaussian norm of columns of $\mathbf{Z}_S(\boldsymbol{\theta}^0)$.*

**Takeaways.** (a) Firstly, the point where the second descent occurs, i.e, the interpolation threshold, is typically $p \approx n$ in the setting of $K = 1$. So our result from Theorem 5 (which is in this setting) not only implies a divergence at the interpolation threshold, but further illustrates that the complexity term will get smaller as its denominator increases for $p >> n$, — thereby also capturing the overall trend of population risk. (b) Second, while the above result holds in the asymptotic setting, we empirically show that such a behaviour also takes place for $p$ and $n$ as small as a few hundreds or thousands, as discussed in Section 4.3 ahead.

**Interpretation of the interpolation threshold location.** An interesting empirical observation, very briefly alluded to in prior works (Belkin et al., 2019; Greydanus, 2020), is that for the MSE loss with $K$ targets, the interpolation threshold is instead located at $p \approx Kn$. This can be reconciled by looking the rank $r$ of the Hessian at the optimum, as inherently the divergence at the interpolation threshold is because the Hessian's minimum non-zero eigenvalue $\lambda_r$ vanishes. More intuitively, double descent occurs at the transition when the Hessian rank starts being dictated by the # of samples (i.e., when over-parameterized) rather than being governed by the # of parameters (i.e., when under-parameterized), e.g., for MSE when $p \approx Kn$ and note $Kn$ is precisely the rank of the Hessian in the over-parameterized regime ($Kn > p$). As a matter of fact, in practice, there might be redundancies due to either duplicates or linearly dependent features/samples, as well as parameters. For instance, in Figure 15 we show a simple example of linear regression where the interpolation threshold can be moved arbitrarily by changing the extent of redundancy in the design matrix. But, as demonstrated therein, thinking in terms of the Hessian rank can help avoid such inconsistencies.

**Other facets of double descent.** (i) Our analysis additionally explains the empirical observation of *label noise* accentuating the peak (Nakkiran et al., 2019), since the complexity term contains a multiplicative factor of $\sigma_{\min}^2$ which increases proportionately with label noise. (ii) The exact term that appears in our proof, before we take the limit of regularization $\lambda \rightarrow 0$, is $\left(\lambda_r(\mathbf{H}_{\mathcal{L}}^S(\boldsymbol{\theta}^{\star})) + \lambda\right)^{-1}$. This reveals why, when using a *regularization* of a suitable magnitude, double descent is not prominent or disappears, as also noted in (Nakkiran et al., 2019; 2020), — since this term can no longer explode.

### 4.3 EMPIRICAL VERIFICATION

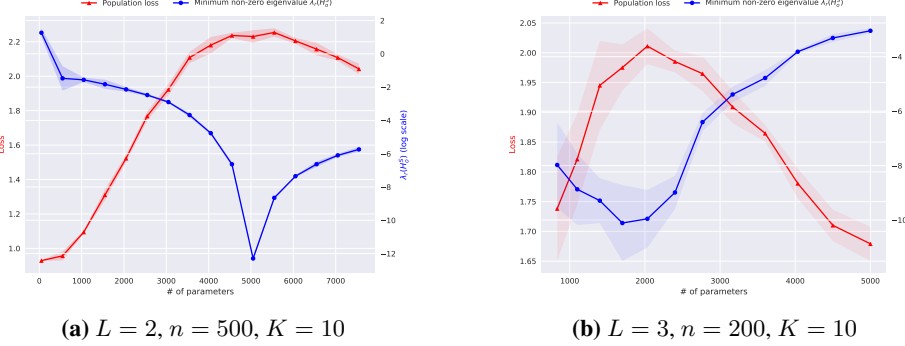

**(a)** $L = 2$, $n = 500$, $K = 10$       **(b)** $L = 3$, $n = 200$, $K = 10$

**Figure 1:** MSE Loss: Behaviour of the population (test) loss and minimum non-zero eigenvalues for the setting of $L = 2$ and $L = 3$ layer networks on downscaled MNIST (Greydanus, 2020). The results are averaged over 5 seeds and the shaded interval denotes the mean $\pm$ std. deviation region.

To empirically demonstrate the validity of our theoretical results, we carry out the entire procedure of obtaining double descent for neural networks. Namely, this involves training a large set of neural networks — with layer widths sampled in regular intervals — for sufficiently many number of epochs. However, in our case, there is another factor which makes this whole process even more arduous — the lower bound depends on the minimum non-zero eigenvalue of the Hessian. As a result, we cannot resort to efficient Hessian approximations based on, say, Hessian-vector products (Pearlmutter, 1994), but rather we need to compute entire Hessian spectrum — which has $\mathcal{O}(p^3)$ computational and memory costs. Hence, we cannot but restrict our empirical investigation to smaller network sizes. Nevertheless, on the positive side we can thoroughly assure the accuracy of our empirical investigations — since we compute the exact Hessian and its spectrum, and that too in FLOAT64 precision.

In terms of the dataset, we primarily utilize MNIST1D (Greydanus, 2020), which is a downscaled version of MNIST yet designed to be significantly harder than the usual version. However, we also present results on CIFAR10 and the usual (easier) MNIST, which alongside other empirical details, can be found in Appendix C. Figure 1 shows the results of running the double descent experiments for the settings of two and three layer fully-connected networks with ReLU activation trained for 5K epochs via SGD. Alongside the population loss (empirically measured on a test set) — which peaks at the interpolation threshold of $p \approx Kn$ — we plot the trend of the minimum non-zero eigenvalue. As predicted by our theory, this eigenvalue indeed tends to zero (note the log scale), in both the settings, around the precise neighborhood where the population loss takes its maximum value.

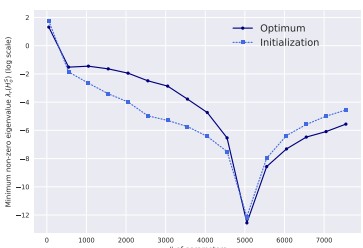

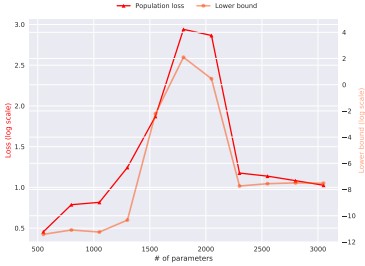

**(a)** Minimum non-zero eigenvalue stays close.     **(b)** Lower bound captures the trend of population loss.

**Figure 2:** Empirical validity of assumption A3 & the lower bound of Theorem 4 in Double Descent.

## 5 THE CASE OF CROSS-ENTROPY

Greydanus (2020) notes that in the case of cross-entropy, the population loss peaks at $p \approx n$ (unlike at $p \approx Kn$ for the MSE loss). To better understand this aspect, let us start by checking the form of the Hessian at the optimum for the cross-entropy loss, which by our assumption A2, will be given by the outer-product term. as in eq. 2. Now, the main difference is that the Hessian of the loss with respect to the function (named 'output-Hessian'), instead of being identity like in MSE, is given by $\nabla_{\boldsymbol{f}}^2 \ell(\boldsymbol{f_\theta}(\boldsymbol{x}), \boldsymbol{y}) = \mathrm{diag}(\boldsymbol{p}) - \boldsymbol{p}\boldsymbol{p}^\top$, where $\boldsymbol{p} = \mathrm{softmax}(\boldsymbol{f_\theta}(\boldsymbol{x}))$ denotes the predicted class-probabilities obtained from applying the softmax operation. In general, this output-Hessian matrix of size $K \times K$, is rank-deficient with rank $K - 1$. Thus, the $\mathrm{rank}(\mathbf{H}_o^S(\boldsymbol{\theta})) = \min(p, (K-1)n)$ for any $\boldsymbol{\theta}$, and the initial surmise would be that the interpolation threshold is at $p \approx (K-1)n$.

But, this is not in line with the stated observation from Greydanus (2020). Hence, let us take another closer look at the Hessian, and in particular, $\nabla_{\boldsymbol{f}}^2 \ell$. Notice that near interpolation, when the loss on individual training samples tends to zero, the predicated class-probability $\boldsymbol{p}$ will tend to 1 for the correct class (as per the training label) and 0 elsewhere. This suggests that Hessian matrix collapses to $\mathbf{0}$ since $\nabla_{\boldsymbol{f}}^2 \ell(\boldsymbol{f_\theta}(\boldsymbol{x}), \boldsymbol{y}) = \mathrm{diag}(\boldsymbol{p}) - \boldsymbol{p}\boldsymbol{p}^\top \to \mathbf{0}$. And indeed, this is true based on our empirical results, displayed in Figure 3 given the network is trained sufficiently long. We clearly observe that the population loss diverges at $p \approx n$, however the test error, although not as conspicuous, still shows a slight peak (similar to the curves in (Nakkiran et al., 2019) without label-noise). Next, from the right sub-figure (plotted in log-scale), it also becomes evident that the entire Hessian spectrum — from the maximum to the minimum eigenvalue — collapses to zero near the interpolation threshold.

**Fact 6.** *Thus, we have that for the cross-entropy loss, near the interpolation threshold of $p \approx n$, the entire Hessian matrix vanishes at the optimum, i.e., $\mathbf{H}_{\mathcal{L}}^S(\boldsymbol{\theta}^\star) = \mathbf{0}$.*

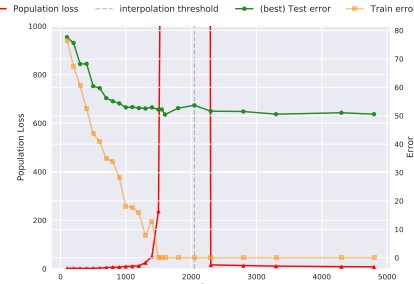 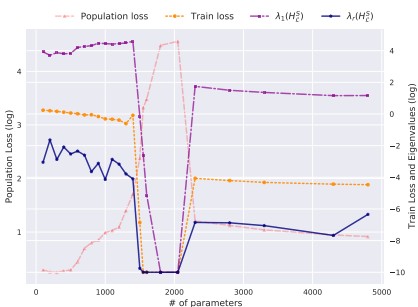

**Figure 3: (Left)** Double descent curve for CE loss and **(right)** log-scale plots of the test and train loss, alongside Hessian eigenvalues, for a one-hidden layer neural network trained for 40K epochs.

Importantly, the above fact (and our empirical results) report such a behaviour only around $p \approx n$, and not well into the over-parameterized regime $p >> n$. Therefore, a straightforward consequence of the above fact and empirical observations is that $\lambda_r(\mathbf{H}_\mathcal{L}^S(\boldsymbol{\theta}^\star)) = 0$ near this interpolation threshold, implying that the population loss in case of cross-entropy loss diverges at $p \approx n$.

**Vanishing Hessian hypothesis.** Figure 3 right, seems to suggest, rather surprisingly, that even the training loss has the same trend as these Hessian eigenvalue statistics. Note, this is not the case that networks were not trained long enough — rather, we run them for $40,000$ epochs. In fact, we even trained models, right to the interpolation threshold, up to $80,000$ epochs. Yet, the training loss only drops by factor of $2$ and is still $> 1e-5$, compared to exact $0$ at machine precision in half the epochs for networks lying in the region where the test loss diverges. This leads us to posit the following hypothesis: *around $p \approx n$, SGD finds critical points with zero Hessian for neural networks trained with CE loss and over-parameterization beyond the interpolation threshold helps completely avoid or significantly decelerate convergence to such critical points.* Further investigation into this hypothesis is beyond the current scope, but forms an exciting direction for future work.

## 6 LEAVE-ONE-OUT ESTIMATE VIA INFLUENCE FUNCTIONS

Leave-one-out (LOO) is also known to provide a reasonable estimate of the population loss (Pontil, 2002). The principle behind it is to leave behind one sample, optimize the model from scratch on the remaining $n-1$ samples, then evaluate the loss on the left-out sample, and finally average over the choice of the left-out sample. In our setting for double descent, we prefer the add-one-in procedure since it directly gives the population loss itself. However, LOO can still be useful from a practical perspective as it relies only on the training set — assuming we can analytically estimate it via some closed-form expression which avoids the need to train $n$ models in the otherwise naive manner.

Similar to the add-one-in procedure from before, leave-one-out can be cast as a slight contamination of the distribution. We can express the new distribution $\mathcal{D}_{n-1}^{\setminus i}$ with the $i$-th sample removed as follows, $\mathcal{D}_{n-1}^{\setminus i} = (1-\epsilon)\mathcal{D}_n + \epsilon \delta_{\boldsymbol{z}_i}$ with $\epsilon = \frac{-1}{n-1}$, where $\mathcal{D}_n$ refers to the original empirical distribution over the training set. Now, we can carry out similar steps like for add-one-in, and derive the change in the parameter estimate as well as the change in loss over the left-out sample. But, here we additionally analyze the effect of *incorporating the second-order influence function*, apart from the usual first-order influences. This provides us with estimates $\text{LOO}^{(1)}$ and $\text{LOO}^{(2)}$, the expressions of which can be found in Appendix A.6.1. As a quick test-bed, we investigate the fidelity of these two approaches relative to the exact formula $\text{LOO}^{\text{LS}}$ in the case of least-squares.

**Theorem 7.** *Consider the particular case of the ordinary-least squares (OLS) with the training inputs gathered into the data matrix $\mathbf{X} \in \mathbb{R}^{n \times d}$ and the targets collected in the vector $\boldsymbol{y} \in \mathbb{R}^n$. Under the assumption that the number of samples $n$ is large enough, we have that $\text{LOO}^{(2)} =$*
$$\text{LOO}^{\text{LS}} = \frac{1}{n}\sum_{i=1}^n \left(\frac{y_i - \boldsymbol{\theta}^\top \boldsymbol{x}_i}{1 - \mathbf{A}_{ii}}\right)^2, \text{ and } \text{LOO}^{(1)} = \frac{1}{n}\sum_{i=1}^n \left(y_i - \boldsymbol{\theta}^\top \boldsymbol{x}_i\right)^2 \frac{\mathbf{A}_{ii}}{1 - \mathbf{A}_{ii}}. \text{ where, } \mathbf{A}_{ii} =$$
*$\boldsymbol{x}_i^\top (\mathbf{X}^\top \mathbf{X})^{-1} \boldsymbol{x}_i$ denotes the $i$-th diagonal entry of the matrix $\mathbf{A} = \mathbf{X}(\mathbf{X}^\top \mathbf{X})^{-1}\mathbf{X}^\top$ (i.e., the so-called 'hat-matrix') and $\boldsymbol{\theta}$ denotes the usual solution of $(\mathbf{X}^\top \mathbf{X})^{-1}\mathbf{X}^\top \boldsymbol{y}$ obtained via OLS.*

This result is reassuring as it shows that LOO expressions from influence function analysis are accurate, and *we recover the least-squares formula as a special case through* $\mathrm{LOO}^{(2)}$. Further,

**Corollary 8.** *For any finite-width neural network, the first and second-order influence function give similar formulas for LOO like that in Theorem 7, but with* $\mathbf{A} = \mathbf{Z}_S(\boldsymbol{\theta}^\star)^\top \left( \mathbf{Z}_S(\boldsymbol{\theta}^\star) \mathbf{Z}_S(\boldsymbol{\theta}^\star)^\top \right)^{-1} \mathbf{Z}_S(\boldsymbol{\theta}^\star)$, *where* $\mathbf{Z}_S(\boldsymbol{\theta}^\star) := [\nabla_{\boldsymbol{\theta}} \boldsymbol{f}_{\boldsymbol{\theta}^\star}(\boldsymbol{x}_1), \cdots, \nabla_{\boldsymbol{\theta}} \boldsymbol{f}_{\boldsymbol{\theta}^\star}(\boldsymbol{x}_n)]$ *and* $\theta^\star$ *are the parameters at convergence for MSE loss.*

Finally, the above result raises a concern about the sub-optimality of first-order influence functions when used in the leave-one-out framework. While this is not necessarily a significant concern for a theoretical analysis, say that of double descent, this can be relevant from a practical viewpoint (Basu et al., 2020). However, an empirical investigation on this front remains beyond the current scope.

## 7 DISCUSSION

**Summary.** We derived an expression of the population risk via influence functions and obtained a lower bound to it — with fairly minimal assumptions — that applies to any finite-width neural network trained with commonly used loss functions. The lower bound is inversely related to the smallest non-zero eigenvalue of the Hessian of the loss at the optimum. When specialized to the MSE loss, this provably exhibits a double descent behaviour in the asymptotic regime and we empirically demonstrated that this holds even in much smaller non-asymptotic settings. We also analyzed the intriguing phenomenology of double descent across other losses — through our Hessian-based framework — which explained existing empirical observations as well as uncovered novel aspects of neural networks near interpolation. Finally, as a by-product, we presented theoretical results for leave-one-out estimation using influence functions in the case of neural networks.

**Related theoretical work on Double Descent.** We carve out a niche in the growing set of studies on double descent by focusing primarily on — *finite-width neural networks*. For the linear/kernel regression setting (or lately, the nearly equivalent two-layer network with frozen hidden-layer), there is a plethora of existing work (Advani et al., 2020; Bartlett et al., 2020; Mei & Montanari, 2019; Muthukumar et al., 2019; Geiger et al., 2020; Ba et al., 2020), that analyzes double descent. Thus, our aim is not to make these existing analyses tighter, but rather to take a step towards developing analyses that directly hold for finite-width neural networks. Therefore, unlike the above works, we do not impose any restrictive assumptions on the structure of neural network, like two-layer networks, or the optimization methods used to train them, like gradient flow. Yet, our work also bears a natural connection between the matrix whose spectrum comes to be of concern in the prior works — the input-covariance or kernel matrix in linear or kernel regression — while that of the outer-product Hessian in our work for MSE loss (which has the same spectrum as the 'empirical' NTK). But our strategy also makes our work applicable to cross-entropy, e.g., where we bring to light the interesting observations near the interpolation threshold.

A closely related recent work, Kuzborskij et al. (2021), links the population risk *for least-squares* to the minimum non-zero eigenvalue of the input covariance matrix — but although via an upperbound, which is insufficient for illustrating double descent. Nevertheless, in analogy to their regression result, they study the minimum eigenvalue of the covariance matrix consisting of penultimatelayer features and conjecture this as a possible extension to neural networks. However, the inputcovariance matrix in least-squares is also the Hessian, and as we have thoroughly established the Hessian (at the optimum) is indeed the relevant object — thus contradicting their conjecture.

**Limitations and directions for future work.** There are many important aspects surrounding double descent that remain unanswered, in the context of neural networks: (a) Analogous to (Hastie et al., 2019) for linear regression, what are the conditions for the global optimum to lie in the overparameterized regime instead of under-parameterized? (b) Given the vanishing Hessian hypothesis, is there a qualification to the regime where the flat-minima generalizes better hypothesis (Hochreiter & Schmidhuber, 1997; Keskar et al., 2016) holds — since the Hessian is the flattest possible here. (d) On the technical side: better characterization of the mentioned cross-entropy phenomenon as well as non-asymptotic results. Overall, we hope that our work will encourage foray into further studies of double descent, that are specifically built for finite-width neural networks.

## REPRODUCIBILITY STATEMENT

- All the omitted proofs to the theoretical results can be found in the Appendix A.
- In regards to empirical results, we provide all the relevant details and additional results in the Appendix C.
- The corresponding code for the experiments is located at https://github.com/sidak/double-descent.

## ACKNOWLEDGEMENTS

We would like to thank Simon Buchholz for proofreading an early draft of the paper. Besides, we thank the members of DA lab for useful comments. Sidak Pal Singh would also like to acknowledge the financial support from Max Planck ETH Center for Learning Systems and the travel support from ELISE (GA no 951847).

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

## A OMITTED PROOFS

**Proposition 2.** *The influence function IF of the M-estimator $\widehat{\boldsymbol{\theta}}_{\mathcal{D}}$ based on the distribution $\mathcal{D}$, evaluated at point $\boldsymbol{z}$, takes the following form:*

$$\mathrm{IF}(\boldsymbol{z}; \widehat{\boldsymbol{\theta}}_{\mathcal{D}}, \mathcal{D}) = -\left[\mathbf{H}_{\mathcal{L}}(\widehat{\boldsymbol{\theta}}_{\mathcal{D}})\right]^{-1} \nabla_{\boldsymbol{\theta}} \ell(\boldsymbol{z}, \widehat{\boldsymbol{\theta}}_D), \tag{4}$$

*where, the Hessian matrix $\mathbf{H}_{\mathcal{L}}(\widehat{\boldsymbol{\theta}}_{\mathcal{D}}) := \nabla_{\boldsymbol{\theta}}^2 \mathcal{L}(\widehat{\boldsymbol{\theta}}_{\mathcal{D}})$ is the matrix of second-derivatives of the loss $\mathcal{L}(\boldsymbol{\theta}) := \underset{z \sim \mathcal{D}}{\mathbb{E}}[\ell(\boldsymbol{z}, \boldsymbol{\theta})]$ with respect to the parameters $\boldsymbol{\theta}$.*

*Proof.* Let $\widehat{\boldsymbol{\theta}}_{\mathcal{D}} = \arg\min_{\boldsymbol{\theta} \in \boldsymbol{\Theta}} \underset{z \sim \mathcal{D}}{\mathbb{E}}[\ell(\boldsymbol{z}, \boldsymbol{\theta})]$. Instead of this formulation, we can also define the estimator as one that satisfies the following implicit equation (assuming the derivative can be moved inside expectation),

$$\underset{z \sim \mathcal{D}}{\mathbb{E}}[\nabla_{\boldsymbol{\theta}} \ell(\boldsymbol{z}, \boldsymbol{\theta})] = 0, \tag{5}$$

which is nothing but the first-order stationary point condition. Now, consider a contaminated distribution $\widetilde{\mathcal{D}} := (1 - \epsilon)\mathcal{D} + \epsilon \delta_{\boldsymbol{z}}$. The corresponding implicit equation, to be satisfied by the estimator $\widehat{\boldsymbol{\theta}}_{\widetilde{\mathcal{D}}}$ corresponding to $\widetilde{\mathcal{D}}$, can be written as follows:

$$\underset{z \sim \widetilde{\mathcal{D}}}{\mathbb{E}}[\nabla_{\boldsymbol{\theta}} \ell(\boldsymbol{z}, \boldsymbol{\theta})] = 0 \tag{6}$$

$$(1 - \epsilon) \underset{z \sim \mathcal{D}}{\mathbb{E}}[\nabla_{\boldsymbol{\theta}} \ell(\boldsymbol{z}, \boldsymbol{\theta})] + \epsilon \nabla_{\boldsymbol{\theta}} \ell(\boldsymbol{z}, \boldsymbol{\theta}) = 0$$

Take the derivative of the above expression with respect to $\epsilon$ (plus interchanging derivative and expectation), we get:

$$\frac{d}{d\epsilon}(1 - \epsilon) \underset{z \sim \mathcal{D}}{\mathbb{E}}[\nabla_{\boldsymbol{\theta}} \ell(\boldsymbol{z}, \boldsymbol{\theta})] = -\frac{d}{d\epsilon} \epsilon \nabla_{\boldsymbol{\theta}} \ell(\boldsymbol{z}, \boldsymbol{\theta})$$

$$\implies -\underset{z \sim \mathcal{D}}{\mathbb{E}}[\nabla_{\boldsymbol{\theta}} \ell(\boldsymbol{z}, \boldsymbol{\theta})] + (1 - \epsilon) \underset{z \sim \mathcal{D}}{\mathbb{E}}[\nabla_{\boldsymbol{\theta}}^2 \ell(\boldsymbol{z}, \boldsymbol{\theta})]\frac{d\boldsymbol{\theta}}{d\epsilon} = -\nabla_{\boldsymbol{\theta}} \ell(\boldsymbol{z}, \boldsymbol{\theta}) - \epsilon \nabla_{\boldsymbol{\theta}}^2 \ell(\boldsymbol{z}, \boldsymbol{\theta})\frac{d\boldsymbol{\theta}}{d\epsilon} \tag{7}$$

Since, $\widehat{\boldsymbol{\theta}}_{\widetilde{\mathcal{D}}}$ satisfies the above eq. 7, let us substitute it in place of $\boldsymbol{\theta}$ and analyze the case for $\epsilon \to 0$. We are left with the following (after removing terms multiplied with $\epsilon$):

$$-\underset{z \sim \mathcal{D}}{\mathbb{E}}[\nabla_{\boldsymbol{\theta}} \ell(\boldsymbol{z}, \widehat{\boldsymbol{\theta}}_{\widetilde{\mathcal{D}}})] + \underset{z \sim \mathcal{D}}{\mathbb{E}}[\nabla_{\boldsymbol{\theta}}^2 \ell(\boldsymbol{z}, \widehat{\boldsymbol{\theta}}_{\widetilde{\mathcal{D}}})]\frac{d\widehat{\boldsymbol{\theta}}_{\widetilde{\mathcal{D}}}}{d\epsilon} = -\nabla_{\boldsymbol{\theta}} \ell(\boldsymbol{z}, \widehat{\boldsymbol{\theta}}_{\widetilde{\mathcal{D}}}) \tag{8}$$

Now, the first term goes to zero as $\epsilon \to 0$, because eq. 6 holds for $\widehat{\boldsymbol{\theta}}_{\widetilde{\mathcal{D}}}$, as shown below :

$$-\underset{z \sim \mathcal{D}}{\mathbb{E}}[\nabla_{\boldsymbol{\theta}} \ell(\boldsymbol{z}, \widehat{\boldsymbol{\theta}}_{\widetilde{\mathcal{D}}})] = -\underset{z \sim \mathcal{D}}{\mathbb{E}}[\nabla_{\boldsymbol{\theta}} \ell(\boldsymbol{z}, \widehat{\boldsymbol{\theta}}_{\widetilde{\mathcal{D}}})] + \underset{z \sim \widetilde{\mathcal{D}}}{\mathbb{E}}[\nabla_{\boldsymbol{\theta}} \ell(\boldsymbol{z}, \widehat{\boldsymbol{\theta}}_{\widetilde{\mathcal{D}}})]$$

$$= \epsilon \left(\nabla_{\boldsymbol{\theta}} \ell(\boldsymbol{z}_0, \widehat{\boldsymbol{\theta}}_{\widetilde{\mathcal{D}}}) - \underset{z \sim \mathcal{D}}{\mathbb{E}}[\nabla_{\boldsymbol{\theta}} \ell(\boldsymbol{z}, \widehat{\boldsymbol{\theta}}_{\widetilde{\mathcal{D}}})]\right).$$

Further, as $\epsilon \to 0$, one can replace $\widehat{\boldsymbol{\theta}}_{\widetilde{\mathcal{D}}}$ by $\widehat{\boldsymbol{\theta}}_{\mathcal{D}}$ in the expressions of the gradient and Hessian of $\ell$. Then, assuming that the Hessian $\mathbf{H}_{\mathcal{L}}(\widehat{\boldsymbol{\theta}}_{\mathcal{D}}) := \nabla_{\boldsymbol{\theta}}^2 \underset{z \sim \mathcal{D}}{\mathbb{E}}[\ell(\boldsymbol{z}, \boldsymbol{\theta})] = \underset{z \sim \mathcal{D}}{\mathbb{E}}\left[\nabla_{\boldsymbol{\theta}}^2 \ell(\boldsymbol{z}, \widehat{\boldsymbol{\theta}}_{\mathcal{D}})\right]$ is invertible, this yields:

$$\mathrm{IF}(\boldsymbol{z}; \widehat{\boldsymbol{\theta}}_{\mathcal{D}}, \mathcal{D}) = -\left[\mathbf{H}_{\mathcal{L}}(\widehat{\boldsymbol{\theta}}_{\mathcal{D}})\right]^{-1} \nabla_{\boldsymbol{\theta}} \ell(\boldsymbol{z}, \widehat{\boldsymbol{\theta}}_D), \tag{9}$$

$\square$

## A.1 EXPRESSION OF THE POPULATION RISK

**Theorem 3.** *Consider the parameter estimator $\widehat{\boldsymbol{\theta}}_S$ based on the set of input samples $S$ of $|S| = n$. Then the population risk $\mathcal{L}(\widehat{\boldsymbol{\theta}}_S) := \mathbb{E}_{\boldsymbol{z} \sim D}\left[\ell(\boldsymbol{z}, \widehat{\boldsymbol{\theta}}_S)\right]$) takes the following form,*

$$\mathcal{L}(\widehat{\boldsymbol{\theta}}_S) = \widetilde{\mathcal{L}}_S(\widehat{\boldsymbol{\theta}}_S) + \frac{1}{n+1} \, Tr\left(\left[\mathbf{H}^S_{\mathcal{L}}(\widehat{\boldsymbol{\theta}}_S) + \lambda\mathbf{I}\right]^{-1}\mathbf{C}^{\mathcal{D}}_{\mathcal{L}}(\widehat{\boldsymbol{\theta}}_S)\right) + \mathcal{O}\left(\frac{1}{n^2}\right), \qquad (10)$$

*where $\widetilde{\mathcal{L}}_S(\widehat{\boldsymbol{\theta}}_S) := \mathbb{E}_{\boldsymbol{z}' \sim \mathcal{D}}\left[\ell\left(\boldsymbol{z}', \widehat{\boldsymbol{\theta}}_{S \cup \{\boldsymbol{z}'\}}\right)\right]$ denotes the expectation of 'one-sample training loss' and $\mathbf{C}^{\mathcal{D}}_{\mathcal{L}}(\widehat{\boldsymbol{\theta}}_S) := \mathbb{E}_{\boldsymbol{z}' \sim \mathcal{D}}\left[\nabla_{\boldsymbol{\theta}}\ell(\boldsymbol{z}', \widehat{\boldsymbol{\theta}}_S)\nabla_{\boldsymbol{\theta}}\ell(\boldsymbol{z}', \widehat{\boldsymbol{\theta}}_S)^{\top}\right]$ is the (uncentered) covariance of loss gradients.*

*Proof.* Let us recall the expression of influence function for the loss on the new sample $\boldsymbol{z}'$,

$$\mathrm{IF}_{\lambda}(\ell_{\boldsymbol{z}'}; \widehat{\boldsymbol{\theta}}_S, \mathcal{D}_n) = -\nabla_{\boldsymbol{\theta}}\ell(\boldsymbol{z}', \widehat{\boldsymbol{\theta}}_S)^{\top}\left[\mathbf{H}^S_{\mathcal{L}}(\widehat{\boldsymbol{\theta}}_S) + \lambda\mathbf{I}\right]^{-1}\nabla_{\boldsymbol{\theta}}\ell(\boldsymbol{z}', \widehat{\boldsymbol{\theta}}_S). \qquad (11)$$

When $|S| = n$ is large enough to ignore $\mathcal{O}(n^{-2})$ terms, the (infinitesimal) definition of influence function is equivalent to using the finite-difference form. Then the change in loss can be expressed as,

$$\ell(\boldsymbol{z}', \widehat{\boldsymbol{\theta}}_{S \cup \{\boldsymbol{z}'\}}) - \ell(\boldsymbol{z}', \widehat{\boldsymbol{\theta}}_S) = \frac{-1}{n+1}\nabla_{\boldsymbol{\theta}}\ell(\boldsymbol{z}', \widehat{\boldsymbol{\theta}}_S)^{\top}\left[\mathbf{H}^S_{\mathcal{L}}(\widehat{\boldsymbol{\theta}}_S) + \lambda\mathbf{I}\right]^{-1}\nabla_{\boldsymbol{\theta}}\ell(\boldsymbol{z}', \widehat{\boldsymbol{\theta}}_S), \qquad (12)$$

where we have multiplied both sides by $\Delta\epsilon = \epsilon - 0 = \frac{1}{n+1}$. We leverage this relation to derive an expression of the test loss as follows. Starting from eq. 12, we average out over the choice of an additional sample $\boldsymbol{z}' \sim \mathcal{D}$:

$$\mathbb{E}_{\boldsymbol{z}' \sim \mathcal{D}}\left[\ell(\boldsymbol{z}', \widehat{\boldsymbol{\theta}}_{S \cup \{\boldsymbol{z}'\}})\right] - \mathbb{E}_{\boldsymbol{z}' \sim \mathcal{D}}\left[\ell(\boldsymbol{z}', \widehat{\boldsymbol{\theta}}_S)\right] =$$

$$\frac{-1}{(n+1)}\mathbb{E}_{\boldsymbol{z}' \sim \mathcal{D}}\left[\nabla_{\boldsymbol{\theta}}\ell(\boldsymbol{z}', \widehat{\boldsymbol{\theta}}_S)^{\top}\left[\mathbf{H}^S_{\mathcal{L}}(\widehat{\boldsymbol{\theta}}_S) + \lambda\mathbf{I}\right]^{-1}\nabla_{\boldsymbol{\theta}}\ell(\boldsymbol{z}', \widehat{\boldsymbol{\theta}}_S)\right].$$

Using properties of the trace and moving expectation inside, we get

$$\mathbb{E}_{\boldsymbol{z}' \sim \mathcal{D}}\left[\ell(\boldsymbol{z}', \widehat{\boldsymbol{\theta}}_S)\right] = \mathbb{E}_{\boldsymbol{z}' \sim \mathcal{D}}\left[\ell(\boldsymbol{z}', \widehat{\boldsymbol{\theta}}_{S \cup \{\boldsymbol{z}'\}})\right] + \frac{1}{n+1}\mathrm{Tr}\left(\left[\mathbf{H}^S_{\mathcal{L}}(\widehat{\boldsymbol{\theta}}_S) + \lambda\mathbf{I}\right]^{-1}\mathbf{C}^{\mathcal{D}}_{\mathcal{L}}(\widehat{\boldsymbol{\theta}}_S)\right), \quad (13)$$

where, $\mathbf{C}^{\mathcal{D}}_{\mathcal{L}}(\boldsymbol{\theta}) := \mathbb{E}_{\boldsymbol{z}' \sim \mathcal{D}}\left[\nabla_{\boldsymbol{\theta}}\ell(\boldsymbol{z}', \boldsymbol{\theta})\nabla_{\boldsymbol{\theta}}\ell(\boldsymbol{z}', \boldsymbol{\theta})^{\top}\right]$.

Now the term on the left-hand side is nothing but the population risk $\mathcal{L}(\widehat{\boldsymbol{\theta}}_S) := \mathbb{E}_{\boldsymbol{z} \sim D}\left[\ell(\boldsymbol{z}, \widehat{\boldsymbol{\theta}}_S)\right]$), and we coin the first term on the right-hand side as the expectation of 'one-sample training loss'.

$\square$

### A.1.1 REMARKS ON THE FIRST-ORDER INFLUENCE FUNCTION USAGE

Let us better understand when using the first-order influence function suffices by analyzing the respective expression for the change in loss. First, let us apply the influence function of the parameter estimator from eq. 9,

$$\mathrm{IF}(\boldsymbol{z}'; \widehat{\boldsymbol{\theta}}_S, S) = -\left[\mathbf{H}^S_{\mathcal{L}}(\widehat{\boldsymbol{\theta}}_S) + \lambda\mathbf{I}\right]^{-1}\nabla_{\boldsymbol{\theta}}\ell(\boldsymbol{z}', \widehat{\boldsymbol{\theta}}_S), \qquad (14)$$

for the add-one-in case of sample $\boldsymbol{z}'$ to the training set $S$ discussed above. Next, we can substitute $\epsilon = \frac{1}{n+1}$, and thereby obtain the change in parameters $\Delta\widehat{\boldsymbol{\theta}}$ as:

$$\Delta\widehat{\boldsymbol{\theta}} = -\frac{1}{n+1}\left[\mathbf{H}^S_{\mathcal{L}}(\widehat{\boldsymbol{\theta}}_S) + \lambda\mathbf{I}\right]^{-1}\nabla_{\boldsymbol{\theta}}\ell(\boldsymbol{z}', \widehat{\boldsymbol{\theta}}_S), \qquad (15)$$

Then, via first-order influences, we get the change in loss over the sample $z'$ as:

$$\Delta\ell^{(1)}(z') = \nabla_{\boldsymbol{\theta}}\ell(z',\widehat{\boldsymbol{\theta}}_S)^{\top}\,\Delta\widehat{\boldsymbol{\theta}} = -\frac{1}{n+1}\mathrm{Tr}\left(\left[\mathbf{H}_{\mathcal{L}}^S(\widehat{\boldsymbol{\theta}}_S) + \lambda\mathbf{I}\right]^{-1}\nabla_{\boldsymbol{\theta}}\ell(z',\widehat{\boldsymbol{\theta}}_S)\nabla_{\boldsymbol{\theta}}\ell(z',\widehat{\boldsymbol{\theta}}_S)^{\top}\right),$$

(16)

In contrast for the second-order influence, we obtain the change in loss over the sample $z'$ as:

$$\Delta\ell^{(2)}(z') = \nabla_{\boldsymbol{\theta}}\ell(z',\widehat{\boldsymbol{\theta}}_S)^{\top}\,\Delta\widehat{\boldsymbol{\theta}} + \frac{1}{2}\Delta\widehat{\boldsymbol{\theta}}^{\top}\nabla_{\boldsymbol{\theta}}^2\ell(z',\widehat{\boldsymbol{\theta}}_S)\,\Delta\widehat{\boldsymbol{\theta}}$$

(17)

$$= -\frac{1}{n+1}\mathrm{Tr}\left(\left[\mathbf{H}_{\mathcal{L}}^S(\widehat{\boldsymbol{\theta}}_S) + \lambda\mathbf{I}\right]^{-1}\nabla_{\boldsymbol{\theta}}\ell(z',\widehat{\boldsymbol{\theta}}_S)\nabla_{\boldsymbol{\theta}}\ell(z',\widehat{\boldsymbol{\theta}}_S)^{\top}\right)$$

$$+ \frac{\mathrm{Tr}\left(\left[\mathbf{H}_{\mathcal{L}}^S(\widehat{\boldsymbol{\theta}}_S) + \lambda\mathbf{I}\right]^{-1}\nabla_{\boldsymbol{\theta}}^2\ell(z',\widehat{\boldsymbol{\theta}}_S)\left[\mathbf{H}_{\mathcal{L}}^S(\widehat{\boldsymbol{\theta}}_S) + \lambda\mathbf{I}\right]^{-1}\nabla_{\boldsymbol{\theta}}\ell(z',\widehat{\boldsymbol{\theta}}_S)\nabla_{\boldsymbol{\theta}}\ell(z',\widehat{\boldsymbol{\theta}}_S)^{\top}\right)}{2(n+1)^2},$$

Later on, we take the expectation over the distribution, i.e., $z' \sim \mathcal{D}$, but it is not relevant for analyzing the scale of the above mentioned change in loss obtained via first or second order influences. Note, the Hessian is itself $\mathcal{O}(1)$ in terms of number of samples $n$, as it is an average of the per-sample Hessians. Also, the numerator in both the above equations is a trace of a $p \times p$ matrix and, overall, the numerator scales as $\mathcal{O}(p)$. Whereas, if we look at the denominator, the extra term in $\Delta\ell^{(2)}$ scales as $\mathcal{O}(n^{-2})$, while the common term in $\Delta\ell^{(2)}$ and $\Delta\ell^{(1)}$ is of $\mathcal{O}(n^{-1})$. Hence, when $n$ is large enough such that $n^2 >> p$, then the $\mathcal{O}(n^{-2})$ terms can be ignored, we can simply consider the first-order influences.

### A.1.2 INTERPRETATION OF THE 'ONE-SAMPLE TRAINING LOSS'

First, note that the first term on the RHS of eq. 13 is not exactly the training loss but rather some related quantity. To see this better, let us rewrite as follows:

$$\mathbb{E}_{z'\sim\mathcal{D}}\left[\ell(z',\widehat{\boldsymbol{\theta}}_{S\cup\{z'\}})\right] = \mathbb{E}_{z'\sim\mathcal{D}}\underbrace{\left[\ell(z',\widehat{\boldsymbol{\theta}}_{S\cup\{z'\}}) - \frac{1}{|S|}\ell(S,\widehat{\boldsymbol{\theta}}_{S\cup\{z'\}})\right]}_{\Delta\mathrm{TR}(z',S\cup\{z'\})} + \mathbb{E}_{z'\sim\mathcal{D}}\left[\frac{1}{|S|}\ell(S,\widehat{\boldsymbol{\theta}}_{S\cup\{z'\}})\right],$$

where, $\ell(S,\boldsymbol{\theta}) = \sum_{i=1}^{|S|}\ell(z_i,\boldsymbol{\theta})$ is the sum of the loss over the samples in $S$. Further, the expression $\Delta\mathrm{TR}(z',S\cup\{z'\})$ refers to the deviation between the loss of a training sample (here, $z'$) relative to the average loss on rest of the training samples. The 'TR' in this symbol refers to this deviation being computed on the given *training* set.

$$\mathbb{E}_{z'\sim\mathcal{D}}\left[\ell(z',\widehat{\boldsymbol{\theta}}_{S\cup\{z'\}})\right] = \mathbb{E}_{z'\sim\mathcal{D}}[\Delta\mathrm{TR}(z',S\cup\{z'\})] + \mathbb{E}_{z'\sim\mathcal{D}}\left[\frac{1}{|S|}\ell(S,\widehat{\boldsymbol{\theta}}_{S\cup\{z'\}})\right]$$

$$= \mathbb{E}_{z'\sim\mathcal{D}}[\Delta\mathrm{TR}(z',S\cup\{z'\})]$$

$$+ \mathbb{E}_{z'\sim\mathcal{D}}\left[\frac{1}{|S|}\left(\ell(S\cup\{z'\},\widehat{\boldsymbol{\theta}}_{S\cup\{z'\}}) - \ell(z',\widehat{\boldsymbol{\theta}}_{S\cup\{z'\}})\right)\right]$$

Notice, the last term in the expression above is the same as the term on the left hand side — albeit with an additional scaling factor of $\frac{1}{|S|}$ and negative sign in front. Rearranging this results in the following equation:

$$\frac{|S|+1}{|S|}\mathbb{E}_{z'\sim\mathcal{D}}\left[\ell(z',\widehat{\boldsymbol{\theta}}_{S\cup\{z'\}})\right] = \mathbb{E}_{z'\sim\mathcal{D}}[\Delta\mathrm{TR}(z',S\cup\{z'\})]$$

$$+ \frac{|S|+1}{|S|}\mathbb{E}_{z'\sim\mathcal{D}}\left[\frac{1}{|S\cup\{z'\}|}\ell(S\cup\{z'\},\widehat{\boldsymbol{\theta}}_{S\cup\{z'\}})\right]$$

The only extra thing we have done is to multiply and divide by $|S\cup\{z'\}| = |S|+1$ in the last term in the right hand side. Notice the rightmost term is an expectation (over $z'$) of the average training

loss of the training set $S \cup \{z'\}$ and to which we assign the shorthand $\mathrm{TR}(S \cup \{z'\})$. Also, it is evident from here that the 'one-sample training loss' is a quantity very much related to the training loss. Lastly, considering the large $|S| = n$ limit, we have $\lim_{n \to \infty} \frac{n+1}{n} = 1$, and which thereby yields:

$$\widetilde{\mathcal{L}}(\widehat{\boldsymbol{\theta}}_S) := \mathop{\mathbb{E}}_{z' \sim \mathcal{D}} \left[ \ell(z', \widehat{\boldsymbol{\theta}}_{S \cup \{z'\}}) \right] = \mathop{\mathbb{E}}_{z' \sim \mathcal{D}} [\Delta \mathrm{TR}(z', S \cup \{z'\})] + \mathop{\mathbb{E}}_{z' \sim \mathcal{D}} [\mathrm{TR}(S \cup \{z'\})]$$

## A.2  LOWER BOUND TO THE POPULATION RISK

**Lemma 9.** *Consider two matrices $\mathbf{A} \in \mathbb{R}^{m \times m}$ and $\mathbf{B} \in \mathbb{R}^{m \times m}$, where $\mathbf{A}$ is symmetric and $\mathbf{B}$ is symmetric and positive semi-definite. Then the following holds,*

$$\lambda_{min}(\mathbf{A}) Tr(\mathbf{B}) \leq Tr(\mathbf{AB}) \leq \lambda_{max}(\mathbf{A}) Tr(\mathbf{B})$$

*Proof.* See Fang et al. (1994). $\qquad\square$

**Theorem 4.** *Under the assumption A1 and taking the limit of external regularization $\lambda \to 0$ and $K = 1$, we obtain the following lower bound on the population risk ( eq. 1), at the minimum $\boldsymbol{\theta}^\star$*

$$\mathcal{L}(\boldsymbol{\theta}^\star) \geq \widetilde{\mathcal{L}}_S(\boldsymbol{\theta}^\star) + \frac{1}{n+1} \frac{\sigma_{min}^2 \, \alpha \, \lambda_{min} \left( \mathbf{C}_{\boldsymbol{f}}^{\widetilde{D}}(\boldsymbol{\theta}^\star) \right)}{\lambda_r \left( \mathbf{H}_{\mathcal{L}}^S(\boldsymbol{\theta}^\star) \right)} \, . \tag{18}$$

*where we consider the convention $\lambda_1 \geq \cdots \geq \lambda_r$ for the eigenvalues with $r := \mathrm{rank}(\mathbf{H}_{\mathcal{L}}^S(\boldsymbol{\theta}^\star))$, and $\sigma_{min}^2$ denotes the minimum $\sigma_{(\boldsymbol{x},\boldsymbol{y})}^2 = \|\nabla_{\boldsymbol{f}} \ell(\boldsymbol{f}_{\boldsymbol{\theta}^\star}(\boldsymbol{x}), \boldsymbol{y})\|^2$ over $\widetilde{D}$, i.e., $(\boldsymbol{x}, \boldsymbol{y}) \sim \mathcal{D} : \sigma_{(\boldsymbol{x},\boldsymbol{y})}^2 > 0$.*

*Proof.* Let us recall the expression for the population risk that we proved in Theorem 3, for $\widehat{\boldsymbol{\theta}}_S = \boldsymbol{\theta}^\star$,

$$\mathcal{L}(\boldsymbol{\theta}^\star) = \widetilde{\mathcal{L}}_S(\boldsymbol{\theta}^\star) + \frac{1}{n+1} \mathrm{Tr} \left( \left[ \mathbf{H}_{\mathcal{L}}^S(\boldsymbol{\theta}^\star) + \lambda \mathbf{I} \right]^{-1} \mathbf{C}_{\mathcal{L}}^{\mathcal{D}}(\boldsymbol{\theta}^\star) \right) + \mathcal{O} \left( \frac{1}{n^2} \right) , \tag{19}$$

In particular, we would like to analyze the complexity term $T := \mathrm{Tr} \left( \left[ \mathbf{H}_{\mathcal{L}}^S(\boldsymbol{\theta}^\star) + \lambda \mathbf{I} \right]^{-1} \mathbf{C}_{\mathcal{L}}^{\mathcal{D}}(\boldsymbol{\theta}^\star) \right)$ on the right-hand side and lower bound it. The full expression of this term $T$ is given by,

$$T = \mathop{\mathbb{E}}_{(\boldsymbol{x},\boldsymbol{y}) \sim \mathcal{D}} \left[ \nabla_{\boldsymbol{\theta}} \ell(\boldsymbol{f}_{\boldsymbol{\theta}^\star}(\boldsymbol{x}), \boldsymbol{y})^\top \left[ \mathbf{H}_{\mathcal{L}}^S(\boldsymbol{\theta}^\star) + \lambda \mathbf{I} \right]^{-1} \nabla_{\boldsymbol{\theta}} \ell(\boldsymbol{f}_{\boldsymbol{\theta}^\star}(\boldsymbol{x}), \boldsymbol{y}) \right] \tag{20}$$

Now using the chain rule, we have that $\nabla_{\boldsymbol{\theta}} \ell(\boldsymbol{f}_{\boldsymbol{\theta}^\star}(\boldsymbol{x}), \boldsymbol{y}) = \nabla_{\boldsymbol{\theta}} \boldsymbol{f}_{\boldsymbol{\theta}^\star}(\boldsymbol{x}) \nabla_{\boldsymbol{f}} \ell(\boldsymbol{f}_{\boldsymbol{\theta}^\star}(\boldsymbol{x}), \boldsymbol{y})$. Then, for $K = 1$, the above equation is equivalent to,

$$T = \mathop{\mathbb{E}}_{(\boldsymbol{x},\boldsymbol{y}) \sim \mathcal{D}} \left[ \underbrace{\nabla_{\boldsymbol{\theta}} \boldsymbol{f}_{\boldsymbol{\theta}^\star}(\boldsymbol{x})^\top \left[ \mathbf{H}_{\mathcal{L}}^S(\boldsymbol{\theta}^\star) + \lambda \mathbf{I} \right]^{-1} \nabla_{\boldsymbol{\theta}} \boldsymbol{f}_{\boldsymbol{\theta}^\star}(\boldsymbol{x})}_{\mathbf{A}_{(\boldsymbol{x},\boldsymbol{y})}} \cdot \underbrace{\|\nabla_{\boldsymbol{f}} \ell(\boldsymbol{f}_{\boldsymbol{\theta}^\star}(\boldsymbol{x}))\|^2}_{\mathbf{B}_{(\boldsymbol{x},\boldsymbol{y})}} \right] . \tag{21}$$

Next, we take the lower bound by considering the minimum over all non-zero $\sigma_{(\boldsymbol{x},\boldsymbol{y})}^2 = \|\nabla_{\boldsymbol{f}} \ell(\boldsymbol{f}_{\boldsymbol{\theta}^\star}(\boldsymbol{x}), \boldsymbol{y})\|^2$,

$$T \geq \sigma_{min}^2 \, \alpha \mathop{\mathbb{E}}_{(\boldsymbol{x},\boldsymbol{y}) \sim \widetilde{D}} \left[ \mathbf{A}_{(\boldsymbol{x},\boldsymbol{y})} \right]$$

$$= \sigma_{min}^2 \, \alpha \, \mathrm{Tr} \left( \left[ \mathbf{H}_{\mathcal{L}}^S(\boldsymbol{\theta}^\star) + \lambda \mathbf{I} \right]^{-1} \mathbf{C}_{\boldsymbol{f}}^{\widetilde{D}}(\boldsymbol{\theta}^\star) \right) \tag{22}$$

where $\sigma_{min}^2 = \min\limits_{(\boldsymbol{x},\boldsymbol{y}) \sim \mathcal{D} : \sigma_{(\boldsymbol{x},\boldsymbol{y})}^2 > 0} \sigma_{(\boldsymbol{x},\boldsymbol{y})}^2$ and whose non-zero probability $\alpha$ is guaranteed by the assumption A1. Finally, in the last line, we use the cyclic property of the trace once again and move the expectation inside the trace, obtaining the covariance of function gradients $\mathbf{C}_{\boldsymbol{f}}^{\widetilde{D}}(\boldsymbol{\theta}^\star)$.

Proceeding further, we again make use of Lemma 9 in the eq. 22, since these matrices are also symmetric positive semi-definite. This yields,

$$T \geq \sigma_{\min}^2 \, \alpha \, \lambda_{\min} \left( \mathbf{C}_{\boldsymbol{f}}^{\widetilde{D}}(\boldsymbol{\theta}^\star) \right) \mathrm{Tr} \left( \left[ \mathbf{H}_{\mathcal{L}}^S(\boldsymbol{\theta}^\star) + \lambda \mathbf{I} \right]^{-1} \right) \tag{23}$$

$$= \sigma_{\min}^2 \, \alpha \, \lambda_{\min} \left( \mathbf{C}_{\boldsymbol{f}}^{\widetilde{D}}(\boldsymbol{\theta}^\star) \right) \sum_{i=1}^{p} \frac{1}{\lambda_i \left( \mathbf{H}_{\mathcal{L}}^S(\boldsymbol{\theta}^\star) \right) + \lambda} \tag{24}$$

$$\tag{25}$$

where, the notation $\lambda_i(\cdot)$ denotes the $i$-th eigenvalue of the corresponding matrix and we will use the convention that for some matrix in $\mathbb{R}^{m \times m}$,

$$\lambda_1 \geq \cdots \geq \lambda_m \, .$$

Let us suppose $r$ denotes the rank of the Hessian at the optimum, i.e., $r := \mathrm{rank}(\mathbf{H}_{\mathcal{L}}^S(\boldsymbol{\theta}^0))$. Since the Hessian is positive semi-definite by the second-order necessary conditions of local minima, all the eigenvalues are non-negative and we can further lower bound the previous expression to $T$ as follows:

$$T \geq \sigma_{\min}^2 \, \alpha \, \lambda_{\min} \left( \mathbf{C}_{\boldsymbol{f}}^{\widetilde{D}}(\boldsymbol{\theta}^\star) \right) \sum_{i=1}^{r} \frac{1}{\lambda_i \left( \mathbf{H}_{\mathcal{L}}^S(\boldsymbol{\theta}^\star) \right) + \lambda}$$

$$\geq \sigma_{\min}^2 \, \alpha \, \lambda_{\min} \left( \mathbf{C}_{\boldsymbol{f}}^{\widetilde{D}}(\boldsymbol{\theta}^\star) \right) \frac{1}{\lambda_r \left( \mathbf{H}_{\mathcal{L}}^S(\boldsymbol{\theta}^\star) \right) + \lambda}$$

where in the second line, we have used the fact that a sum of non-negative numbers can be lower bounded by the maximum summand. The maximum here will correspond to using the inverse of the minimum non-zero eigenvalue $\lambda_r$.

Now, substituting the following lower bound together with expression of population risk and taking the limit of $\lambda \to 0$, finishes the proof.

$$\mathcal{L}(\boldsymbol{\theta}^\star) \geq \widetilde{\mathcal{L}}_S(\boldsymbol{\theta}^\star) + \frac{1}{n+1} \frac{\sigma_{\min}^2 \, \alpha \, \lambda_{\min} \left( \mathbf{C}_{\boldsymbol{f}}^{\widetilde{D}}(\boldsymbol{\theta}^\star) \right)}{\lambda_r \left( \mathbf{H}_{\mathcal{L}}^S(\boldsymbol{\theta}^\star) \right)} \, . \tag{26}$$

$$\square$$

**Note.** For the purposes of this lower bound, we ignored the $\mathcal{O}\left(\frac{1}{n^2}\right)$ part in Theorem 3. This is because they take the form, e.g. for second-order influences as shown in Section A.6.1 and A.1.1.

### A.2.1 ANALOGOUS UPPER BOUND

We can in fact derive an upper bound to the population risk by following analogous steps to that in the lower bound. First, let us recall the expression of the complexity term $T$,

$$T = \mathbb{E}_{(\boldsymbol{x},\boldsymbol{y})\sim\mathcal{D}} \left[ \nabla_{\boldsymbol{\theta}}\ell(\boldsymbol{f}_{\boldsymbol{\theta}^\star}(\boldsymbol{x}),\boldsymbol{y})^\top \left[\mathbf{H}_{\mathcal{L}}^S(\boldsymbol{\theta}^\star) + \lambda\mathbf{I}\right]^{-1} \nabla_{\boldsymbol{\theta}}\ell(\boldsymbol{f}_{\boldsymbol{\theta}^\star}(\boldsymbol{x}),\boldsymbol{y}) \right] \tag{27}$$

The existence of a non-zero residual via the $\sigma_{\min}^2$ assumption A1 also implies that there exists an analogous $\sigma_{\max}^2$, defined as follows:

$$\sigma_{\max}^2 = \max_{(\boldsymbol{x},\boldsymbol{y})\sim\mathcal{D}: \sigma_{(\boldsymbol{x},\boldsymbol{y})}^2 > 0} \sigma_{(\boldsymbol{x},\boldsymbol{y})}^2$$

Thus, we get the following upper bound by also considering the chain rule of $\nabla_{\boldsymbol{\theta}}\ell(\boldsymbol{f}_{\boldsymbol{\theta}^\star}(\boldsymbol{x}),\boldsymbol{y}) = \nabla_{\boldsymbol{\theta}}\boldsymbol{f}_{\boldsymbol{\theta}^\star}(\boldsymbol{x})\nabla_{\boldsymbol{f}}\ell\big(\boldsymbol{f}_{\boldsymbol{\theta}^\star}(\boldsymbol{x}),\boldsymbol{y}\big)$ and repeating similar steps as before,

$$T \le \sigma_{\max}^2\,\alpha\,\mathrm{Tr}\left(\left[\mathbf{H}_{\mathcal{L}}^S(\boldsymbol{\theta}^\star) + \lambda\mathbf{I}\right]^{-1}\mathbf{C}_{\boldsymbol{f}}^{\widetilde{D}}(\boldsymbol{\theta}^\star)\right) \tag{28}$$

Then we can use Lemma 9 as both the matrices inside trace are symmetric positive semi-definite, which gives us our initial upper bound:

$$T \le \sigma_{\max}^2\,\alpha\,\mathrm{Tr}\left(\left[\mathbf{H}_{\mathcal{L}}^S(\boldsymbol{\theta}^\star) + \lambda\mathbf{I}\right]^{-1}\right)\lambda_{\max}\left(\mathbf{C}_{\boldsymbol{f}}^{\widetilde{D}}(\boldsymbol{\theta}^\star)\right) \tag{29}$$

While additional steps can be further carried out, depending on the required context, but the objective of this discussion is to show that many of the steps in our lower bound can be likewise generalized to get a corresponding upper bound.

### A.2.2 EMPIRICAL APPROXIMATIONS OF TIC LIKE EXPRESSIONS

Consider the case when both the Hessian and covariance in the complexity term that shows up in our lower-bound expression,

$$\mathrm{Tr}\left(\left[\mathbf{H}_{\mathcal{L}}^S(\boldsymbol{\theta}^\star) + \lambda\mathbf{I}\right]^{-1}\mathbf{C}_{\boldsymbol{f}}^{\mathcal{D}}(\boldsymbol{\theta}^\star)\right)$$

*are based/approximated on the training set.*

Let us further assume the case of MSE loss and that we are at the optimum, where by A2, $\mathbf{H}_{\mathcal{L}}^S(\boldsymbol{\theta}^\star) = \mathbf{C}_{\boldsymbol{f}}^S(\theta^\star)$. Under these set of assumptions, the term above reduces to,

$$\mathrm{Tr}\left((\mathbf{C}_{\boldsymbol{f}}^S(\boldsymbol{\theta}^\star) + \lambda\mathbf{I})^{-1}\mathbf{C}_{\boldsymbol{f}}^S(\boldsymbol{\theta}^\star)\right)$$

Since we can always express a positive semi-definite matrix as some $\mathbf{Z}\mathbf{Z}^\top$. Let us substitute this in the above expression and consider the limit of regularization $\lambda \to 0$.

$$\lim_{\lambda\to 0} \mathrm{Tr}\left((\mathbf{Z}\mathbf{Z}^\top + \lambda\mathbf{I})^{-1}\mathbf{Z}\mathbf{Z}^\top\right) = \mathrm{Tr}\left(\mathbf{Z}^\dagger\mathbf{Z}\right) \tag{30}$$

where in the last line we have used the result from Tikhonov regularization, and $\mathbf{Z}^\dagger$ denotes the pseudo-inverse of $\mathbf{Z}$. But this expression is nothing but the $\mathrm{rank}(\mathbf{Z})$ and thus shows the ineffectiveness of such an analysis where both the Hessian and the covariance of function gradients are based on the training set.

### A.3 DOUBLE DESCENT BEHAVIOUR

**Theorem 5.** *For the MSE loss, under the setting of Theorem 4 and the assumptions A2, A3, A4, the population risk takes the following form and diverges almost surely to $\infty$ at $p = \frac{1}{\sqrt{c}}n$,*

$$\mathcal{L}(\boldsymbol{\theta}^{\star}) \geq \widetilde{\mathcal{L}}_S(\boldsymbol{\theta}^{\star}) + \frac{\sigma_{min}^2 \, \alpha \, A \, \lambda_{min}\left(\mathbf{C}_{\boldsymbol{f}}^{\widetilde{D}}(\boldsymbol{\theta}^0)\right)}{B \, \|\mathbf{C}_{\boldsymbol{f}}^{\mathcal{D}}(\boldsymbol{\theta}^0)\|_2 \, \left(\sqrt{n} - c\sqrt{p}\right)^2} \; ,$$

*in the asymptotic regime of $p, n \to \infty$ but their ratio is a fixed constant, and where $c > 0$ is a constant that depends only on the sub-Gaussian norm of columns of $\mathbf{Z}_S(\boldsymbol{\theta}^0)$.*

*Proof.* Let us start from the lower bound shown in Theorem 4.

$$\mathcal{L}(\boldsymbol{\theta}^{\star}) \geq \widetilde{\mathcal{L}}_S(\boldsymbol{\theta}^{\star}) + \frac{1}{n+1} \cdot \frac{\sigma_{\min}^2 \, \alpha \, \lambda_{\min}\left(\mathbf{C}_{\boldsymbol{f}}^{\widetilde{D}}(\boldsymbol{\theta}^{\star})\right)}{\lambda_r\left(\mathbf{H}_{\mathcal{L}}^S(\boldsymbol{\theta}^{\star})\right)} \; . \tag{31}$$

For the MSE loss, we have the Hessian $\mathbf{H}_{\mathcal{L}}^S(\boldsymbol{\theta}^{\star}) = \mathbf{C}_{\boldsymbol{f}}^S(\boldsymbol{\theta}^{\star})$, from assumption A2. We then use the assumption A3 to bound the minimum non-zero eigenvalue (here, $\lambda_r$) of $\mathbf{C}_{\boldsymbol{f}}^S(\boldsymbol{\theta}^{\star})$ to the corresponding minimum non-zero eigenvalue at initialization. This results in the following lower bound,

$$\mathcal{L}(\boldsymbol{\theta}^{\star}) \geq \widetilde{\mathcal{L}}_S(\boldsymbol{\theta}^{\star}) + \frac{1}{n+1} \cdot \frac{\sigma_{\min}^2 \, \alpha \, \lambda_{\min}\left(\mathbf{C}_{\boldsymbol{f}}^{\widetilde{D}}(\boldsymbol{\theta}^{\star})\right)}{B \, \lambda_r\left(\mathbf{C}_{\boldsymbol{f}}^S(\boldsymbol{\theta}^0)\right)} \; . \tag{32}$$

Again we utilize the assumption A3 to upper bound the minimum non-zero eigenvalue of $\mathbf{C}_{\boldsymbol{f}}^{\mathcal{D}}(\boldsymbol{\theta}^{\star})$ to the corresponding minimum non-zero eigenvalue at initialization, thus obtaining:

$$\mathcal{L}(\boldsymbol{\theta}^{\star}) \geq \widetilde{\mathcal{L}}_S(\boldsymbol{\theta}^{\star}) + \frac{1}{n+1} \cdot \frac{\sigma_{\min}^2 \, \alpha \, A \, \lambda_{\min}\left(\mathbf{C}_{\boldsymbol{f}}^{\widetilde{D}}(\boldsymbol{\theta}^0)\right)}{B \, \lambda_r\left(\mathbf{C}_{\boldsymbol{f}}^S(\boldsymbol{\theta}^0)\right)} \; . \tag{33}$$

Now notice that, via assumption A4, $\mathbf{C}_{\boldsymbol{f}}^S(\boldsymbol{\theta}^0) = \frac{1}{|S|}\mathbf{Z}_S\mathbf{Z}_S^{\top}$ is a covariance matrix with the columns of $\mathbf{Z}_S$ containing independent, sub-gaussian random vectors in $\mathbb{R}^p$.

Thus, we can leverage the results of Vershynin (2010) on the extremal eigenvalues of covariance matrices. Specifically, given a random matrix, $\mathbf{Z} \in \mathbb{R}^{m \times n}$ whose columns are isotropic, independent, sub-gaussian random vectors in $\mathbb{R}^m$, Vershynin (2010) states that the extremal eigenvalues of $\frac{1}{n}\mathbf{Z}\mathbf{Z}^{\top}$, in the asymptotic regime where $m, n \to \infty$ but their ratio $\frac{m}{n} \to \gamma \in (0, 1]$, we have:

$$\lambda_{\min}\left(\frac{1}{n}\mathbf{Z}\mathbf{Z}^{\top}\right) \to \left(1 - c\sqrt{\frac{m}{n}}\right)^2 \quad \text{and} \quad \lambda_{\max}\left(\frac{1}{n}\mathbf{Z}\mathbf{Z}^{\top}\right) \to \left(1 + c\sqrt{\frac{m}{n}}\right)^2 \quad \text{a.s.,} \tag{34}$$

where $c$ is a constant that depends on the subgaussian norm.

Since the above result holds in the isotropic case, let us first ensure this aspect. Consider the matrix $\widetilde{\mathbf{Z}}_S := \mathbf{C}_{\boldsymbol{f}}^{\mathcal{D}}(\boldsymbol{\theta}^0)^{-\frac{1}{2}}\mathbf{Z}_S$, which is possible since $\mathbf{C}_{\boldsymbol{f}}^{\mathcal{D}}(\boldsymbol{\theta}^0)$ is clearly positive semi-definite and its spectrum being bounded away from zero is a typical assumption, c.f. Du et al. (2019); Nguyen et al. (2021). As a result, the columns of this new matrix $\widetilde{\mathbf{Z}}_S$ are isotropic, besides being independent, subgaussian random vectors. Thus, we apply the above RMT result to the matrix $\frac{1}{|S|}\widetilde{\mathbf{Z}}_S\widetilde{\mathbf{Z}}_S^{\top}$, and we then obtain the following relation on $\lambda_r\left(\mathbf{C}_{\boldsymbol{f}}^S(\boldsymbol{\theta}^0)\right)$:

$$\lambda_r\left(\mathbf{C}_{\boldsymbol{f}}^S(\boldsymbol{\theta}^0)\right) \leq \|\mathbf{C}_{\boldsymbol{f}}^{\mathcal{D}}(\boldsymbol{\theta}^0)\|_2 \left(1 - c\sqrt{\frac{p}{n}}\right)^2 \tag{35}$$

This is because, $s_{\min}(\mathbf{A}\mathbf{B}) \leq \|\mathbf{A}\|_2 \, s_{\min}(\mathbf{B})$, where $s$ denotes the singular value, and follows from using the definition of spectral norm together with min-max characterization of singular values

(see Theorem 3.3.16 in Horn & Johnson (1991) for more). Also, we know that $\lambda_i(\mathbf{M}^\top \mathbf{M}) = \lambda_i(\mathbf{M}\mathbf{M}^\top) = s_i^2(\mathbf{M})$ for some matrix $\mathbf{M} \in \mathbb{R}^{m \times n}$ and for $1 \leq i \leq \min(m, n)$. Therefore, using this for $\mathbf{A} := \mathbf{C}_{\boldsymbol{f}}^{\mathcal{D}}(\boldsymbol{\theta}^0)^{\frac{1}{2}}$, and $\mathbf{B} := \mathbf{C}_{\boldsymbol{f}}^{\mathcal{D}}(\boldsymbol{\theta}^0)^{-\frac{1}{2}}\mathbf{Z}_S$ gives the above bound.

Finally, combining all these together yields,

$$\mathcal{L}(\boldsymbol{\theta}^\star) \geq \widetilde{\mathcal{L}}_S(\boldsymbol{\theta}^\star) + \frac{1}{n+1} \cdot \frac{\sigma_{\min}^2 \, \alpha \, A \, \lambda_{\min}\left(\mathbf{C}_{\boldsymbol{f}}^{\widetilde{D}}(\boldsymbol{\theta}^0)\right)}{B \, \|\mathbf{C}_{\boldsymbol{f}}^{\mathcal{D}}(\boldsymbol{\theta}^0)\|_2 \, \left(1 - c\sqrt{\frac{p}{n}}\right)^2} \; . \tag{36}$$

Then, by a simple rearrangement and noting that we are in the asymptotic regime where $n \to \infty$, we recover our desired lower bound, thus finishing the proof.

$\square$

**Remark 1.** In the over-parameterized case, where $\gamma > 1$, the similar procedure follows by using the random matrix theory result on $\frac{1}{|S|}\mathbf{Z}_S^\top \mathbf{Z}_S$ and the fact that in our lower bound we anyways have minimum non-zero eigenvalue $\lambda_r$.

**Remark 2.** The ratio of the terms $\|\mathbf{C}_{\boldsymbol{f}}^{\mathcal{D}}(\boldsymbol{\theta}^0)\|_2$ and $\lambda_{\min}\left(\mathbf{C}_{\boldsymbol{f}}^{\widetilde{D}}(\boldsymbol{\theta}^0)\right)$ looks like a condition number, but notice the the minimum eigenvalue is for the covariance over $\widetilde{D}$ and not $\mathcal{D}$. Thus, if we are to write in the form of condition number, the above result will take the form:

$$\mathcal{L}(\boldsymbol{\theta}^\star) \geq \widetilde{\mathcal{L}}_S(\boldsymbol{\theta}^\star) + \frac{1}{n+1} \cdot \frac{\sigma_{\min}^2 \, \alpha \, A \, \lambda_{\min}\left(\mathbf{C}_{\boldsymbol{f}}^{\widetilde{D}}(\boldsymbol{\theta}^0)\right) / \lambda_{\min}\left(\mathbf{C}_{\boldsymbol{f}}^{\mathcal{D}}(\boldsymbol{\theta}^0)\right)}{B \, \kappa(\mathbf{C}_{\boldsymbol{f}}^{\mathcal{D}}) \, \left(1 - c\sqrt{\frac{p}{n}}\right)^2} \; . \tag{37}$$

## A.4  One-hidden layer neural network with trained output weights

In this section, we discuss the concrete case of a one-hidden layer neural network, but where only the output layer weights $\boldsymbol{v}$ are trained (akin to a random feature model). Also, for simplicity we assume that the input $\boldsymbol{x} \in \mathbb{R}^d$ is sampled from a sub-Gaussian distribution (i.e. a distribution with a tail decay).

**Linear case.** Let us begin with the case of a linear neural network, and then we can write the network function as follows,

$$f_{\boldsymbol{\theta}}(\boldsymbol{x}) = \boldsymbol{v}^\top \boldsymbol{W} \boldsymbol{x}, \quad \boldsymbol{W} \in \mathbb{R}^{m \times d}.$$

Notice, since $\frac{\partial f}{\partial \boldsymbol{v}_i} = [\boldsymbol{W}\boldsymbol{x}]_i$, the second-derivatives $\frac{\partial^2 f}{\partial \boldsymbol{v}_i^2} = 0, \forall i$. Thus, Assumption A2 holds trivially for all parameter configurations.

Next, note that the columns of $\mathbf{Z}_S$, consisting of the vectors $\nabla_{\boldsymbol{\theta}} f_{\boldsymbol{\theta}}(\boldsymbol{x}) = \frac{\partial f(\boldsymbol{x})}{\partial \boldsymbol{v}}$, are sub-Gaussian random vectors when conditioned on the initialization weights $\boldsymbol{W}$, thus satisfying assumption A4.

Lastly, since the trainable parameters are only $\boldsymbol{v}$ and the matrix $\boldsymbol{W}$ remains fixed, then the covariance of network Jacobians remains fixed as well during training, i.e., $\mathbf{C}_{\boldsymbol{f}}^S(\boldsymbol{\theta}^0) = \frac{1}{|S|}\mathbf{Z}_S(\boldsymbol{\theta}^0)\mathbf{Z}_S(\boldsymbol{\theta}^0)^\top = \frac{1}{|S|}\mathbf{Z}_S(\boldsymbol{\theta}^\star)\mathbf{Z}_S(\boldsymbol{\theta}^\star)^\top = \mathbf{C}_{\boldsymbol{f}}^S(\boldsymbol{\theta}^\star)$. Hence, assumption A3 holds trivially with equality and both constants equal to 1.

**Non-linear case.** Now, consider the general case where we have an elementwise non-linearity $\phi$. So, the network function can be expressed as:

$$f_{\boldsymbol{\theta}}(\boldsymbol{x}) = \boldsymbol{v}^\top \phi(\boldsymbol{W}\boldsymbol{x}), \quad \boldsymbol{W} \in \mathbb{R}^{m \times d}.$$

The functional Hessian is still zero, as the gradient of function with respect to the parameters does not depend on the parameters. Similarly, the covariance of network Jacobian will remain the same during training, as the matrix $\boldsymbol{W}$ is fixed. Thus both assumptions A2, A3 hold. The subgaussian

assumption holds as well for a wide set of activation functions, including for instance the ReLU non-linearity, $\phi(z) = \max(0, z)$. Indeed, each component of the columns of $\mathbf{Z}_S$ is still independent and random. The difference relative to the linear case is that we squash to zero the part of the distribution which is in the second quadrant, which gives a sub-Gaussian distribution as its tail is dominated by a Gaussian. We note that if we assume the input $\boldsymbol{x}$ to be Gaussian, then we get the well-known rectified Gaussian distribution (Harva & Kabán, 2007), which is itself sub-Gaussian.

In summary, these two examples illustrate concrete scenarios of finite-width neural networks *where all the assumptions are clearly satisfied simultaneously*.

### A.5 LEAVE-ONE-OUT DERIVATION

**Influence calculation**   Let us recall the equation which the estimator $\widehat{\boldsymbol{\theta}}_{\tilde{\mathcal{D}}}$ should satisfy:

$$- \mathop{\mathbb{E}}_{\boldsymbol{z} \sim \mathcal{D}}[\nabla_{\boldsymbol{\theta}} \ell(\boldsymbol{z}, \boldsymbol{\theta})] + (1 - \epsilon) \mathop{\mathbb{E}}_{\boldsymbol{z} \sim \mathcal{D}}[\nabla_{\boldsymbol{\theta}}^2 \ell(\boldsymbol{z}, \boldsymbol{\theta})]\frac{d\boldsymbol{\theta}}{d\epsilon} = -\nabla_{\boldsymbol{\theta}} \ell(\boldsymbol{z}_0, \boldsymbol{\theta}) - \epsilon \nabla_{\boldsymbol{\theta}}^2 \ell(\boldsymbol{z}_0, \boldsymbol{\theta})\frac{d\boldsymbol{\theta}}{d\epsilon} \tag{38}$$

Here, $G := (1 - \epsilon)F + \epsilon \delta_{\boldsymbol{z}_0}$. For leave-one-out, we have that $G = D_{n-1}^{\backslash i}$ and $F = D_n$. Without loss of generality, assume that the removed sampled index $i = n$ and so $\boldsymbol{z}_i = \boldsymbol{z}_n$, and consider the shorthand $\tilde{D} = D_{n-1}^{\backslash n}$. Overall, then we are considering the contamination: $\tilde{D} := (1 - \epsilon)D_n + \epsilon \delta_{\boldsymbol{z}_n}$, with $\epsilon = \frac{-1}{n-1}$. Let us substitute all of this back into the equation above:

$$- \mathop{\mathbb{E}}_{\boldsymbol{z} \sim D_n}[\nabla_{\boldsymbol{\theta}} \ell(\boldsymbol{z}, \widehat{\boldsymbol{\theta}}_{\tilde{D}})] + (1 - \epsilon) \mathop{\mathbb{E}}_{\boldsymbol{z} \sim D_n}[\nabla_{\boldsymbol{\theta}}^2 \ell(\boldsymbol{z}, \widehat{\boldsymbol{\theta}}_{\tilde{D}})]\frac{\Delta \boldsymbol{\theta}}{\Delta \epsilon} = -\nabla_{\boldsymbol{\theta}} \ell(\boldsymbol{z}_n, \widehat{\boldsymbol{\theta}}_{\tilde{D}}) - \epsilon \nabla_{\boldsymbol{\theta}}^2 \ell(\boldsymbol{z}_n, \widehat{\boldsymbol{\theta}}_{\tilde{D}})\frac{\Delta \boldsymbol{\theta}}{\Delta \epsilon} \tag{39}$$

$$- \mathop{\mathbb{E}}_{\boldsymbol{z} \sim D_n}[\nabla_{\boldsymbol{\theta}} \ell(\boldsymbol{z}, \widehat{\boldsymbol{\theta}}_{\tilde{D}})] + \mathop{\mathbb{E}}_{\boldsymbol{z} \sim \tilde{D}}[\nabla_{\boldsymbol{\theta}}^2 \ell(\boldsymbol{z}, \widehat{\boldsymbol{\theta}}_{\tilde{D}})]\frac{\Delta \boldsymbol{\theta}}{\Delta \epsilon} = -\nabla_{\boldsymbol{\theta}} \ell(\boldsymbol{z}_n, \widehat{\boldsymbol{\theta}}_{\tilde{D}}) \tag{40}$$

$$- \mathop{\mathbb{E}}_{\boldsymbol{z} \sim D_n}[\nabla_{\boldsymbol{\theta}} \ell(\boldsymbol{z}, \widehat{\boldsymbol{\theta}}_{\tilde{D}})] + \mathop{\mathbb{E}}_{\boldsymbol{z} \sim D_n}[\nabla_{\boldsymbol{\theta}}^2 \ell(\boldsymbol{z}, \widehat{\boldsymbol{\theta}}_{\tilde{D}})]\frac{\Delta \boldsymbol{\theta}}{\Delta \epsilon} = -\nabla_{\boldsymbol{\theta}} \ell(\boldsymbol{z}_n, \widehat{\boldsymbol{\theta}}_{\tilde{D}}) - \epsilon \nabla_{\boldsymbol{\theta}}^2 \ell(\boldsymbol{z}_n, \widehat{\boldsymbol{\theta}}_{\tilde{D}})\frac{\Delta \boldsymbol{\theta}}{\Delta \epsilon}$$
$$+ \epsilon \mathop{\mathbb{E}}_{\boldsymbol{z} \sim D_n}[\nabla_{\boldsymbol{\theta}}^2 \ell(\boldsymbol{z}, \widehat{\boldsymbol{\theta}}_{\tilde{D}})]\frac{\Delta \boldsymbol{\theta}}{\Delta \epsilon}$$
$$\implies - \mathop{\mathbb{E}}_{\boldsymbol{z} \sim D_n}[\nabla_{\boldsymbol{\theta}} \ell(\boldsymbol{z}, \widehat{\boldsymbol{\theta}}_{\tilde{D}})] + \mathop{\mathbb{E}}_{\boldsymbol{z} \sim D_n}[\nabla_{\boldsymbol{\theta}}^2 \ell(\boldsymbol{z}, \widehat{\boldsymbol{\theta}}_{\tilde{D}})]\frac{\Delta \boldsymbol{\theta}}{\Delta \epsilon} = -\nabla_{\boldsymbol{\theta}} \ell(\boldsymbol{z}_n, \widehat{\boldsymbol{\theta}}_{\tilde{D}}) + \epsilon \mathop{\mathbb{E}}_{\boldsymbol{z} \sim \tilde{D}}[\nabla_{\boldsymbol{\theta}}^2 \ell(\boldsymbol{z}, \widehat{\boldsymbol{\theta}}_{\tilde{D}})]\frac{\Delta \boldsymbol{\theta}}{\Delta \epsilon}$$

The last equation holds because notice that in the second term $\boldsymbol{z} \sim \widehat{\boldsymbol{\theta}}_{\tilde{D}}$ (where $\tilde{D} = (1 - \epsilon)D_n + \epsilon \delta_{\boldsymbol{z}_n}$) and not $\boldsymbol{z} \sim D_n$. Since, both $D_n$ and $\tilde{D}$ are empirical distributions, we can compute the expectation as a finite sum,

$$- \sum_{i=1}^{n} \frac{1}{n}\nabla_{\boldsymbol{\theta}} \ell(\boldsymbol{z}_i, \widehat{\boldsymbol{\theta}}_{\tilde{D}}) + \left(\frac{1}{n-1}\sum_{i=1}^{n-1}\nabla_{\boldsymbol{\theta}}^2 \ell(\boldsymbol{z}_i, \widehat{\boldsymbol{\theta}}_{\tilde{D}})\right)\frac{\Delta \boldsymbol{\theta}}{\Delta \epsilon} = -\nabla_{\boldsymbol{\theta}} \ell(\boldsymbol{z}_n, \widehat{\boldsymbol{\theta}}_{\tilde{D}}) \tag{41}$$

We can split the term involving the gradient of the loss into two parts: one based on the $n-1$ samples (of which $\widehat{\boldsymbol{\theta}}_{\tilde{D}}$ is also the parameter estimator) and the other based on the left-out sample with index $n$.

$$\underbrace{- \frac{n-1}{n}\sum_{i=1}^{n-1}\frac{1}{n-1}\nabla_{\boldsymbol{\theta}} \ell(\boldsymbol{z}_i, \widehat{\boldsymbol{\theta}}_{\tilde{D}})}_{=0} - \frac{1}{n}\nabla_{\boldsymbol{\theta}} \ell(\boldsymbol{z}_n, \widehat{\boldsymbol{\theta}}_{\tilde{D}}) + \left(\frac{1}{n-1}\sum_{i=1}^{n-1}\nabla_{\boldsymbol{\theta}}^2 \ell(\boldsymbol{z}_i, \widehat{\boldsymbol{\theta}}_{\tilde{D}})\right)\frac{\Delta \boldsymbol{\theta}}{\Delta \epsilon} = -\nabla_{\boldsymbol{\theta}} \ell(\boldsymbol{z}_n, \widehat{\boldsymbol{\theta}}_{\tilde{D}})$$

$$\tag{42}$$

The first term is zero since $\widehat{\boldsymbol{\theta}}_{\tilde{D}}$ is the parameter estimator of the first $n-1$ samples, and will satisfy the first-order stationarity conditions. Now, rearranging terms results in:

$$\left(\frac{1}{n-1}\sum_{i=1}^{n-1}\nabla_{\boldsymbol{\theta}}^2\ell(z_i,\widehat{\boldsymbol{\theta}}_{\tilde{D}})\right)\frac{\Delta\boldsymbol{\theta}}{\Delta\epsilon} = -\frac{n-1}{n}\nabla_{\boldsymbol{\theta}}\ell(z_n,\widehat{\boldsymbol{\theta}}_{\tilde{D}})$$

The term on the left is nothing but the Hessian computed over $\tilde{D}$ or in other words, the first $n-1$ samples. Let us denote it by $\mathbf{H}_{\mathcal{L}}^{\backslash n}(\boldsymbol{\theta}_{\tilde{D}})$, where the $\backslash n$ in the superscript emphasizes that the Hessian is computed over samples excluding the $n$-th sample. Further, $\Delta\epsilon = \epsilon - 0 = \epsilon = -\frac{1}{n-1}$. Also, $\Delta\boldsymbol{\theta} = \widehat{\boldsymbol{\theta}}_{\tilde{D}} - \widehat{\boldsymbol{\theta}}_D$. Thus we obtain that the change in parameter estimator (i.e., influence on the parameter estimator) is,

$$\Delta\boldsymbol{\theta} = \frac{1}{n}\mathbf{H}_{\mathcal{L}}^{\backslash n}(\widehat{\boldsymbol{\theta}}_{\tilde{D}})^{\dagger}\nabla_{\boldsymbol{\theta}}\ell(z_n,\widehat{\boldsymbol{\theta}}_{\tilde{D}}) \tag{43}$$

**Linearizing the loss at the left-out sample.**  In order to look at the change in loss evaluated on the $n$-th sample (left-out sample), we can consider linearizing the loss at $\widehat{\boldsymbol{\theta}}_{\tilde{D}}$.

$$\Delta\ell_n = \frac{1}{n}\nabla_{\boldsymbol{\theta}}\ell(z_n,\widehat{\boldsymbol{\theta}}_{\tilde{D}})^{\top}\mathbf{H}_{\mathcal{L}}^{\backslash n}(\widehat{\boldsymbol{\theta}}_{\tilde{D}})^{\dagger}\nabla_{\boldsymbol{\theta}}\ell(z_n,\widehat{\boldsymbol{\theta}}_{\tilde{D}})$$

But, $\Delta\ell_n = \ell(z_n,\widehat{\boldsymbol{\theta}}_{\tilde{D}}) - \ell(z_n,\widehat{\boldsymbol{\theta}}_D)$. And, so we have:

$$\ell(z_n,\widehat{\boldsymbol{\theta}}_{\tilde{D}}) = \ell(z_n,\widehat{\boldsymbol{\theta}}_D) + \frac{1}{n}\nabla_{\boldsymbol{\theta}}\ell(z_n,\widehat{\boldsymbol{\theta}}_{\tilde{D}})^{\top}\mathbf{H}_{\mathcal{L}}^{\backslash n}(\widehat{\boldsymbol{\theta}}_{\tilde{D}})^{\dagger}\nabla_{\boldsymbol{\theta}}\ell(z_n,\widehat{\boldsymbol{\theta}}_{\tilde{D}})$$

Now, we would like to average over the choice of the left-out sample $n$. However, we have to be careful and remember that $\tilde{D} = D_{n-1}^{\backslash i}$ depends on the particular left-out sample $i$. Thus, the full expression we get is the following:

$$\text{LOO}_S = \mathcal{L}_S(\widehat{\boldsymbol{\theta}}_D) + \frac{1}{n^2}\sum_{i=1}^{n}\nabla_{\boldsymbol{\theta}}\ell\left(z_i,\widehat{\boldsymbol{\theta}}_{D_{n-1}^{\backslash i}}\right)^{\top}\mathbf{H}_{\mathcal{L}}^{\backslash i}\left(\widehat{\boldsymbol{\theta}}_{D_{n-1}^{\backslash i}}\right)^{\dagger}\nabla_{\boldsymbol{\theta}}\ell\left(z_i,\widehat{\boldsymbol{\theta}}_{D_{n-1}^{\backslash i}}\right)$$

Let us recall the change in parameters with the leave-one-out contamination. From eq. 43, this was

$$\Delta\boldsymbol{\theta} = \frac{1}{n}\mathbf{H}_{\mathcal{L}}^{\backslash n}(\widehat{\boldsymbol{\theta}}_{\tilde{D}})^{\dagger}\nabla_{\boldsymbol{\theta}}\ell(z_n,\widehat{\boldsymbol{\theta}}_{\tilde{D}})$$

where, $\tilde{D} := D_{n-1}^{\backslash n} = (1-\epsilon)D_n + \epsilon\delta_{z_n}$, with $\epsilon = \frac{-1}{n-1}$ and assuming $n$ is large enough.

Under this asymptotic $n$ scenario, it is reasonable to estimate the Hessian $\mathbf{H}_{\mathcal{L}}^{\backslash n}$ and $\nabla_{\boldsymbol{\theta}}\ell$ at the distribution $D_n$ instead of $\tilde{D} := D_{n-1}^{\backslash n}$. We thereby get the following (where we also make the dependence of $\Delta\boldsymbol{\theta}$ explicit on the particular left-out sample index),

$$\begin{aligned}\Delta\boldsymbol{\theta}^{(n)} = \widehat{\boldsymbol{\theta}}_{D_{n-1}^{\backslash n}} - \widehat{\boldsymbol{\theta}}_S &= \frac{1}{n}\mathbf{H}_{\mathcal{L}}^{\backslash n}(\widehat{\boldsymbol{\theta}}_S)^{\dagger}\nabla_{\boldsymbol{\theta}}\ell(z_n,\widehat{\boldsymbol{\theta}}_{D_n}) \\ &= \frac{1}{n}\left(\frac{1}{n-1}\sum_{i=1}^{n-1}\nabla_{\boldsymbol{\theta}}^2\ell\left(z_i,\widehat{\boldsymbol{\theta}}_S\right)\right)^{\dagger}\nabla_{\boldsymbol{\theta}}\ell(z_n,\widehat{\boldsymbol{\theta}}_{D_n}) \\ &= \left(\sum_{i=1}^{n-1}\nabla_{\boldsymbol{\theta}}^2\ell\left(z_i,\widehat{\boldsymbol{\theta}}_S\right)\right)^{\dagger}\nabla_{\boldsymbol{\theta}}\ell(z_n,\widehat{\boldsymbol{\theta}}_{D_n})\end{aligned} \tag{44}$$

Now, we have a nice expression — in the sense that everything on the right-hand side is in terms of the parameters $\widehat{\boldsymbol{\theta}}_S$ obtained by training with the usual (empirical) distribution $D_n$. Since, we are interested in the leave-one-out loss, let us analyze how the above change in the estimated parameters (i.e., those obtained at convergence) brings about a change in the loss incurred on the $n$-th sample itself $\Delta\ell^{(n)}$. To this end, we consider a **quadratic approximation** of the $n$-th sample loss at $\widehat{\boldsymbol{\theta}}_S$, as shown below (this can alternatively be thought of :

$$\ell(\boldsymbol{z}_n, \widehat{\boldsymbol{\theta}}_{D_{n-1}^{\backslash n}}) = \ell(\boldsymbol{z}_n, \widehat{\boldsymbol{\theta}}_S) + \nabla_{\boldsymbol{\theta}}\ell(\boldsymbol{z}_n, \widehat{\boldsymbol{\theta}}_S)^\top \Delta\boldsymbol{\theta}^{(n)} + \frac{1}{2}\Delta\boldsymbol{\theta}^{(n)\top}\nabla_{\boldsymbol{\theta}}^2\ell(\boldsymbol{z}_n, \widehat{\boldsymbol{\theta}}_S)\Delta\boldsymbol{\theta}^{(n)} + \mathcal{O}(\|\Delta\boldsymbol{\theta}^{(n)}\|^3). \tag{45}$$

Next, we consider this procedure for every choice of the left-out-sample, and then average out to get the expression of overall leave-one-out loss. We will also assume $\|\Delta\boldsymbol{\theta}^{(n)}\|^3$ and thus ignore the third-order term.

$$\begin{aligned}
\text{LOO} &= \frac{1}{n}\sum_{i=1}^n \ell\left(\boldsymbol{z}_i, \widehat{\boldsymbol{\theta}}_{D_{n-1}^{\backslash i}}\right) \\
&= \frac{1}{n}\sum_{i=1}^n \ell(\boldsymbol{z}_i, \widehat{\boldsymbol{\theta}}_S) + \nabla_{\boldsymbol{\theta}}\ell(\boldsymbol{z}_i, \widehat{\boldsymbol{\theta}}_S)^\top\Delta\boldsymbol{\theta}^{(i)} + \frac{1}{2}\Delta\boldsymbol{\theta}^{(i)\top}\nabla_{\boldsymbol{\theta}}^2\ell(\boldsymbol{z}_i, \widehat{\boldsymbol{\theta}}_S)\Delta\boldsymbol{\theta}^{(i)} \\
&= \mathcal{L}(\widehat{\boldsymbol{\theta}}_S) + \frac{1}{n}\sum_{i=1}^n \underbrace{\nabla_{\boldsymbol{\theta}}\ell(\boldsymbol{z}_i, \widehat{\boldsymbol{\theta}}_S)^\top\Delta\boldsymbol{\theta}^{(i)}}_{=:B_i} + \frac{1}{2n}\sum_{i=1}^n \underbrace{\Delta\boldsymbol{\theta}^{(i)\top}\nabla_{\boldsymbol{\theta}}^2\ell(\boldsymbol{z}_i, \widehat{\boldsymbol{\theta}}_S)\Delta\boldsymbol{\theta}^{(i)}}_{=:C_i} \tag{46}
\end{aligned}$$

To go further, we need to plug in the change in parameters computed in eq. 44 above. However, we first establish the validity of this general expression via the following result.

### A.6 LEAVE-ONE-OUT FORMULA PROOFS

**Theorem 7.** *Consider the particular case of the ordinary least-squares, where $\ell(\boldsymbol{z}, \boldsymbol{\theta}) := \ell((\boldsymbol{x}, y), \boldsymbol{\theta}) = (y - \boldsymbol{\theta}^\top\boldsymbol{x})^2$. We assume that we are given a training set $S = \{(\boldsymbol{x}_i, y_i)\}_{i=1}^n$ of points sampled i.i.d from the uniform empirical distribution $D_n$. Let the inputs be gathered into the data matrix $\mathbf{X} \in \mathbb{R}^{n \times d}$ where $d$ is the input dimension and the targets are collected in the vector $\boldsymbol{y} \in \mathbb{R}^n$. Under the assumption that the number of samples $n$ is large enough such that $n \approx n - 1$, we have that the leave-one-out expression from eq. 46 (with parameter change given in eq. 44) is equal to the widely-known closed-form formula of leave-one-out for least-squares $\text{LOO}^{LS}$. Mathematically, we have that,*

$$\text{LOO} = \text{LOO}^{LS} = \frac{1}{n}\sum_{i=1}^n \left(\frac{y_i - \boldsymbol{\theta}^\top\boldsymbol{x}_i}{1 - \mathbf{A}_{ii}}\right)^2.$$

*where, $\mathbf{A}_{ii} = \boldsymbol{x}_i^\top(\mathbf{X}^\top\mathbf{X})^{-1}\boldsymbol{x}_i$ denotes the $i$-th diagonal entry of the matrix $\mathbf{A} = \mathbf{X}(\mathbf{X}^\top\mathbf{X})^{-1}\mathbf{X}^\top$ (i.e., the so-called 'hat-matrix') and $\boldsymbol{\theta}$ denotes the usual solution of $\boldsymbol{\theta} = (\mathbf{X}^\top\mathbf{X})^{-1}\mathbf{X}^\top\boldsymbol{y}$ obtained via ordinary least-squares.*

*Proof.* We will first separately analyze the terms in the summation corresponding to the first-order and second-order parts in the eq. 46, which have been accorded the shorthand $B_i$ and $C_i$ respectively. Also, to make expressions simpler, we will define the residual as $r_i := \boldsymbol{\theta}^\top\boldsymbol{x}_i - y_i$ and denote the matrix $\mathbf{X}^\top\mathbf{X}$ by $\boldsymbol{\Sigma}$.

Let us write down the individual loss gradients and Hessian in this case. We have

$$\nabla\ell_i := \nabla_{\boldsymbol{\theta}}\ell((\boldsymbol{x}_i, y_i), \boldsymbol{\theta}) = 2r_i\boldsymbol{x}_i,$$

$$\text{and,}\ \nabla^2 \boldsymbol{\ell}_i := \nabla^2_{\boldsymbol{\theta}} \ell((\boldsymbol{x}_i, y_i), \boldsymbol{\theta}) = 2\,\boldsymbol{x}_i \boldsymbol{x}_i^\top\,.$$

The term $B_i$ in summation corresponding to the first-order part can be computed as follows.

$$
\begin{aligned}
B_i := \nabla \boldsymbol{\ell}_i^\top \Delta \boldsymbol{\theta}^{(i)} = \nabla \boldsymbol{\ell}_i^\top \left( \sum_{j \neq i} \nabla^2 \boldsymbol{\ell}_j \right)^{-1} \nabla \boldsymbol{\ell}_i &= 2r_i\,\boldsymbol{x}_i^\top \left( 2 \sum_{j \neq i} \boldsymbol{x}_j \boldsymbol{x}_j^\top \right)^{-1} 2r_i\,\boldsymbol{x}_i \\
&= 2r_i^2\,\boldsymbol{x}_i^\top \left( \mathbf{X}^\top \mathbf{X} - \boldsymbol{x}_i \boldsymbol{x}_i^\top \right)^{-1} \boldsymbol{x}_i \\
&= 2r_i^2\,\boldsymbol{x}_i^\top \left( \boldsymbol{\Sigma} - \boldsymbol{x}_i \boldsymbol{x}_i^\top \right)^{-1} \boldsymbol{x}_i \\
&= 2r_i^2\,\boldsymbol{x}_i^\top \left( \boldsymbol{\Sigma}^{-1} + \frac{\boldsymbol{\Sigma}^{-1} \boldsymbol{x}_i \boldsymbol{x}_i^\top \boldsymbol{\Sigma}^{-1}}{1 - \boldsymbol{x}_i^\top \boldsymbol{\Sigma}^{-1} \boldsymbol{x}_i} \right) \boldsymbol{x}_i \\
&= 2r_i^2 \left( \mathbf{A}_{ii} + \frac{(\mathbf{A}_{ii})^2}{1 - \mathbf{A}_{ii}} \right) \\
&= 2r_i^2 \frac{\mathbf{A}_{ii}}{1 - \mathbf{A}_{ii}}\,.
\end{aligned}
$$

Essentially, we have used the Sherman-Morrison-Woodbury formula in the fourth line above and rest is mere manipulation. Similarly, we compute the term $C_i$ from the second-order part:

$$
\begin{aligned}
C_i := \Delta \boldsymbol{\theta}^{(i)\top} \nabla^2 \boldsymbol{\ell}_i \Delta \boldsymbol{\theta}^{(i)} = \nabla \boldsymbol{\ell}_i^\top \left( \sum_{j \neq i} \nabla^2 \boldsymbol{\ell}_j \right)^{-1} \nabla^2 \boldsymbol{\ell}_i \left( \sum_{j \neq i} \nabla^2 \boldsymbol{\ell}_j \right)^{-1} \nabla \boldsymbol{\ell}_i \\
= 2r_i^2\,\boldsymbol{x}_i^\top \left( \boldsymbol{\Sigma} - \boldsymbol{x}_i \boldsymbol{x}_i^\top \right)^{-1} \boldsymbol{x}_i \boldsymbol{x}_i^\top \left( \boldsymbol{\Sigma} - \boldsymbol{x}_i \boldsymbol{x}_i^\top \right)^{-1} \boldsymbol{x}_i \\
= 2r_i^2 \left( \boldsymbol{x}_i^\top \left( \boldsymbol{\Sigma} - \boldsymbol{x}_i \boldsymbol{x}_i^\top \right)^{-1} \boldsymbol{x}_i \right)^2 \\
= 2r_i^2 \left( \frac{\mathbf{A}_{ii}}{1 - \mathbf{A}_{ii}} \right)^2
\end{aligned}
$$

In the last line, we just reuse the computation that we already did in the previous part for $\boldsymbol{x}_i^\top \left( \boldsymbol{\Sigma} - \boldsymbol{x}_i \boldsymbol{x}_i^\top \right)^{-1} \boldsymbol{x}_i$. Finally, let us combine everything we have got with eq. 46

$$
\begin{aligned}
\text{LOO} = \mathcal{L}(\boldsymbol{\theta}) + \frac{1}{n} \sum_{i=1}^n B_i + \frac{1}{2n} \sum_{i=1}^n C_i \\
= \frac{1}{n} \sum_{i=1}^n r_i^2 + \frac{1}{n} \sum_{i=1}^n 2r_i^2 \frac{\mathbf{A}_{ii}}{1 - \mathbf{A}_{ii}} + \frac{1}{n} \sum_{i=1}^n r_i^2 \left( \frac{\mathbf{A}_{ii}}{1 - \mathbf{A}_{ii}} \right)^2 \\
= \frac{1}{n} \sum_{i=1}^n r_i^2 \left( 1 + 2\frac{\mathbf{A}_{ii}}{1 - \mathbf{A}_{ii}} + \left( \frac{\mathbf{A}_{ii}}{1 - \mathbf{A}_{ii}} \right)^2 \right) \\
= \frac{1}{n} \sum_{i=1}^n r_i^2 \left( 1 + \frac{\mathbf{A}_{ii}}{1 - \mathbf{A}_{ii}} \right)^2 \\
= \frac{1}{n} \sum_{i=1}^n \left( \frac{r_i}{1 - \mathbf{A}_{ii}} \right)^2\,.
\end{aligned}
$$

Thus, remembering our shorthand $r_i := \boldsymbol{\theta}^\top \boldsymbol{x}_i - y_i$, we conclude our proof.

$\square$

**Corollary 8.** *For any finite-width neural network, the first and second-order influence function give similar formulas for LOO like that in Theorem 7, but with* $\mathbf{A} = \mathbf{Z}_S(\boldsymbol{\theta}^\star)^\top \left(\mathbf{Z}_S(\boldsymbol{\theta}^\star)\mathbf{Z}_S(\boldsymbol{\theta}^\star)^\top\right)^{-1} \mathbf{Z}_S(\boldsymbol{\theta}^\star)$, *where* $\mathbf{Z}_S(\boldsymbol{\theta}^\star) := [\nabla_{\boldsymbol{\theta}} f_{\boldsymbol{\theta}^\star}(\boldsymbol{x}_1), \cdots, \nabla_{\boldsymbol{\theta}} f_{\boldsymbol{\theta}^\star}(\boldsymbol{x}_n)]$ *and* $\theta^\star$ *are the parameters at convergence for MSE loss.*

*Proof.* Simply replace $\boldsymbol{x}_i$ to $\nabla_{\boldsymbol{\theta}} f_{\boldsymbol{\theta}}(\boldsymbol{x}_i)$ in the proof of Theorem 7 and repeat the procedure, since all the steps hold under the assumption of MSE loss and considering the Hessian $\mathbf{H}_{\mathcal{L}}^S(\boldsymbol{\theta}^\star) = \mathbf{H}_o^S(\boldsymbol{\theta}^\star)$ provided for by the assumption A2.

$\square$

### A.6.1 OVER-PARAMETERIZED CASE

In our above discussion, we considered that $\boldsymbol{\Sigma} = \mathbf{X}^\top \mathbf{X}$ was invertible (and likewise in the corollary we considered $\mathbf{Z}_S(\boldsymbol{\theta}^\star)\mathbf{Z}_S(\boldsymbol{\theta}^\star)^\top$ was invertible). However, in the over-parameterized case this many not necessarily be the case. Nevertheless, there is one simple fix to this issue, as often carried out in the literature. We consider $\widehat{\boldsymbol{\Sigma}} = \boldsymbol{\Sigma} + \lambda \mathbf{I}$ in place of $\boldsymbol{\Sigma}$ for $\lambda > 0$, and in regards to influence functions, this would basically amount to having an $\ell_2$ regularization in the loss function. As a result, we can exactly repeat the same steps in the proof of the Theorem 7, except with $\widehat{\boldsymbol{\Sigma}}$ instead. Eventually, we recover the same formulas but now the expression of the resulting matrix $\mathbf{A}$ is slightly different as mentioned below:

$$\mathbf{A} = \mathbf{X}\left(\mathbf{X}^\top \mathbf{X} + \lambda \mathbf{I}\right)^{-1} \mathbf{X}^\top$$

But, we can further use the well-known push-through identity and obtain $\mathbf{A} = \left(\mathbf{X}\mathbf{X}^\top + \lambda \mathbf{I}\right)^{-1} \mathbf{X}\mathbf{X}^\top$, where we consider the shorthand $\mathbf{K} := \mathbf{X}\mathbf{X}^\top$ to indicate the kernel or the gram matrix. Rewriting this gives, $\mathbf{A} = (\mathbf{K} + \lambda \mathbf{I})^{-1} \mathbf{K}$ which is the familiar expression as seen in regularized kernel regression. For the case of Corollary 8, the same extension can be carried out, except that now the matrix $\mathbf{K}$ will be the empirical Neural Tangent Kernel (NTK) (Jacot et al., 2018) matrix *at the optimum* $\boldsymbol{\theta}^\star$.

## B  INFLUENCE FUNCTIONS: A PRIMER

In this primer on influence functions, we closely follow the textbook of Hampel et al. (1986). The key objective of influence function is to investigate the infinitesimal behaviour of functionals, such as $T(\mathcal{D}_n)$. In particular, this is defined when the change in underlying distribution can be expressed in the form of a Dirac distribution. Formally, this is defined as follows:

**Definition 10.** *The influence function* IF *of the estimator T, at some distribution F, evaluated at a point $z$ (where such a limit exists) is given by,*

$$\mathrm{IF}(z; T, F) = \lim_{\epsilon \to 0} \frac{T\left((1 - \epsilon)F + \epsilon \delta_z\right) - T(F)}{\epsilon}$$

Essentially, the influence function (IF) involves a directional derivative of $T$ at $F$ along the direction of $\delta_z$. Further, in order to interpret the above definition, substitute $F$ by $\mathcal{D}_{n-1}$ and put $\epsilon = \frac{1}{n}$. This implies that (IF) measures $n$ times the change of statistic $T$ due to an additional observation $z$. In other words, it describes the (standardized) effect of an infinitesimal contamination on the estimate $T$. E.g., in the case of parameter estimator, the change in estimated parameters due to presence of an additional datapoint. By now, the astute reader can already see its natural application for leave-one-out error, but we ask for some patience so as to discuss some other important aspects of influence functions.

More generally, one can view influence functions from the perspective of a Taylor series expansion (to be precise, the first-order von Mises expansion) of $T$ at $F$, evaluated on some distribution $G$ "close" to $F$,

$$T(G) = T(F) + \int \mathrm{IF}(z; T, F) d(G - F)(z) + \text{remainder terms}. \tag{47}$$

So, as evident from this, it is also possible to consider higher-order influence functions and use them in a combined manner, as considered in Debruyne et al. (2008). However, since the first-order term is often the dominating term — as well as to ensure tractability when we later consider neural networks — we will restrict our attention to only first-order influence functions hereafter.

### B.1  PROPERTIES OF INFLUENCE FUNCTIONS

**(i) Zero expectation.**  The expectation of influence function over the same distribution is zero, i.e., $\int \mathrm{IF}(z; T, F) dF(z) = 0$. This should actually be quite intuitive as we are basically averaging out all possible deviations of the estimator. But, the formal reasoning is that influence function is essentially akin to Gâteaux derivative at the distribution $F$,

$$\lim_{\epsilon \to 0} \frac{T\left((1 - \epsilon)F + \epsilon G\right) - T(F)}{\epsilon} = \int \mathrm{IF}(z) dG(z).$$

Now, replace $G$ by $F$ in the above equation and the stated property follows.

**(ii) Variance of IF provides asymptotic variance of corresponding estimator.**  When the observations $z_i$ are sampled i.i.d. according to $F$, then the empirical distribution $\mathcal{D}_n$ converges to F, for $n$ sufficiently large, by Glivenko-Cantelli theorem. So, replacing $G$ by $\mathcal{D}_n$ in eq. 47 and using the first property, we get:

$$T_n(\mathcal{D}_n) \approx T(F) + \int \mathrm{IF}(z; T, F) d\mathcal{D}_n(z) + \text{remainder terms}.$$

Integrating over the empirical distribution $\mathcal{D}_n$, we obtain:

$$\sqrt{n}(T_n(\mathcal{D}_n) - T(F)) \approx \frac{1}{\sqrt{n}} \sum_{i=1}^{n} \mathrm{IF}(z_i; T, F) + \text{remainder terms}.$$

The term on the right-hand side involving IF is asymptotically normal by (multi-variate) Central Limit Theorem. Further, the remainder terms can often be neglected for $n \to \infty$, and thereby the

estimator $T_n$ is also asymptotically normal. Thus, $\sqrt{n}(T_n(\mathcal{D}_n) - T(F)) \xrightarrow{d} \mathcal{N}_p(0, V(T, F))$, with the asymptotic variance (more accurately, covariance matrix) $V(T, F)$ as follows:

$$V(T, F) \;=\; \int \mathrm{IF}(\boldsymbol{z}; T, F)\, \mathrm{IF}(\boldsymbol{z}; T, F)^\top \, dF(\boldsymbol{z})\,.$$

For the one-dimensional case, i.e., $T(F) \in \mathbb{R}$, we simply get $V(T, F) \;=\; \int \mathrm{IF}(\boldsymbol{z}; T, F)^2 \, dF(\boldsymbol{z})$.

**(iii) Chain rule for influence functions.** Suppose our estimator $\widehat{\boldsymbol{\theta}}(F)$ depends on some other estimators, i.e., $\widehat{\boldsymbol{\theta}}(F) := T\left(\widehat{\boldsymbol{\theta}}_1(F), \cdots, \widehat{\boldsymbol{\theta}}_k(F)\right)$, then we can expression the influence function for $\widehat{\boldsymbol{\theta}}(F)$ as,

$$\mathrm{IF}(\boldsymbol{z}; \widehat{\boldsymbol{\theta}}, F) \;=\; \sum_{j=1}^{k} \frac{\partial T}{\partial \widehat{\boldsymbol{\theta}}_j} \, \mathrm{IF}(\boldsymbol{z}; \widehat{\boldsymbol{\theta}}_j, F)\,. \tag{48}$$

**Remark.** We refer the mathematically-oriented reader to Huber (2004) for details on the regularity conditions needed to ensure the existence of IF and the like.

# C  EMPIRICAL RESULTS AND DETAILS

## C.1  EMPIRICAL DETAILS

We train all the networks via SGD with learning rate $0.5$ and learning rate decay by a factor of $0.75$ after each quarter of the target number of epochs.

For the experiments based on MNIST1D (Greydanus, 2020), the input dimension size is $d = 40$ while number of classes is $K = 10$.

**Double-descent model sizes**  (a) For the 3-layer double descent plot, we consider hidden widths $m_1, m_2 \in \{10, 20, 30, 40, 50\}$ and choose all those pairs as experiments where $|m_1 - m_2| \le 20$.

(b) For the 2-layer plot in MSE, we take the hidden layer widths from $[1, 151]$ at an interval of $10$.

(c) For the 2-layer plot corresponding to CE, we take hidden layer sizes from $[2, 32]$ at intervals of $2$ and then for reducing the computation load when the network sizes increase, we take coarser intervals with a gap $5$ and for even bigger sizes, at a gap of later $10$. But this is purely to reduce computational load that comes with Hessian computation.

In all the double descent curves, we ensure that near the interpolation threshold all models are at interpolation, in the sense that the training accuracy is $100\%$.

## C.2  VERIFICATION OF ASSUMPTION A3

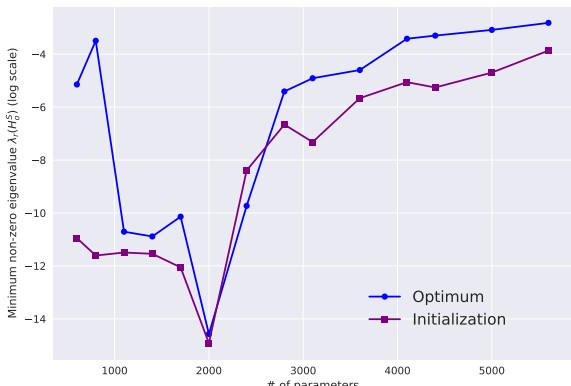

**(a)** 2-hidden layer case, MNIST1D.

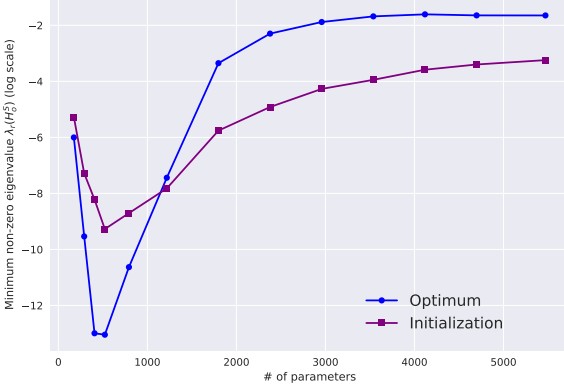

**(b)** 1-hidden layer, CIFAR10.

**Figure 4:** Additional results on verifying the assumption A3

## C.3 Cross Entropy double descent

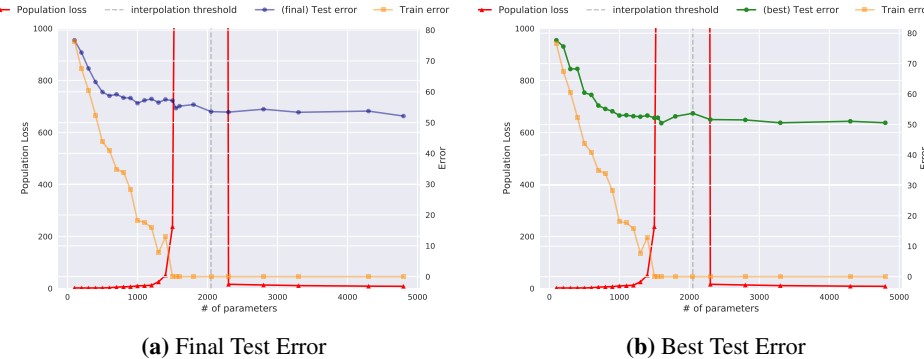

**(a)** Final Test Error

**(b)** Best Test Error

**Figure 5:** Both test error at the final epoch as well as the 'best' test error in terms of the one selected based on a small validation set.

## C.4 MSE double descent

For the sake of visualization in the 3-layer case, we smooth the quantities involved by considering a moving average of them over three successive model sizes.

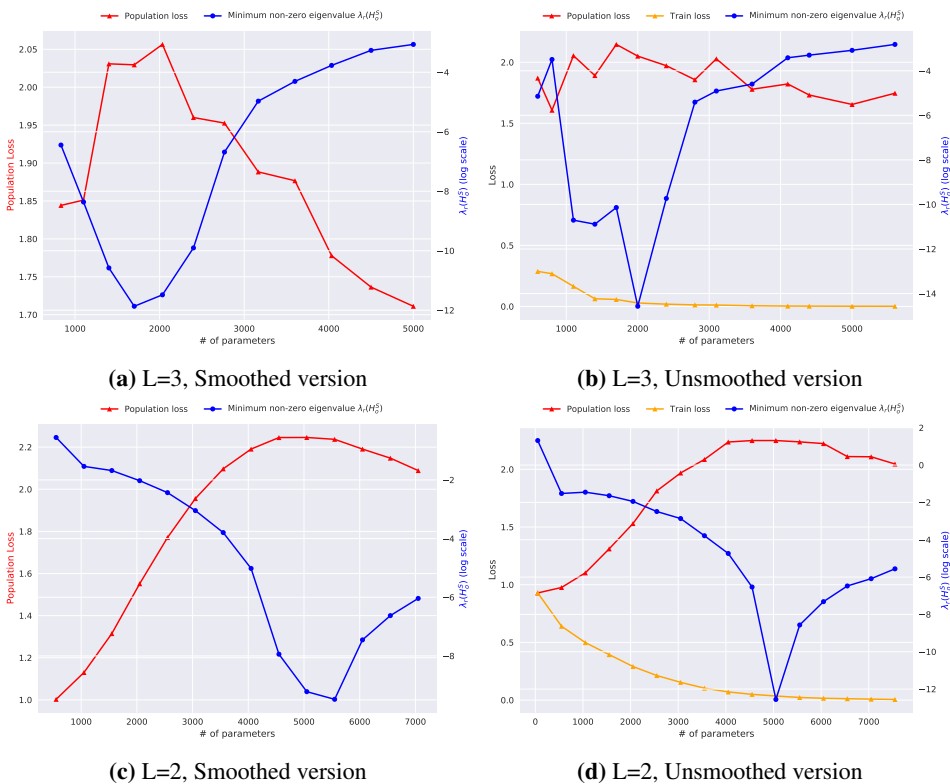

**(a)** L=3, Smoothed version

**(b)** L=3, Unsmoothed version

**(c)** L=2, Smoothed version

**(d)** L=2, Unsmoothed version

**Figure 6:** We see that the same trend holds in both the cases — In fact, the unsmoothed version perhaps even more starkly shows the minimum non zero eigenvalue vanishes at the maximum population loss.

Also, in the unsmoothed version of the above plots, we also show the corresponding train loss. It must be emphasized that near interpolation the training accuracy is $100\%$, even though there might be some non-zero training loss.

## C.5 RESULTS ON ADDITIONAL DATASETS

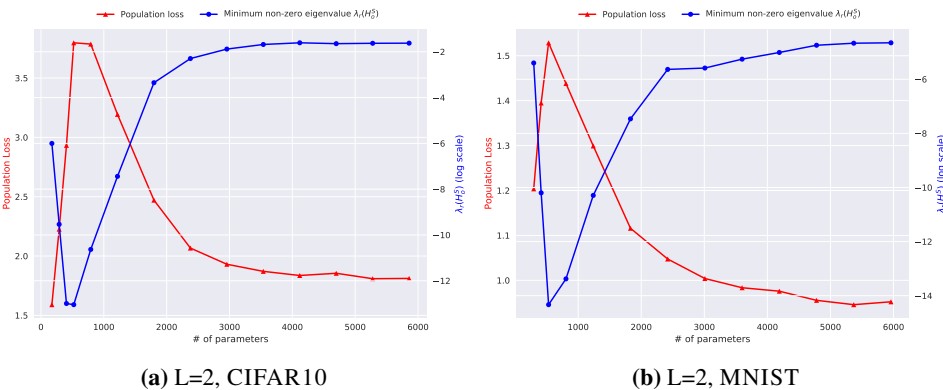

**(a)** L=2, CIFAR10       **(b)** L=2, MNIST

**Figure 7:** Population risk behaviour alongside the minimum non-zero eigenvalue of the Hessian at the optimum for CIFAR10 and MNIST with MSE loss.

**Empirical details.** The optimization details are the same as mentioned before in the setting of MNIST1D, except that we train longer for $20K$ epochs. We downsample the input dimension for CIFAR10 and MNIST to about the same size so as to ensure consistency, and in particular, $d = 48$ for CIFAR10 (by downsampling $32 \times 32 \times 3$ images to $4 \times 4 \times 3$ and then flattening) while $d = 49$ for MNIST (by downsampling $28 \times 28$ images to $7 \times 7$ and then flattening). We find that on harder datasets like these, it takes even longer to drive the networks to interpolation and thus consider $n = 40$ samples. We consider the sizes of the hidden layer from the set $\{1, 3, 5, 7, 9, 11, 21, 31, 41, 51, 61, 71, 81, 91, 101, 111\}$. In other words, we sample at a finer granularity near the interpolation threshold ($\approx 400$), while increase the gaps later on. Also, we find that networks with hidden layer width $m = 1$ fails to train and gives NANs, so we exclude its result. Overall, we find that the population risk and the minimum non-zero eigenvalue of the Hessian at the optimum show a very similar trend like in the case of MNIST1D. *As a result, this implies that our empirical findings also generalize to other datasets as well.*

## C.6 2-HIDDEN LAYER RESULTS FOR LARGE DIFFERENCE IN SUCCESSIVE LAYER SIZES

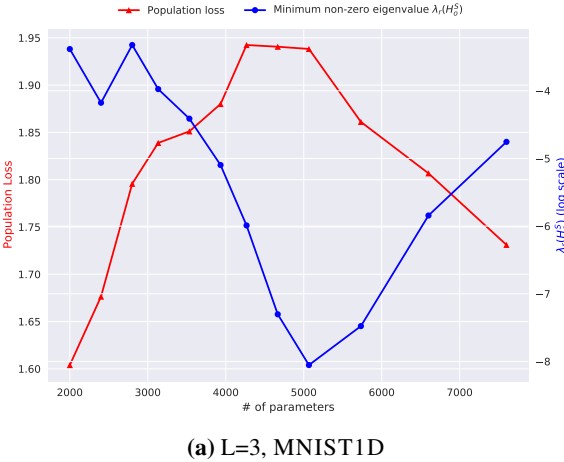

**(a)** L=3, MNIST1D

**Figure 8:** Population risk behaviour alongside the minimum non-zero eigenvalue of the Hessian at the optimum for a 2-hidden layer network on MNIST1D, *but where the successive hidden layer sizes $m_1$, $m_2$ can have much bigger differences — i.e., $|m_1 - m_2| \leq 120$ with MSE loss.*

Previously, in our double descent experiments for 2-hidden layers, we considered that hidden widths $m_1$, $m_2 \in \{10, 20, 30, 40, 50\}$ and chose all those pairs where $|m_1 - m_2| \leq 20$. Now, a question might arise what happens in the case when the successive hidden layer sizes are very 'imbalanced' (or non-uniform), i.e., they have a big difference between their sizes. The Figure 8 shows the setting of double descent with 2 hidden layers with sizes $m_1$ and $m_2$, which are chosen as per $m_1 \in \{20, 40\}$ and $m_2 \in \{20, 40, 60, 80, 100, 120, 140\}$. Rest of the empirical details regarding training and dataset are identical to the MNIST1D setting considered earlier. We find the population risk and minimum non-zero eigenvalue behaviour which is very similar to Figure 1b and in accordance with our theoretical predictions. This also confirms that our theoretical analysis generalizes to diverse architecture patterns, not just the 'balanced' setting in which double descent was has been considered before, which leads to the following remark.

**Closing remark.** Before we finish this discussion, let us emphasize that almost all prior works on double descent Nakkiran et al. (2019); Nakkiran (2019) consider the case of 'balanced' hidden layer sizes. In other words, these works consider a fixed architectural pattern, say $\{m\} \rightarrow \{2m\} \rightarrow \{m\}$ and then increase the common width multiplier $m$. *In this respect, we are one of the first works that not just demonstrates the occurrence of double descent in the imbalanced hidden-layer settings — but also explains the behaviour via the trend of the minimum non-zero eigenvalue of the Hessian at the optimum.*

## C.7 NATURE OF CONSTANTS FOR THE EIGENVALUE ASSUMPTION

In the assumption A3, we assume the existence of constants $0 < A_{\boldsymbol{\theta}}^0 < B_{\boldsymbol{\theta}}^0 < \infty$ such that the following holds:

$$A \lambda_{\min}(\mathbf{C}_{\boldsymbol{f}}^S(\boldsymbol{\theta}^0)) \leq \lambda_{\min}(\mathbf{C}_{\boldsymbol{f}}^S(\boldsymbol{\theta}^\star)) \leq B \lambda_{\min}(\mathbf{C}_{\boldsymbol{f}}^S(\boldsymbol{\theta}^0))$$

So in this section, how these constants actually behave in practice and if they are $\mathcal{O}(1)$ across varying network sizes? Or, equivalently, how the ratio $\frac{\lambda_{\min}(\mathbf{C}_{\boldsymbol{f}}^S(\boldsymbol{\theta}^\star))}{\lambda_{\min}(\mathbf{C}_{\boldsymbol{f}}^S(\boldsymbol{\theta}^0))}$ behaves and if it is $\mathcal{O}(1)$?

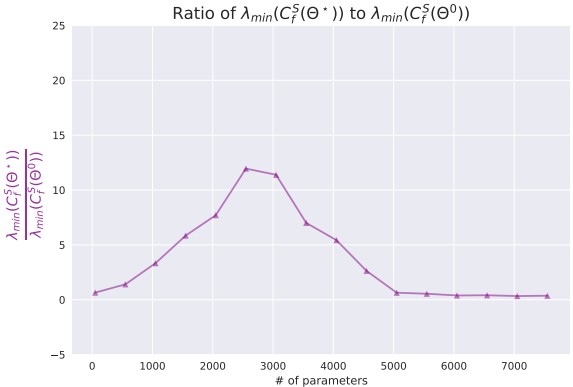

**Figure 9:** The ratio $\frac{\lambda_{\min}(\mathbf{C}_{\boldsymbol{f}}^S(\boldsymbol{\theta}^\star))}{\lambda_{\min}(\mathbf{C}_{\boldsymbol{f}}^S(\boldsymbol{\theta}^0))}$

We consider the setting of one-hidden layer neural networks trained with MSE loss for $5K$ epochs as discussed in the main text (for other training details, see Appendix C.1). To remind, we consider networks of hidden-layer sizes sampled uniformly from $[1, 151]$ at an interval of 10. The Figure 9 plots this desired ratio. We find that indeed this ratio is $\mathcal{O}(1)$. More precisely, the Table 1 details the summary statistics of this ratio.

Overall, the trend in Figure 9 and the precise summary statistics in the above Table 1 even implies the existence of universal constants for $A_{\boldsymbol{\theta}}^0$ and $B_{\boldsymbol{\theta}}^0$ (e.g., one setting would be to 0.3 and 12 respectively). This ratio naturally depends on the problem, so it is likely that it will change slightly (e.g., on CIFAR10, the mean changes to 7.812) but we are able to confirm the existence of universal constants

| Minimum | Median | Mean | Maximum |
|---------|--------|------|---------|
| 0.336 | 2.019 | 3.754 | 11.955 |

**Table 1:** Summary statistics of the ratio $\frac{\lambda_{\min}(\mathbf{C}_f^S(\boldsymbol{\theta}^\star))}{\lambda_{\min}(\mathbf{C}_f^S(\boldsymbol{\theta}^0))}$.

across all our experiments. Lastly, there is also a theoretical basis to it, as remarked in the main text, since the map $\mathbf{A} \mapsto \lambda_i(\mathbf{A})$ is Lipschitz-continuous on the space of Hermitian matrices, which follows from Weyl's inequality (Tao, 2012, p. 56).

**Final remarks.** *Importantly, we would thus like to reiterate that we never impose constraints on the minimum eigenvalue of the covariance of the function Jacbians. But, rather inspired by the above empirical observation, we consider the existence of such constants.*

### C.8 CONSTITUENT OF HESSIAN AT THE OPTIMUM

For our analysis, we made the assumption A2 based on the prior works of (Sagun et al., 2017; Singh et al., 2021). We also find in our own experiments that the very same observation made in these papers holds — namely, that the Hessian $\mathbf{H}_{\mathcal{L}}$ at the optimum is only composed of the outer-product Hessian $\mathbf{H}_o$, while the functional Hessian $\mathbf{H}_f = \mathbf{0}$.

To show this, we plot in Figure 10 the nuclear norm, spectral norms, as well as the minimum non-zero eigenvalue of the the overall loss Hessian $\mathbf{H}_{\mathcal{L}}$ and the outer-product Hessian $\mathbf{H}_o$ for one-hidden layer networks trained for 20K epochs and with rest of the training details identical to all other experiments.

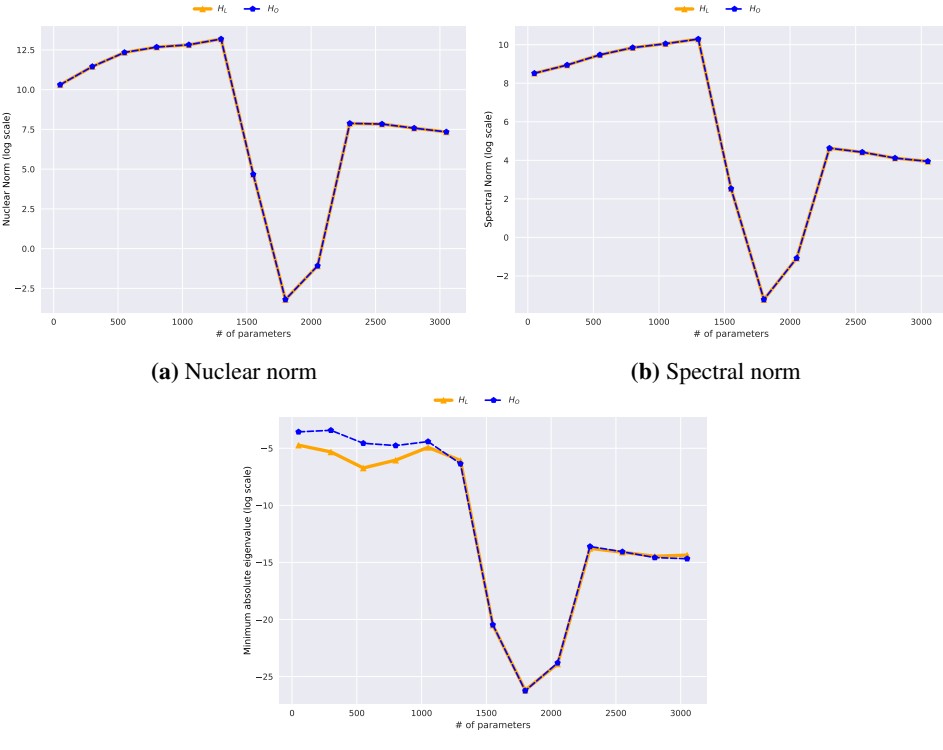

**(a)** Nuclear norm          **(b)** Spectral norm

**(c)** Minimum non-zero eigenvalue

**Figure 10:** Comparison of the spectra of $\mathbf{H}_{\mathcal{L}}$ and $\mathbf{H}_o$ at the optimum $\boldsymbol{\theta}^{\star}$ (across the set of networks trained for double descent). We find that across all the above-plotted measures the curves for $\mathbf{H}_{\mathcal{L}}$ and $\mathbf{H}_o$ coincide. Thus showing that the functional Hessian $\mathbf{H}_f$ goes to zero at the optimum and thereby establishing the merit of the assumption A2.

### C.9 LOWEST FEW EIGENVALUES ARE THE DOMINATING FACTOR BEHIND DIVERGENCE NEAR INTERPOLATION THRESHOLD

We consider the settings of CIFAR10, MNIST, and MNIST1D, all of which correspond to the double descent shown in Figures 7a, 7b, and 1 respectively. To demonstrate how many of the lowest eigenvalues capture the double descent trend, for each model in all of the above settings, we plot the % of the trace (of the Hessian inverse) captured by the lowest eigenvalues of the Hessian, since the lower bounds exhibit this dependence. In order to ensure consistent comparisons across varying model sizes, we consider the lowest eigenvalues in $0.5\%, 1\%, 2\%, 5\%, 10\%$ of the total number of eigenvalues of that model. The results can be found in the following plots:

**Observations.** We see that across all these cases just $0.5\%$ of the lowest eigenvalues are enough to capture the double descent behaviour — capturing a minimum of $60\%$ of the trace near interpolation across all these settings.

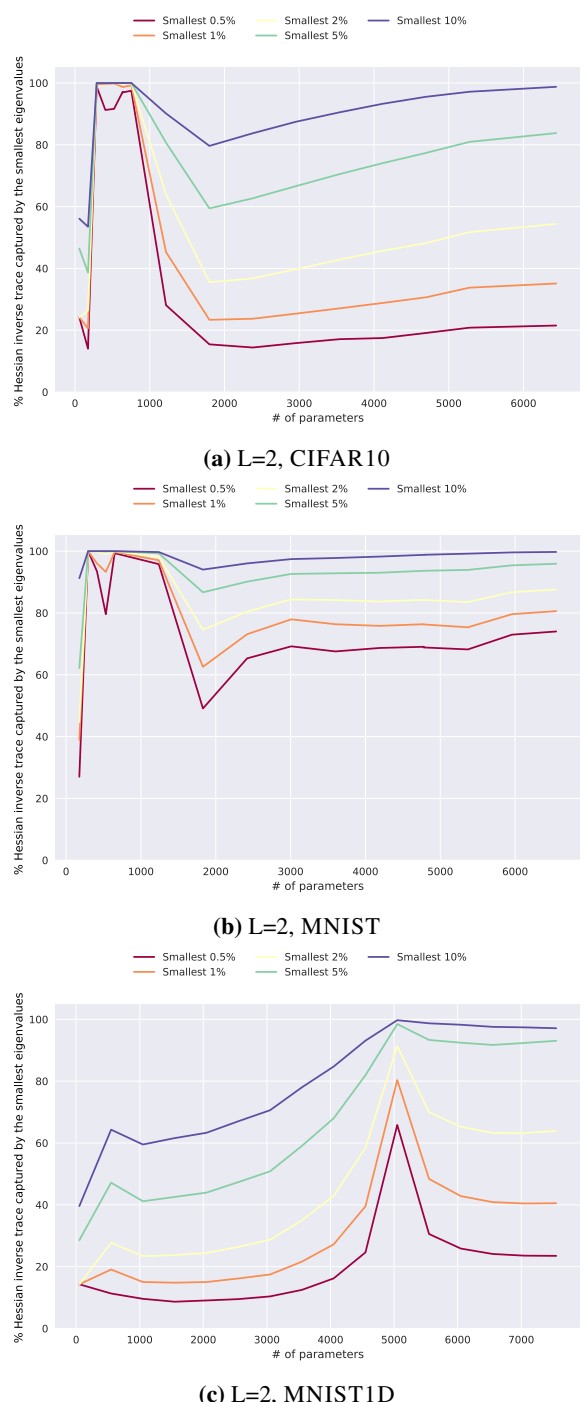

**(a)** L=2, CIFAR10

**(b)** L=2, MNIST

**(c)** L=2, MNIST1D

**Figure 11:** The % of the Hessian inverse trace captured by varying proportions of smallest eigenvalues of the Hessian across multiple datasets for MSE loss.

Further $> 98\%$ of the trace is captured as soon as we have $1\%$ of the lowest eigenvalues for CIFAR10, $0.5\%$ for MNIST, and $5\%$ for MNIST1D. *This clearly shows that the double descent behaviour is indeed captured by a very small handful of the lowest eigenvalues.* (As a matter of fact, near the interpolation threshold, even just using the minimum non-zero eigenvalue alone captures $85.56\%$ of the trace for CIFAR10 and $97.80\%$ for MNIST.)

### C.10 LOWER BOUND COMPUTATION

At an initial glance, it might seem that the computed lower bounds could be very small in magnitude to be of use — as the bound is scaled by $\sigma_{\min}^2$.

However, while practically computing one can consider a tighter lower bound, by using a particular tolerance $\tau$.

Let us illustrate by considering evaluating this bound on a test set $S'$ (just like we actually do). This is because in the lower bound we have the expression,

$$LHS = \frac{1}{|S'|} \sum_{(\boldsymbol{x},\boldsymbol{y}) \in S'} \left[ \text{Tr} \left( \mathbf{A}_{(\boldsymbol{x},\boldsymbol{y})} \right) \cdot \| \nabla_{\boldsymbol{f}} \ell \left( \boldsymbol{f}_{\boldsymbol{\theta}^\star}(\boldsymbol{x}), \boldsymbol{y} \right) \|^2 \right]$$

Now, since $\text{Tr}\left(\mathbf{A}_{(\boldsymbol{x},\boldsymbol{y})}\right)$ is non-negative, if we are given a tolerance $\tau$, we filter out those samples with $\|\nabla_{\boldsymbol{f}}\ell\left(\boldsymbol{f}_{\boldsymbol{\theta}^\star}(\boldsymbol{x}),\boldsymbol{y}\right)\|^2 < \tau$

$$LHS \geq \frac{1}{|S'|} \sum_{(\boldsymbol{x},\boldsymbol{y}) \in S'} \left[ \text{Tr} \left( \mathbf{A}_{(\boldsymbol{x},\boldsymbol{y})} \right) \cdot \tau \cdot \mathbb{1}\{ \| \nabla_{\boldsymbol{f}} \ell \left( \boldsymbol{f}_{\boldsymbol{\theta}^\star}(\boldsymbol{x}), \boldsymbol{y} \right) \|^2 \geq \tau \} \right]$$

where, $\mathbb{1}$ is the indicator function. This helps us compute lower bounds which can be evaluated in practice.

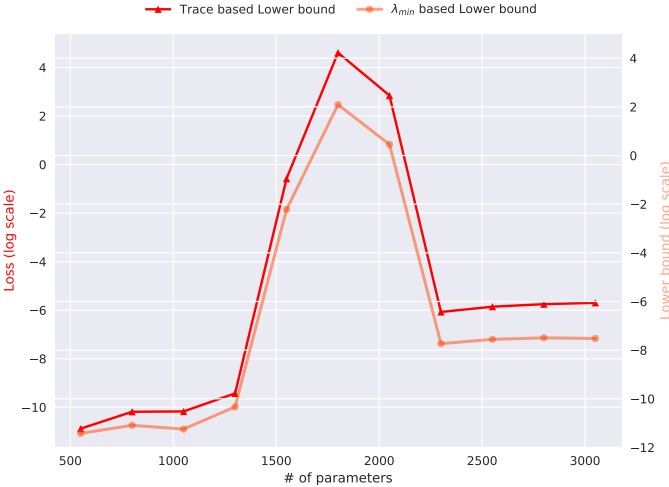

**Figure 12:** Comparison of lower bounds with trace vs minimum eigenvalue

## C.11 EMPIRICAL OBSERVATIONS FROM LINEAR REGRESSION

### C.11.1 HESSIAN STATISTICS ON TRAINING SET

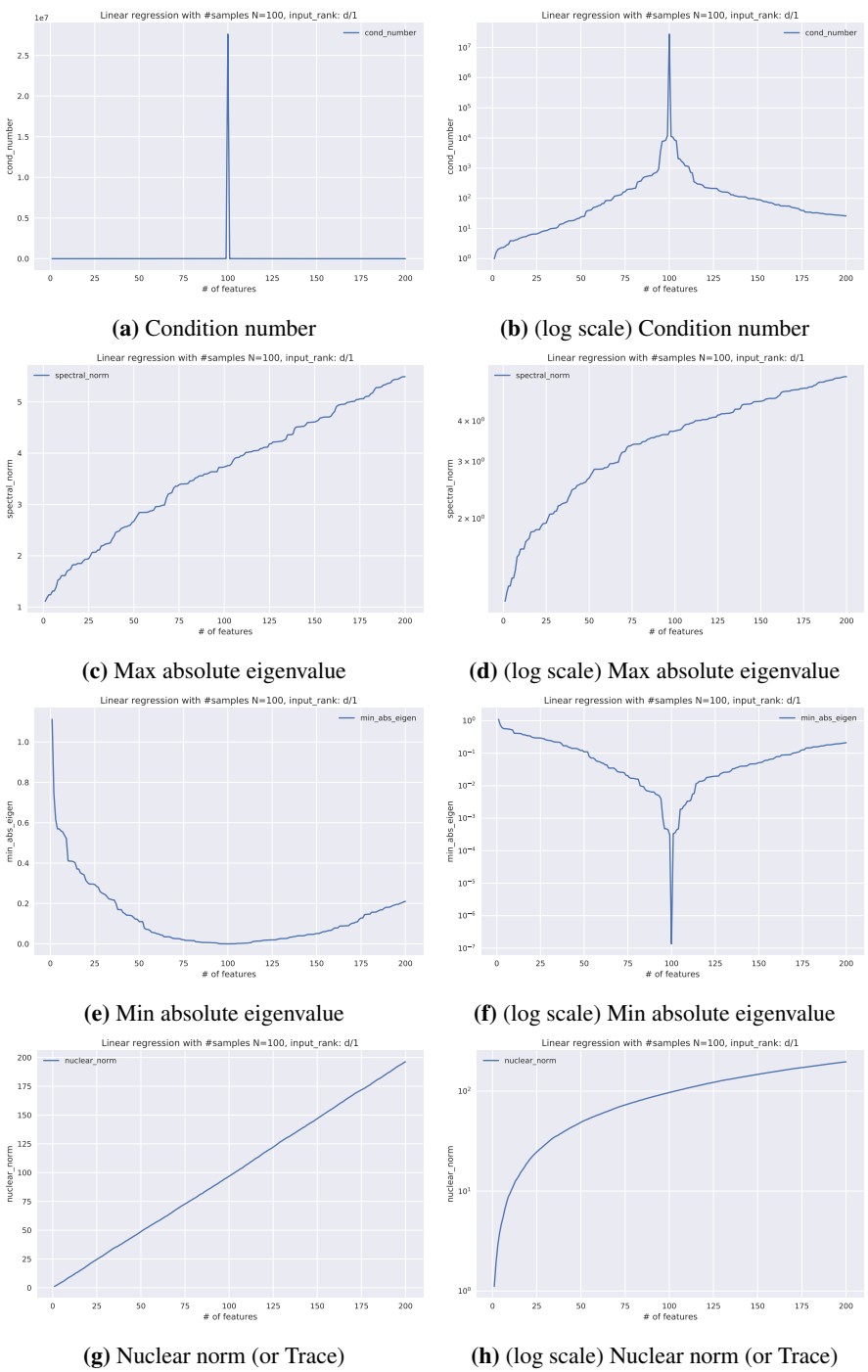

**(a)** Condition number

**(b)** (log scale) Condition number

**(c)** Max absolute eigenvalue

**(d)** (log scale) Max absolute eigenvalue

**(e)** Min absolute eigenvalue

**(f)** (log scale) Min absolute eigenvalue

**(g)** Nuclear norm (or Trace)

**(h)** (log scale) Nuclear norm (or Trace)

**Figure 13:** Hessian statistics computed over the training set. We observe that condition number diverges at the double descent peak, owing to minimum absolute eigenvalue becoming close to zero at this threshold.

### C.11.2 HESSIAN STATISTICS ON TEST SET

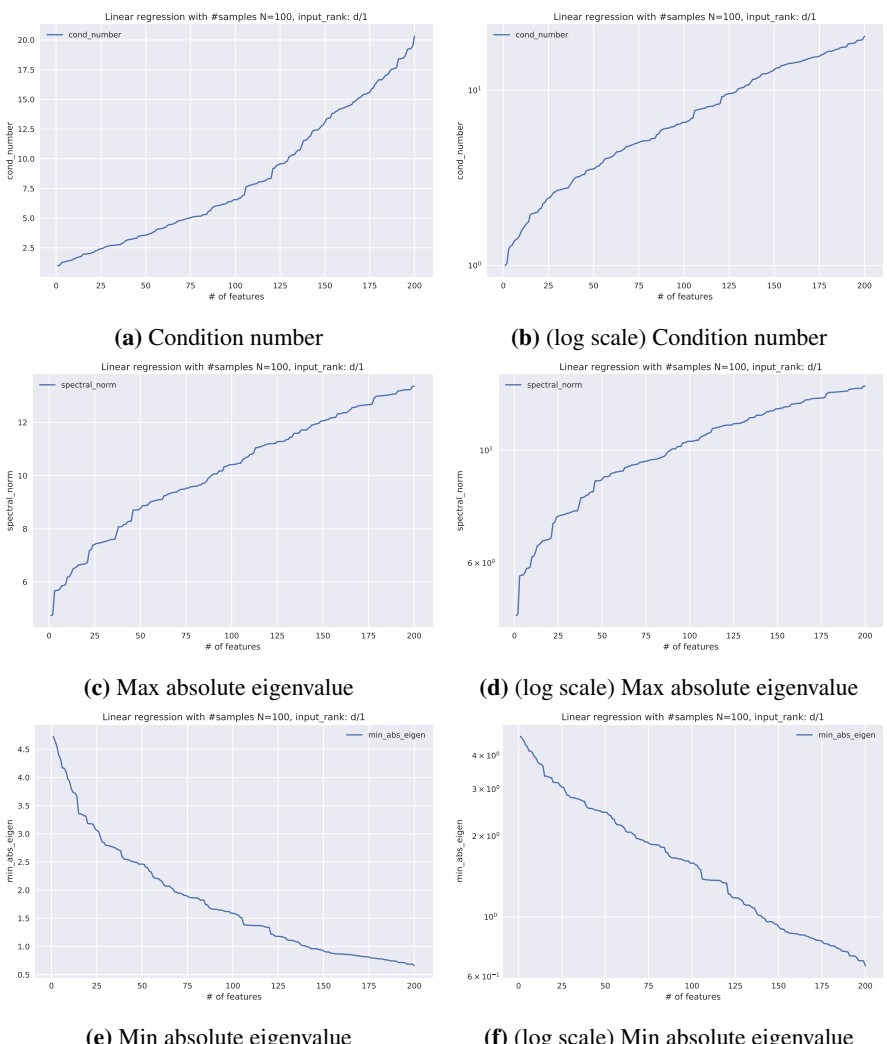

**(a)** Condition number

**(b)** (log scale) Condition number

**(c)** Max absolute eigenvalue

**(d)** (log scale) Max absolute eigenvalue

**(e)** Min absolute eigenvalue

**(f)** (log scale) Min absolute eigenvalue

**Figure 14:** Hessian statistics computed over the test set (size 500).

### C.11.3 DRAWING DD PLOTS AT ARBITRARY THRESHOLDS

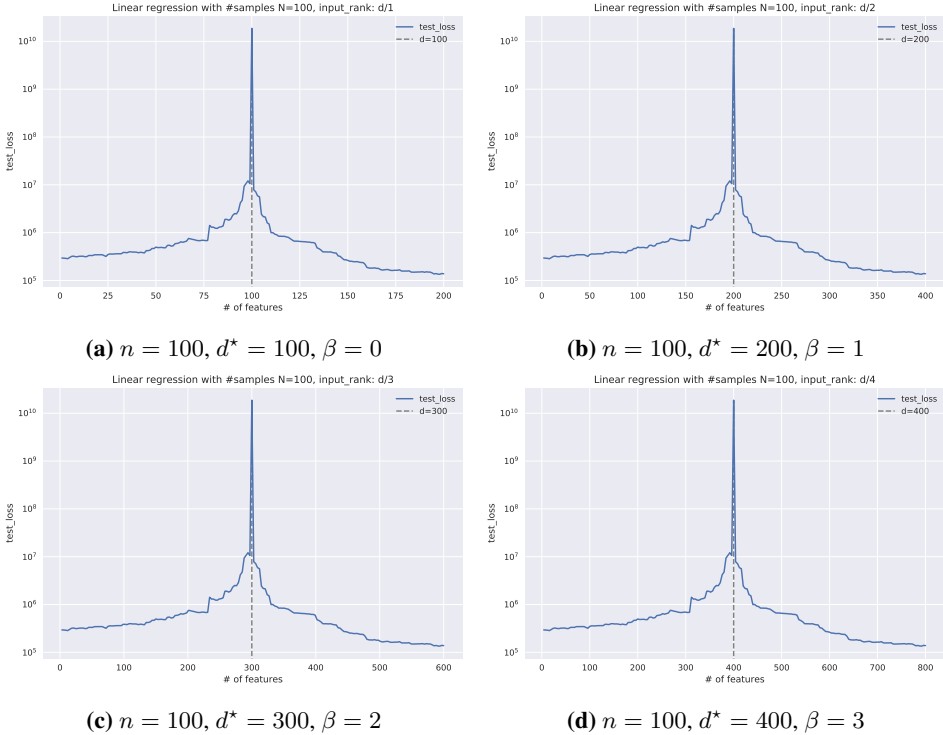

**(a)** $n = 100$, $d^\star = 100$, $\beta = 0$

**(b)** $n = 100$, $d^\star = 200$, $\beta = 1$

**(c)** $n = 100$, $d^\star = 300$, $\beta = 2$

**(d)** $n = 100$, $d^\star = 400$, $\beta = 3$

**Figure 15:** Changing the position of the interpolation threshold by adjusting the rank of the underlying input-covariance matrix.

The input design matrix is constructed in such a way so that given some $d'$ many input features, the number of base features is $d$. But we additionally have $\beta\, d$ many redundant features (composed of linear combinations of the base features). Also, as the number of features is increased in the above test loss curve, these redundant features follow in succession to the base features. Therefore, at any point in these graphs along the x-axis for some number of features $d'$, the effective number of features are $\dfrac{d'}{(\beta + 1)}$. Hence, the peak occurs when the effective number of features equals the (effective) number of samples, i.e., at $\dfrac{d^\star}{\beta + 1} = n$, which is nothing but $d^\star = n(\beta + 1)$. Thereby, looking from the perspective of rank (the Hessian here is simply the covariance matrix), the position of the interpolation threshold can be reconciled.

