# OpenReview forum: "Phenomenology of Double Descent in Finite-Width Neural Networks"
_ICLR.cc/2022/Conference — ICLR 2022 Poster_

### Official Review · Reviewer_E8Hk · 2021-10-26

**Correctness:** 2
**Technical Novelty And Significance:** 1
**Empirical Novelty And Significance:** 2
**Recommendation:** 3
**Confidence:** 3

**Main Review:**

The topic and the goal are both very important. However, the theorems are crucially limited and incremental. Theorem 3 simply rewrites the population risk by putting the difficult term into the expectation or population covariance of the gradient. The right-hand of (1) cannot be evaluated because of the population covariance. In the most cases, if we can evaluate it, then we can evaluate the population risk too. Moreover, the second term includes the trance of the matrix of size of the number of parameters. Thus, this can grow linearly as we increase the number of parameters in the worst case, which is not good for over-parametrized regime that this paper focuses on; otherwise, the first term is no longer negligible.

The proof of Theorem 3 is very unclear regarding the O(n^2) term. It reads “when |S| = n is large enough to ignore O(n^2) terms, the (infinitesimal) definition of influence function is equivalent to using the finite-difference form”. However, the O(n^2) term can also grow in practice as we increase n. Thus, the authors must include the exact expression of the O(n^2) term and a more precise argument on why we can ignore this term.

In Theorem 4, Assumption 5 is problematic in many ways. First of all, the main goal of paper, as stated, is to avoid the neural tangent kernel regime, which is known to be not practical, and to analyze the practical finite-width regime of neural networks. However, Assumption 5 almost results in the neural tangent kernel regime, because the covariance contains neural tangent kernel by the chain rule, and the Assumption 5 restricts the practical change of the neural tangent kernel during the training, which is exactly the main issue of the neural tangent kernel regime. In the neural tangent kernel regime, the neural tangent kernel is not allowed to change a lot, which means that neural networks in the neural tangent kernel regime is equivalent to linear models with a random feature (or the initial kernel). The proof of this paper is not addressing this main limitation.

Minor comments: why equation (10) in Theorem 10 does not contain O(n^2) term? There is a typo on page 3 with undesired space before `let us consider who the parameter estimator’.

Update after author response: the authors' responses do not address my concerns. Assumption 5 (and the minimum eigenvalue in the revised version) is problematic. The modes studied in this paper are essentially still in the lazy/kernel regime. Based on these, I keep my original score.

**Summary Of The Paper:**

This paper derives an expression of the population risk with influence function, and then use that to derive a lower bound on the population risk, which provides an explanation for double descent phenomena in the finite-width regime at the optimum point. The paper provides experiments to support the theoretical arguments. Moreover, the authors investigate how the choice of loss functions affect the double descent phenomena.

**Summary Of The Review:**

The topic and the goal are both very important. However, the theorems are crucially limited and incremental as explained above. Moreover, the main claim to analyze the finite-width neural networks outside of the neural tangent kernel regime is not well supported because of the Assumption 5.

---

> ### Author Response · Authors · 2021-11-22
> **Response to reviewer E8Hk (2/2)**
>
> ### Theorem 4 (continued):
> \
> &nbsp;
> > “the Assumption 5 restricts the practical change of the neural tangent kernel during the training,”
>
> - **No, this is not true.** All we ask in the Assumption A5 (or what is now called A3) is that there exists finite constants, *but we never impose any limits on their scale* and even more importantly, **we are by no means constraining the training of neural networks.**
>
>
> - Rather, this assumption A5 is actually inspired from an empirical observation in the case of finite-width neural networks **(hidden layer width in the range $[1, 151]$)**, trained via usual stochastic gradient descent, as detailed in **Appendix C.8.**
>
>
> - More empirical verifications of this assumption can be found in Figure 4, where we see that the minimum non-zero eigenvalue at optimization and the one at initialization have very similar trends.
>
>
> - From a theoretical perspective, the existence of these constants is provided because the map $\mathbf{A}\mapsto\lambda_i(\mathbf{A})$ is *Lipschitz-continuous* on the space of Hermitian matrices, which follows from Weyl's inequality, see reference to Tao 2012 provided in the paper.
>
>
> *Thus, to reiterate, the assumption this is solely for the purpose of getting the lower bound in Theorem 5. Hence, the statement “almost results in the neural tangent kernel regime” is **false**.*
> \
> &nbsp;
>
> ----
>
> ### “The proof of this paper is not addressing this main limitation [w.r.t RF/NTK-based  analyses]”
> \
> &nbsp;
>
> **This remark is far remote from what is actually the case.** To illustrate this, we would like to remind the usual assumptions made in the analyses of double descent for the random feature RF  or Neural Tangent Kernel NTK regime:
> \
> &nbsp;
>
> 1. First, RF/NTK based analyses [2,3, 4, 5, 6] consider the *minimum norm interpolation solution* (amounts to simply using the pseudoinverse of the data matrix). In contrast, we consider solutions obtained by training finite-width neural networks using any of the standard optimization methods, like (stochastic) gradient descent.
>
>
> 2. Next, RF/NTK based analyses [2, 3, 4, 5], typically *consider 1-hidden layer networks* and that too with *training only the output layer*. Our analysis does not restrict the number of hidden layers or whether some layers should not be trained.
>
>
> 3. Further, RF/NTK based analyses [2, 3, 4, 5, 6] by definition consider the *infinite-width regime*. Our approach works for any finite-width neural networks, as also evident from the experiments.
>
>
> 4. Also, RF/NTK based analyses [4, 6] assume *particular scaling of the parameterization* (NTK style, i.e., lazy regime). In our case, we do not require any particular scaling and our results utilize the default PyTorch initialization.
>
>
>
> 5. Lastly and very importantly, RF/NTK based analyses [2,3, 4, 5, 6] assume a *teacher model*, i.e., a true underlying parametric form of the solution which generates the output. We consider no such restriction.
>
> \
> &nbsp;
> The above list just enumerates the most significant assumptions that RF/NTK based analyses impose, while we consider a general setting and require none of these assumptions — *something surely the other reviewers will concur on*. In fact, there are many more involved technical assumptions that these works employ, which we recommend you to check.
>
> Lastly, we would also encourage you to peruse the **Appendix A.4**, that provides concrete examples where all our assumptions hold simultaneously, and which also implies that results in RF/NTK regime forms a (basic) special case of our analysis.
> \
> &nbsp;
> ---
> \
> &nbsp;
> *To conclude, we hope that these detailed responses to your comments resolve your concerns. We ourselves will also strive to better convey these points in the text, with even more emphasis, so as to avoid any confusion that we may have caused before. Thus, we sincerely hope that you will consider reevaluating and rejudging the merits of the paper, afresh.*
>
> Please do not hesitate to ask us any further questions that you might have.
>
> ----
> \
> &nbsp;
>
> [1] Singh et. al., 2021, Analytic Insights into Structure and Rank of Neural Network Hessian Maps https://openreview.net/pdf?id=otDgw7LM7Nn, NeurIPS 2021
>
> [2] Hastie et. al., 2019, Surprises in High-Dimensional Ridgeless Least Squares Interpolation, https://arxiv.org/abs/1903.08560
>
> [3] Belkin et. al., 2020, Two Models of Double Descent for Weak Features https://epubs.siam.org/doi/pdf/10.1137/20M1336072
>
> [4] Mei and Montanari, 2019, The generalization error of random features regression: Precise asymptotics and double descent curve, https://arxiv.org/pdf/1908.05355.pdf
>
> [5] Bartlett et. al., 2020, Benign Overfitting in Linear Regression, https://arxiv.org/pdf/1906.11300.pdf
>
> [6] Harzli et. al., 2021 Double-descent curves in neural networks: a new perspective using Gaussian processes, https://arxiv.org/pdf/2102.07238.pdf

---

> ### Author Response · Authors · 2021-11-22
> **Response to reviewer E8Hk (1/2)**
>
> Thank you for reviewing our work and giving your critical feedback, which helped us to rethink our presentation. We are happy to note that you see the topic and goal as important.
>
> ----
> \
> &nbsp;
> Let us start by saying that we strongly believe the reviewer’s judgement that “the theorems are crucially limited and incremental” — *with all due respect — stems from judging the merits of our work from a perspective that is different from where we believe it should actually be judged. We accept that our writing was perhaps at fault, in that we could have communicated this more explicitly, and avoided misinterpretations. We hope we can engage in a discussion with you to rectify this.*
>
> Thus, we sincerely ask that the reviewer give us an opportunity by starting afresh as we together deconstruct the comments, point by point:
> \
> &nbsp;
> -----
>
> ### Theorem 3:
> \
> &nbsp;
> > “simply rewrites the population risk by putting the difficult term into the expectation or population covariance of the gradient. “
>
> - Firstly, there is a nuance to the expression of the population risk that we derive. We cannot simply consider the already existing TIC like expressions, since there is no dependence on the training set in the complexity factor therein. Plus, branching off TIC by merely approximating the Hessian and gradient covariance on the training set, does not work either (please see the discussion in Appendix A.2.2).
>
>
> - But, in any case, as **we explicitly mention** in our paper, the main results of our work are Theorem 4 and Theorem 5.
> \
> &nbsp;
>
>
> > “ In the most cases, if we can evaluate it, then we can evaluate the population risk too”,
>
> - Yes, but we would like to clarify that throughout our paper we never claim to provide a lower bound that can also be evaluated practically. So, we do not deny your valid observation.
>
>
> - Rather, the **primary objective** of our work, as stated from the very beginning, is to provide an **amenable analysis for double descent** in the case of finite-width neural networks. While our approach can indeed be an inspiration for “practically-minded” lower bounds, for us this is secondary to the goal of our work.
>
>
> - We have added a comment under Theorem 4 that elaborates on this aspect. But, we are happy to further refine or edit it based on your feedback.
> \
> &nbsp;
>
>
> > “[trace] can grow linearly as we increase the number of parameters in the worst case, which is not good for over-parametrized regime that this paper focuses on”
>
> - Our focus here is to specifically analyze what causes the population loss to peak near the interpolation threshold $p\approx n$. This can be analyzed perfectly well with a term which has a numerator of the order $\mathcal{p}$ while a denominator of the order $\mathcal{n}$.
>
>
> - Besides, to be precise, the trace will grow proportional to the rank. Recent work [1] has shown that for deep linear networks the rank is provably much smaller than the number of parameters and for non-linear networks a similar observation holds in practice (in terms of effective rank).
> \
> &nbsp;
>
>
> > “The proof of Theorem 3 is very unclear regarding the O(n^2) term.”
>
> - True, we could have been more clear on this aspect. We have now added a detailed discussion on this in **Appendix A.1.1**, which we recommend you to take a look at. The essential reason is that the additional term obtained via second-order influence function contains the trace of a $p\times p$ matrix in the numerator, while a factor of $(n+1)^2$ in the denominator. Thus, when $n$ is such that $n^2 >> p$, then we can safely ignore the additional contribution from the second-order influence.
>
>
> - This is precisely the case in our setting since we focus on analyzing the behaviour of the population loss near the interpolation threshold $p\approx n$.
> \
> &nbsp;
>
>
>
> -----
> ### Theorem 4:
> \
> &nbsp;
>
> First of all, we would like to bring your attention to the decoupling of Theorem 4 and Theorem 5. Theorem 4 now only *requires a single assumption A1* (failing which population loss would be zero, so not interesting anyways) and thus it holds much more broadly. Having said this, let us now proceed with your comments about the assumptions
> \
> &nbsp;
>
> (continued in the next comment)

---

> ### Author Response · Authors · 2021-12-02
> **Response to update**
>
> Apologies for the delay in responding -- as only a short while back were we able to, barely, spot your update in the review. Nevertheless, let us directly go to the point made in your comment.
>
> \
> &nbsp;
> We are, as a matter of fact, **already careful about the subtleties** that might be involved here, and it is precisely the **non-zero** minimum eigenvalue that we are talking about --- both in our Assumption 5 (now indexed as assumption 3, and which only Theorem 5 -- in regards to MSE -- requires) as well as in our experiments.
>
> - Specifically, in all our experiments we consider (the inverse of) the minimum non-zero eigenvalue of the Hessian by calculating the **maximum eigenvalue of its pseudoinverse**.
> \
> &nbsp;
>   -  This is because the Hessian need not be full rank at the optimum (nor do we require so in our analysis), and hence because of its singularity there can trivially be zero eigenvalues.  Even in our computations (done rigorously via the higher Float64 precision), we do observe the occurrence of zero eigenvalues (note: threshold for 'zero eigenvalues' is based on the default criteria used in common libraries such as NumPy https://numpy.org/doc/stable/reference/generated/numpy.linalg.matrix_rank.html and as also done in prior work, e.g., [Singh et. al., 2021]).
> \
> &nbsp;
>    - So it is **incorrect** to say that
>     > 'cannot be supported by the experiments in this paper [...] minimum eigenvalue is nonzero almost surely in the parameter space'
>
>       since in the usual, strict, sense of 'minimum', there are zero eigenvalues that occur even empirically. But, of course, we are *not talking about these trivial zero eigenvalues* when talking about "minimum eigenvalue" --- something which we precisely refer to as the minimum non-zero eigenvalue, throughout our paper.  \
> &nbsp;
>    - Therefore, this assumption **can be supported** by the experiments in our paper and **is indeed supported** by the experiments in our paper.  \
> &nbsp;
> - Similar is the case with our stated assumptions and terms appearing in the Theorem, which reflect this minimum non-zero eigenvalue. E.g, if you see Theorem 4,  we have $\lambda_r$ where $r$ denotes the rank of the Hessian at the optimum and eigenvalues are denoted in descending order $\lambda_1 > \cdots > \lambda_r$.  \
> &nbsp;
> - As far as the *minimum non-zero* eigenvalue of the Hessian at the optimum is concerned --- the trend of which we have thoroughly demonstrated in our paper --- it is far from obvious why this would decrease to zero as $p \rightarrow n, p < n$ and then increase again when $p > n$ as well as importantly why it would cause double descent for finite-width neural networks. This is what we have thoroughly established in the paper (via Theorem 5) among other things.
> \
> &nbsp;
>
> **Thus, one can rest assured that we are taking care of the technical subtleties that can possibly arise, with regards to both theoretical and empirical results.**
> \
> &nbsp;
>
> [Singh et. al., 2021] Analytic Insights into Structure and Rank of Neural Network Hessian Maps https://openreview.net/pdf?id=otDgw7LM7Nn, NeurIPS 2021
>
> ----
> \
> &nbsp;
>
> We hope this clarifies the concerns you had and, in light of which, you will reconsider your score. Also, let us know if you have any other questions left -- regarding this or even anything else -- please feel free to describe what they are and we will be happy to answer.

---

### Official Review · Reviewer_hRBt · 2021-10-27

**Correctness:** 3
**Technical Novelty And Significance:** 3
**Empirical Novelty And Significance:** 3
**Recommendation:** 8
**Confidence:** 4

**Main Review:**

This work derives an interesting and to the best of my knowledge new, lower bound on DNN generalization and draws connections with a timely topic - the behaviour of finite width DNNs. The assumptions, proofs, and references to the relevant works are laid out in an accessible manner. A simple picture of the double descent behaviour arises from their work, creating a direct link with the spectrum of Wishart matrices.

Notwithstanding I found the work lacking in the following aspects -

** Assumptions: To derive their bound, the authors make 6 assumptions (A1-A6). Some of which seem quite reasonable however for some I didn't find sufficient support.

Concerning the A2 assumption, arguing a certain term in the Hessian is zero, the authors provide two empirical references supporting this (one being very recent). However, looking at these Refs. I could not find where this is shown or implied. In addition, the authors argue that the loss term ($l_i$) would be close to zero near the interpolation threshold - but on what scale? Notably, the A2 assumption is crucial for one of the main novel features of their bound -- being robust to zero Hessian eigenvalues --- however, as far as I could tell the crux of it is the fact that $C$ always accompanies $[H+\lambda I]^{1}$ and if $C=H$ zero eigenvalues do not contribute. Thus any small deviation of $C$ from being aligned with $H$ may affect this result.

Concerning the A5 assumption, here the authors relate the low and high eigenvalue of the spectrum of the Hessian after training to that before training by some non-universal constants. For this to be of any quantitative use, these constants should be of order $1$ (otherwise the assumption is null). The authors justify this using experiments, however, due to the hardness of computing the low Hessian eigenvalues - these experiments are quite limited in scope. This again is a very crucial assumption - it allows them to glance over the difference between trained DNNs and DNNs at initialization. As such I think it needs more extensive support.

** Tightness of lower bound - in their experiments the authors compare the lower bound with the actual population risk. The aim is to show they exhibit the same trend however if I understood correctly, a noticeable difference appears in their scale, with the bound being orders of magnitude lower than the actual risk slightly away from the interpolation threshold and a factor of $10$, or $e$ smaller at the interpolation threshold. This, along with the assumptions and low amount of experiments, raises concern in how intimately does their bound really affects and reflects performance.

** Other real-world effects - The bound does not depend much on the high eigenvalues of the Hessian, which according to several works also affect generalization quite strongly. It also misses out on the fact that infinite width DNNs often under-perform their finite width counterparts thereby showing evidence of a second over-parameterization scale, not tied with the interpolation threshold (see for instance https://arxiv.org/pdf/2106.06529.pdf). It's true that this happens less for fully connected DNNs considered in the current work, but that's maybe a testimony against taking conclusions on finite width fully connected and applying it to real-world CNNs. In addition, while it may or may not be that the minimal Hessian eigenvalues of the true Hessian will provide a second change of trend - I find it hard to see how the analytical approach, modelling the latter as a random matrix, will provide such a second scale.

** Experiments - Since this work relies on various assumptions, its potential impact is tied with verifying these assumptions. In this aspect I found the work lacking - experiments we're only done on fully-connected DNNs trained on MNIST1D reduced to 500 points. The main text promised some results on CIFAR10 but I couldn't find those in the supplementary material. In addition, I didn't find the little bump in Figure 3 (left) convincing as evidence for an interpolation peak.


**Summary Of The Paper:**

The current work deals with the generalization properties of finite-width DNNs from the Hessian perspective combined with the influence functional approach. Based on several assumptions, they derive a lower bound, bounding the loss by a quantity proportional to the inverse of the minimal non-zero Hessian eigenvalue. In addition, by assuming the Hessian is close to a random matrix, they estimate its lowest (non-zero) eigenvalue and show that their bound can reflect double descent behaviour at the anticipated interpolation threshold. They do so both in the context of MSE loss and cross-entropy loss. Their theoretical results are compared with small scale experiments on downsized (n=500) MNIST datasets.

**Summary Of The Review:**

An interesting exploration into the low eigenvalues of the Hessian and their links with generalization in finite DNNs. Some underlying assumptions need more support, the bound seems non-tight and not clearly reflective of the actual test loss, and the relevance to real-world settings, where other effects come into play, is unclear.

Following discussions with the authors I currently raise my score to 8

---

> ### Author Response · Authors · 2021-11-22
> **Response to reviewer hRBt (2/2)**
>
> > “Tightness of lower bound”
>
> - In our paper, the primary purpose of this lower bound (Theorem 5) is to theoretically showcase the phenomenon of double descent. *Note, we never claim in the paper that this is a tight lower bound.* During the process of obtaining this lower bound, our objective is to isolate the source of double descent --- which, of course, has the consequence of loosening the bounds away from the interpolation threshold.
>
>
> - One of the obvious reasons is that the lower bound is proportional to the minimum non-zero risk $\sigma^2_{\text{min}}$, and thus reduces the scale. Nevertheless, there are still tighter lower bounds that one compute through our framework, for example via the Trace of Hessian inverse **(see Figure 13)**.
>
>
> - Besides, we have to remember that even practically-speaking, we are much more interested in the qualitative trend implied by the current lower bound. This is because the lower bound carries a dependence on some distribution-based quantities (such as the population condition number, $\kappa$). Thus, tightening the lower bounds becomes a priority only when we have solely training-set based quantities.
>
>
> - Nevertheless, obtaining suitable, practically relevant, bounds is indeed an important research problem — and a widely open one. Given that our lower bound is able to successfully explain the behaviour of double descent, we hope that future works can take inspiration from our derived lower bound. Unfortunately, at present, it is beyond the scope of the paper.
> \
> &nbsp;
>
>
> > “real-world effects [..] does not depend much on the high eigenvalues of the Hessian”
>
> *This is not the case.* As mentioned in the response to the above comment, while lower bounding our aim is to isolate the source of double descent, however, it is not the case the the higher eigenvalues of the Hessian do not show up. E.g., we lower bound  $\sum\limits_{i=1}^r\frac{1}{\lambda_i\left(\mathbf{H}^S_L(\theta^\star)\right)}$ by $\frac{1}{\lambda_r\left(\mathbf{H}^S_L(\theta^\star)\right)}$, although it might be of interest to keep the original quantity in a different context.
> \
> &nbsp;
>
> > “ provide a second change of trend”
>
> - One possibility is that a second change of trend can be provided via **higher-order influence functions**. As discussed in detail in the Appendix A.1.1, the second-order influence becomes relevant when $p\approx n^2$, and so this might serve as the basis for such a second scale. And, this would definitely be the next logical question to investigate for future work.
>
>
> - Lastly, we would like to stress that theoretical works on the second over-parameterization scale and Triple descent  [Adlam and Pennington, 2020] are themselves in the nascent stages (e.g., [Adlam and Pennington, 2020] can not even prove this using a single kernel, but have to resort to two separate kernels, let alone finite-width neural networks), so it would be expecting too much — for one of the first works on double descent focussed on finite-width neural networks — to simultaneously explain the second over-parameterization scale.
> \
> &nbsp;
> ----
> \
> &nbsp;
> **Experiments:**
> \
> &nbsp;
>
>
>
> >*CIFAR10, MNIST:*
>
> We, unfortunately, missed adding these results at the time of the original submission. These can be found in **Appendix C.5.**
> \
> &nbsp;
>
> > “*little bump in Figure 3 (left) convincing as evidence for an interpolation peak”*
>
> - We would first like to remark that larger interpolation peaks are usually only observed with the *label noise settings, as shown in prior work [2]*.
>
> - Next, our theory applies on the population loss which has to be clearly twice differentiable, thus not for a `0-1 error’-based loss function which is what this ‘little bump’ corresponds to. (In accordance with our theoretical predictions, the population loss indeed diverges rather conspicuously near the interpolation threshold in this Figure 3.
> \
> &nbsp;
>
> > *Scale of experiments:*
>
> Because of the inherent hardness of computing the minimum non-zero eigenvalue of the Hessian, we are forced to downscale the experiments. Especially, in the case of MSE, as the interpolation threshold is not $p\approx n$, but $p\approx Kn$, where $K$ is the number of classes, which leads us to limit the number of samples $n=500$. But, note that in the case of CE, we do run with a (slightly) larger number of samples $n=2000$.
> \
> &nbsp;
>
> *In the end, we sincerely hope that our response has addressed the concerns you had, and that you will consider reevaluating and updating your score. We are happy to take any further questions or comments that you may have.*
>
> ----
>
> \
> &nbsp;
>
>
> [1] Singh et. al., 2021, Analytic Insights into Structure and Rank of Neural Network Hessian Maps https://openreview.net/pdf?id=otDgw7LM7Nn
>
> [2] Nakkiran et al., 2020, Deep Double Descent: Where Bigger Models and More Data Hurt, https://openreview.net/pdf?id=B1g5sA4twr
>
> [3] Nakkiran et al., 2021, Optimal Regularization can Mitigate Double Descent, https://openreview.net/forum?id=7R7fAoUygoa

---

> > ### Comment · Reviewer_hRBt · 2021-11-29
> > **Reply**
> >
> > I thank the authors for their detailed response.
> >
> > I raised several concerns in my review. The authors' reply convinced me that their A2 assumption about certain terms in the hessian being zero is reasonable. Also, I tend to accept their claim that violating this assumption only perturbs the results.
> >
> > My original and main concern was that the bound itself is too crude to capture quantitative and/or real-world effects [even limited to the region near the interpolation threshold, although virtually all DNNs I know work well away from it]. In their reply, the authors showed that the constants involved in the bound change by almost two orders of magnitude. Furthermore, they provided no additional evidence that their bound is tight, even just near the interpolation threshold and on a log scale. They did provide further evidence that the minimal non-zero eigenvalue follows the double descent trend. Still, my general view on that matter stayed the same.
> >
> > In that aspect, I want to comment that the trend of zero eigenvalues is also likely to follow the interpolation threshold. Indeed after the interpolation threshold, there'll be a growing number of zero eigenvalues so eigenvalue repulsion should push the positive ones further away from zero. Before the threshold, there is no reason for zero eigenvalues since all parameters matter.

---

> > > ### Author Response · Authors · 2021-11-29
> > > **Response to your reply**
> > >
> > > Thanks for going through our rebuttal and for your response. It is good to know that we were able to address your concerns, except for that surrounding the quantitative nature of the lower bound (i.e., with respect to its tightness) --- which, too, hopefully, we will be able to address by the following response:
> > > \
> > > &nbsp;
> > >
> > > ----
> > > >*no additional evidence that their bound is tight, even just near the interpolation threshold and on a log scale*:
> > > There are few important aspects to your concern that we would like to emphasize:
> > >
> > > (1) We do in fact provide such evidence and we would be happy to emphasize this in a revision. Specifically,consider Figure 13, which contains the trace-based lower bound, i.e., $\sum\limits_{i=1}^r\frac{1}{\lambda_i\left(\mathbf{H}^S_L(\theta^\star)\right)}$. Here, closest to the interpolation threshold which is $p=2050$  (as recall, the number of training samples $n = 2000$), the lower bound is $701.4005$ while the population loss is $732.8059$ (in absolute scales). Thus, we do *not* think that the lower bound qualifies as being 'too crude to capture quantitative' effects 'even just near the interpolation'.
> > > \
> > > &nbsp;
> > >
> > > (2) In regards to models away from interpolation threshold, we would like to bring your attention to the fact that we use a common $\sigma^2_{\text{min}}$ across all of them. This is part of the reason for what makes the bound tighter near interpolation as compared to further away. We could of course tune the lower bound for each model separately, but that would be stretching the limits of this --- already secondary, tangential --- exercise compared to the main focus of the paper.
> > > \
> > > &nbsp;
> > >
> > > (3) Most importantly, this leads us to reiterate what we have already emphasized in the text,  that the  **'original and main'** purpose of our lower-bound is to **isolate the source of double-descent** --- not to present a tight lower bound for quantitative purposes!
> > >
> > > - Thus, for our purposes i.e.,  for the eventual result in Theorem 5 --- as is clearly evident in the proof --- we use inequalities (for positive semi-definite matrices) such as $\mathop{tr}(AB) \geq \mathop{tr}(A) \lambda_{\text{min}}(B)$,  Trace $\mathop{Tr}(A)$ being lower bounded by the maximum eigenvalue $\lambda_{\text{max}}(A)$. It is rather natural that using these inequalities will  loosen the bounds.
> > >
> > > - Had the quantitative application been our primary objective, we would have simply skipped these additional lower bounding steps, hung up our boots at Theorem 3 (which by the way is an equality) or at most the starting inequality in Theorem 4 (eqn. 22), and then discussed ways to approximate it efficiently.  From this standpoint, it is worth noting that the final lower bound not only depicts the accurate qualitative picture, but also is quantitatively close at interpolation ($701.4005$ vs $732.8059$).
> > > \
> > > &nbsp;
> > >
> > > (4) Lastly, if you still demand that a theoretical work provably showing *double descent for finite-width neural networks, with any number of hidden layers (and all of them learnable), trained via usual gradient descent, applicable to almost any loss used in practice* --- which is being done for the first time, to the best of our knowledge ---  must also simultaneously present lower-bounds that are the tightest and the most practically useful and scalable to real-world networks, then we cannot but suggest that you peruse some of the related work in the literature on double descent for neural networks [1, 2, 3]. You will quickly find the *(absence of)* practical, quantitative, and/or real-world effects captured by their generalization lower bounds *(if they are provided at all)* --- let alone their restricted theoretical frameworks.
> > >
> > > [1] Harzli et. al., 2021, Double-descent curves in neural networks: a new perspective using Gaussian processes
> > >
> > > [2] Kuzborskij et al., NeurIPS 2021, On the Role of Optimization in Double Descent: A Least Squares Study
> > >
> > > [3] Ba et. al., ICLR 2020, GENERALIZATION OF TWO-LAYER NEURAL NETWORKS: AN ASYMPTOTIC VIEWPOINT
> > > \
> > > &nbsp;
> > >
> > > -----
> > >
> > > >*trend of zero eigenvalues is also likely to follow the interpolation threshold, ...., eigenvalue repulsion should push the positive ones further away from zero*:
> > >
> > > This seems like a very interesting comment, so to avoid any misinterpretation --- can you please clarify what do you precisely mean by your comment? Especially, since the repulsion of eigenvalues is discussed typically in the context of Gaussian orthogonal ensembles, and so for Wigner matrices. But, here the focus is on the Hessian at the optimum, which will be a Wishart matrix --- as you noted in the review as well. Hence, we are not entirely sure what do you want to convey in your comment.
> > > \
> > > &nbsp;
> > > ----
> > >
> > > Thanks once again for your response and for giving the time. Let us know if you have further questions or comments.

---

> > > > ### Comment · Reviewer_hRBt · 2021-11-30
> > > > **Reply to reply to reply**
> > > >
> > > > I again thank the authors for these clarifications.
> > > >
> > > > I put an emphasis on tightness here because I don't believe one can make the statement that one captured the essence of the double descent behaviour with a bound--- without the bound being relatively tight. The author now included the trace-based bound which is indeed satisfying in this manner. However, as the authors themselves say in the revised version, the purpose of using the lowest eigenvalue bound was to isolate the source of this behaviour. The extensive numerics studying the lowest non-zero eigenvalue also reflects this. So switching bounds in this manner just to satisfy my wish for tightness sort of misses the point.
> > > >
> > > > More generally had the double descent behaviour been completely dominated by the lowest or lowest few eigenvalues that would have been very interesting in my mind. Without this being shown, I cannot agree with the statement that the authors identified the source of the double descent in a generic setting.
> > > >
> > > > Regarding level repulsion, the purpose of my comment was to provide an alternative explanation for the relation between double descent to the lowest eigenvalue. Moreover, one that doesn't relate to the main result of the paper, as I see it, namely the bound. Eigenvalue repulsion is a generic feature and largely follows from the reductions to a two-level (or 2x2 matrix) argument for nearby eigenvalues -- one would have to fine-tune 2 parameters in a symmetric real 2x2 matrix to have a level crossing. In the Wishart ensemble, the motion of the bulk from zero as the numbers of eigenvalue increase can be viewed in this manner as well. In particular one can consider an $N+1$ matrix where $N$ is the number of zero eigenvalues and $1$ is an additional nearby eigenvalue. At this point, the fine-tuning of at least $N$ variables (the coupling between this extra eigenvalue and the $N$ zero ones) is needed to make this extra eigenvalue join the former.

---

> > > > > ### Author Response · Authors · 2021-11-30
> > > > > **Evidence for double descent behaviour dominated by lowest few eigenvalues**
> > > > >
> > > > > Thanks a lot for your response and for continuing to engage in the discussion with us.
> > > > >
> > > > > ---
> > > > > \
> > > > > &nbsp;
> > > > > We are glad to hear that trace-based bound is indeed satisfying in that context. We are happy to update the form of Theorem 4 to even reflect the trace-based bound as that further maintains the generality.
> > > > >
> > > > > Nevertheless, below we present significant evidence that will convince you that it is indeed the smallest or the lowest few eigenvalues that dominate and capture the double descent behaviour.
> > > > > \
> > > > > &nbsp;
> > > > > > **Evidence for double descent behaviour dominated by lowest few eigenvalues:**
> > > > >
> > > > > We consider the settings of CIFAR10, MNIST, and MNIST1D, all of which correspond to the double descent shown in Figures 7a, 7b, and 4.3a respectively. To demonstrate how many of the lowest eigenvalues capture the double descent trend, for each model in all of the above settings, we plot the % of the trace (of the Hessian inverse)  $\sum\limits_{i=1}^r\frac{1}{\lambda_i\left(\mathbf{H}^S_L(\theta^\star)\right)}$ captured by the lowest eigenvalues $\lambda_i$ of the Hessian, since the lower bounds exhibit this dependence. In order to ensure consistent comparisons across varying model sizes, we consider the lowest eigenvalues in *{0.5%, 1%, 2%, 5%, 10%}* of the total number of eigenvalues of that model. The results can be found in the following plots:
> > > > >
> > > > > CIFAR10 https://imgur.com/a/MgQTeNp
> > > > >
> > > > > MNIST https://imgur.com/a/wraCGh7
> > > > >
> > > > > MNIST1D https://imgur.com/a/MgQTeNp
> > > > > \
> > > > > &nbsp;
> > > > >
> > > > > **Observations**: We see that across all these cases just 0.5% of the lowest eigenvalues are enough to capture the double descent behaviour  --- capturing a minimum of 60% of the trace near interpolation  across all these settings.
> > > > >
> > > > > Further > 98% of the trace is captured as soon as we have 1% of the lowest eigenvalues for CIFAR10, 0.5% for MNIST, and 5% for MNIST1D.
> > > > >
> > > > > *This clearly shows that the double descent behaviour is indeed captured by a very small handful of the lowest eigenvalues*. (As a matter of fact, near the interpolation threshold, even just using the minimum non-zero eigenvalue alone captures 85.56% of the trace for CIFAR10 and 97.80% for MNIST. )
> > > > > \
> > > > > &nbsp;
> > > > >
> > > > > We hope that this answers your concerns, and we hope that you will consider revising your score in the light of this. We are happy to answer any other questions that you might have.

---

> > > > > > ### Comment · Reviewer_hRBt · 2021-11-30
> > > > > > **Yet another reply**
> > > > > >
> > > > > > This is indeed encouraging, however, can you show that the trace-bound is as tight for CIFAR10 as it is for MNIST? I haven't seen such a graph.
> > > > > >
> > > > > > At this point, I'll raise my score to 6.

---

> > > > > > > ### Author Response · Authors · 2021-11-30
> > > > > > > **Quick reply**
> > > > > > >
> > > > > > > Thanks for your prompt response and for raising the score.
> > > > > > >
> > > > > > > We are in the process of running these calculations and making the graph. But before we run out of time due to the impending close of the discussion deadline, we herewith share the results from the first completed run, for the case of $p=522$ (remember, here interpolation threshold is about 400). In particular, for this model, we have population loss of $4.05$, while the trace-based lower bound is at $3.65$ and the minimum non-zero eigenvalue based lower bound is at $2.96$.
> > > > > > >
> > > > > > > Thus, we believe that even in the CIFAR10 case the bound will be about as tight, if not more. We will share the complete plots soon and add them in our revision.

---

> > > > > > > > ### Comment · Reviewer_hRBt · 2021-11-30
> > > > > > > > **Score revised**
> > > > > > > >
> > > > > > > > Thanks for this. Following this, I move to strong acceptance. Good luck.

---

> ### Author Response · Authors · 2021-11-22
> **Response to reviewer hRBt (1/2)**
>
> Thanks for your valuable comments and feedback, and it has certainly improved our paper further. We are also glad to hear that you found our approach novel, timely, and with accessible presentation.
> \
> &nbsp;
> ----
>
> Before we delve into the discussion of the mentioned comments, we would like to raise your attention to our decoupling of Theorem 4 and Theorem 5. In particular, Theorem 4 only *requires a single assumption A1* (failing which population loss would be zero, so not interesting anyways) and thus shows *— with broad applicability — that population risk depends inversely with the minimum non-zero eigenvalue of the Hessian at the optimum.* The additional assumptions A2-A4 are only needed to prove the exact form of double descent for MSE.
> \
> &nbsp;
>
> Now, having clarified this important aspect, let us proceed to answering your comments in detail.
> \
> &nbsp;
>
> > “Concerning the A2 assumption, arguing a certain term in the Hessian is zero”
>
> - This can be precisely found in Figures 5, S2 of [1, NeurIPS version]  (or Figures 8, S1 of its arxiv version), where the authors in [1] show that the **rank of the functional Hessian converges to $0$** when trained sufficiently (and thereby the entire matrix becomes $\mathbf{0}$.
>
>
> - Next, to clarify the aspect of scale, we investigate the components of the Hessian spectrum in the case of one-hidden layer networks trained to interpolation for 20K epochs. **Appendix C.9** compares the spectrum of overall loss Hessian $\mathbf{H}_L$ and outer-product Hessian $\mathbf{H}_o$ at the optimum by looking at the nuclear norm, spectral norm, minimum non-zero eigenvalue. We observe that across all three measures, the curves for $\mathbf{H}_L$ and $\mathbf{H}_o$ closely track each other — if not coincide. This serves to show that at the scales we consider, the functional Hessian is indeed close to zero, and ‘small deviation of C from being aligned with H’ need not significantly affect our results.
>
>
> - Also notice that training, for a large number of epochs, until interpolation is the key to even observing double descent as noted by other works [2, 3]. This is essentially because early stopping acts as a regularization and prevents the occurrence of double descent. Hence, this also points to the necessity of considering sufficiently large training duration, under which as shown by [1], the functional Hessian goes to zero.
>
>
> - To summarize, the need for assumption A2, in this form, is just to obtain the precise characterization of double descent in Theorem 5 (but Theorem 4 still continues to hold even if this assumption is excluded).
> \
> &nbsp;
>
> > “Concerning the A5 assumption, [...] these constants should be of order 1 [...] it needs more extensive support.”
>
> - We perform a detailed analysis of the nature of these constants in **Appendix C.8**, where show that (equivalently) the ratio of the minimum non-zero eigenvalue of $\mathbf{C}^S_f$ at the optimum to that at initialization is largely $\mathcal{O}(1)$ and independent of the model size. In particular, we find this observation to hold across the model sizes, in both under- and over-parameterized regimes. *Specifically, this ratio has a mean $3.754$, minimum $0.366$, maximum $11.955$.*
>
>
> - We also carry out additional verifications of this assumption in the case of **CIFAR10**, c.f. Figure 4, where we also find that this ratio has a mean of $7.812$ and largely remains independent of the model size.
>
>
> - Further, we would like to emphasize that – via this assumption – we are *by no means constraining the growth of minimum eigenvalue*. All we are claiming is the existence of such finite constants with which we can bound the growth. The *theoretical existence of these constants* can be justified from the fact that the map $\mathbf{A}\mapsto\lambda_i(\mathbf{A})$ is *Lipschitz-continuous* on the space of Hermitian matrices, which follows from Weyl's inequality (please see the reference to Tao 2012 in the paper for further details).
>
>
> - As a result, given the above empirical and theoretical justifications, we believe there is strong evidence to support this assumption.
> \
> &nbsp;
>
> (continued in next comment)

---

### Official Review · Reviewer_SyuK · 2021-11-02

**Correctness:** 3
**Technical Novelty And Significance:** 4
**Empirical Novelty And Significance:** 2
**Recommendation:** 8
**Confidence:** 4

**Main Review:**

Novelty: While the impact of the Hessian on the learning dynamics is expected, analysing it using the influence functions is novel to the best of my knowledge, and provides valuable  insights,  as  it  allows  to  express  the  population  loss  differently,  and therefore explicit a lower bound showing the double descent behavior.

Clarity:The  paper  is  generally  well  structured  and  clearly  written.  The  argument  is outlined  from  the  beginning  and  detailed  afterwards,  which  makes  it  easy  to follow.

Significance of the theory: The  overall  argument  is  well  constructed,  and  to  the  best  of  my  knowledge,the  provided  proofs  seem  correct.   In  particular,  the  analysis  in  terms  of  the Hessian  spectrum  and  rank  and  the  approach  taken  by  the  authors  to  do  so provide explanations to many of the latest observations in the double descent phenomena (e.g. the effect of regularization [Nakkiran et al., 2020]).  A major strength of the paper is the focus on finite-width models,  and the analysis of the cross entropy loss, which are of particular interest to practitioners.There are however a few weaknesses, in my opinion, of this analysis:
* The  authors  claim  that  the  only  interesting  term  to  analyze  when  the model grows is the Hessian’s smallest non-zero eigenvalue.  There are however  in  the  bounds  other  quantities  that  can  hide  a  dependence  on the size of the model, and on which the authors do not provide sufficient comments - in particular the constants $A_{θ_0}$ and $B_{θ_0}$.  While the role of these constants is conceptually clear (i.e. the spectrum of the covariance of the gradients at the optimum is close to the one at initialization), it is not obvious that these constants are independent of the size of the model.Can the authors provide a more precise comment on these constants?
* The provided analysis considers the size of the model,  independently of its architecture.  However, growing a model by increasing its depth or its width results in different training dynamics.  Can the authors comment on  this  point?   Can  their  analysis  be  adapted  to  take  into  account  the network’s architecture?  What happens if, for example, the difference in width between the hidden layers is larger than 20?
* Recent works suggest the presence of a triple rather than a double descent in  neural  networks  [d’Ascoli  et  al.,  2020].   The  study  of  the  eigenvalues of the Hessian can only provide an explanation for a single descent (by the nature of the random matrix theory it is built on).  Can the authors comment on this line of works and its link to their analysis?

Significance of the experiments: The  experiments  provided  in  the  paper  are  in  line  with  the  theory  and  provide an interesting support for it.  There are however a few weaknesses in the experimental section:
* In figure 1, the authors use a different size of subset for the 2 and 3 layer networks.  While the results show a drop of the eigenvalue and a peak of the test loss in the expected threshold in both cases, it would probably be better to use similar training subsets for a matter of comparability.
* Both the initialization of the network and the randomly chosen training subset can influence the results.  I think that it would be better to show averaged results across different seeds (10 seeds would be great, but given how complicated and time consuming to sweep over the model size, 5 can already be enough to give an indication on the consistency and variance of the results).
* The authors mention results on MNIST and CIFAR 10 in the main text,but they are not available in the appendix. It would be interesting to show the consistency of the results across different datasets.

Minor comment: The authors cite a recent work in the discussion section that suggests that thespectrum of the features of the penultimate layer of the network can explain the double descent phenomena [Kuzborskij et al., 2021], and claim that the currentwork  contradicts  this  conjecture.   There  is  however  no  inherent  contradiction between the two approaches, they study two different quantities.  These workscan in the contrary be complementary, e.g. by indicating two different thresholds(which goes in the direction of a triple descent).

**Summary Of The Paper:**

The paper provides a new analysis of the phenomena of double-descent in maximum likelihood type of estimators, with a focus on neural networks. This analysis is built on:
* expressing the population loss based on a "add-one-in" procedure (a symmetric procedure to the more classical "leave-one-out", which allows in a second stage to use influence functions to provide a lower bound of this loss,
* proving that the lower bound on the population loss diverges at a certain threshold, by analysing the spectrum of the Hessian at the optimum.

The paper also studies the influence of different loss functions on this behavior. In particular, the authors provide an analysis for the case of cross-entropy. Finally, as a by product of their analysis, they apply the influence function analysis to the ``leave-one-out'' loss, and show that, from a practical point of view, second order influence functions can be interesting to analyse along side the currently considered first order ones to provide a better understanding of deep learning.

**Summary Of The Review:**

The paper provides a novel analysis tool for the double descent phenomena, that is particularly useful in the case of finite width neural networks.  While the work can be of high interest to the community, it has some weaknesses, in some of the theoretical results and in the empirical validation that can limit its impact.This is why a recommend acceptance, but I think the paper can be improved.

Post rebuttal: The authors addressed most of my concerns, as well as most if the other reviewers' comments and questions. The resulting paper is substantially improved, and I therefore recommend accepting it.

---

> ### Author Response · Authors · 2021-11-22
> **Response to reviewer SyuK (2/2)**
>
> **Significance of the theory (continued):**
>
> > “presence of a triple rather than a double descent in neural networks [d’Ascoli et al., 2020]”
>
> This is a great question, and especially, since this line of work Triple descent is itself very intriguing.
>
> - The work of  [d’Ascoli et al., 2020], from our understanding, discusses only sample-wise triple descent and not the parameter-wise (which is the focus of our paper). Likewise, [Adlam and Pennington, 2020] seem to show a triple descent but where the input dimension grows along with the number of samples (whereas, input dimension is held constant in double descent for neural networks). Another important thing to notice in their work is that the triple descent is illustrated using two separate kernels choices, i.e., kernel with respect to second-layer weights and the full kernel. So, their theoretical analysis does not even prove triple descent using a single kernel itself (let alone neural networks). Thus, from a theoretical perspective, many aspects of the phenomenon of triple descent still remain unclear — even in the kernel regime — before it can be understood for finite-width neural networks.
>
>
> - Nevertheless, we do agree that this is definitely the logical next step to consider as  [Adlam and Pennington, 2020] do briefly mention some interesting preliminary empirical evidence of triple descent for one-hidden layer networks (albeit with slight differences in initialization as compared to neural networks used in practice).
>
> - In reference to extending our analysis: Of course, there will be differences in the way the third descent manifests. Thus, it would require accounting for these other aspects. E.g., one might have to study the trace of Hessian inverse instead of the minimum non-zero eigenvalue alone, or it may altogether require a finer study of the inherent layerwise structure contained in the Hessian, or perhaps via higher-order influence functions, *(notice, e.g., the second-order influence becomes relevant when $p\approx n^2$, which is also the quadratic scaling regime mentioned in [Adlam and Pennington, 2020])* — yet, in principle, Theorem 3 and (large parts of) Theorem 4 will hold and could still be relevant.
>
>
>
> \
> &nbsp;
> ----
> **Significance of the experiments:**
>
>
> > “5 [seeds] can already be enough to give an indication on the consistency and variance of the results).”
>
> We have updated our **Figure 1** to contain results that are averaged over 5 seeds, and where we also shade the region corresponding to [mean - std. deviation, mean + std. deviation]. While the 2-hidden layer case (Figure 1 right) displays some more variance as compared to the 1-hidden layer case (Figure 1 left), but overall we notice the behaviour and the results still remain consistent. Also, we should bear in mind that the experiments in Figure 1 are run for 5K epochs, and it is likely that running them even longer should further help to consolidate the results. (Due to the constraints of time, we could not run 5 seeds for a much larger number of epochs, but this is an experiment that we aim to do by the time of camera-ready submission.)
>
>
> > “different size of subset for the 2 and 3 layer networks.”
>
> True, it would be better to use similar training sets for comparability. Going forward, we will try to address this aspect, by doing additional experiments by the camera-ready.
>
> > “ results on MNIST and CIFAR 10”
>
> We, unfortunately, missed adding these results at the time of the original submission. These can be found in **Appendix C.5**.
>
> > Contradiction wrt [Kuzborskij et al., 2021]:
>
> It is possible that ours and their approach behave in a complementary manner by indicating two different thresholds. However, the issue (and thereby the contradiction) currently is that [Kuzborskij et al., 2021] claim their conjecture at the threshold of $p\approx n$, but not a different one.
>
> ----
>
> *Thanks once again for your comprehensive review. We really hope that our responses have addressed your concerns in detail and that in the light of which you reconsider your evaluation.*

---

> > ### Comment · Reviewer_SyuK · 2021-11-29
> > **Reply**
> >
> > I thank the authors for their detailed and extensive response.
> >
> > The authors addressed most of my questions and concerns, which I appreciate.
> >
> > Concerning the contradiction with [Kuzborskij et al., 2021]:
> > In this work, the threshold coincides with $p \approx n$ (the threshold indicated in Belkin's work) for the linear case. For their experiments on neural networks, $p$ corresponds to the width of the layer before the last in the network, which makes the thresholds indicated by the two papers potentially different in this case. I think the comparison between the two works is worth further consideration.
> >
> > As I mentioned in my initial review, however, this is a minor comment.

---

> > > ### Author Response · Authors · 2021-11-30
> > > **Response**
> > >
> > > Thanks a lot for your reply and we are very happy to know that we were able to address your concerns.
> > >
> > > Regarding [Kuzborskij et al., 2021] and triple descent in general: On second thought this could indeed be very interesting. This is because, when the layer width or the number of neurons exceed the number of samples, this would lead to the activation matrices --- which are also present in the Hessian --- to have the rank captured in terms of number of samples rather than usual case of being captured by number of neurons, which will bring with it rank deficiency and thereby influence the trend of the minimum non-zero eigenvalue. Further, this would also correspond to the parameterization scale of $p^2 \approx n$, where [Adlam and Pennington, 2020] find the triple descent to occur, and going forward it would be worthwhile to analyse via our framework.
> > >
> > > Lastly, we also ask you to give another thought about updating your score, in light of the fact that we were able to address your questions and concerns in detail via our rebuttal.
> > >
> > > Thanks for your time and engagement in the discussion. We are happy to take any further comments or questions that you might have.

---

> ### Author Response · Authors · 2021-11-22
> **Response to reviewer SyuK (1/2)**
>
> Thanks for your very valuable comments and feedback. We are glad to hear that you found the approach to be novel and the presentation to be structured and clearly written. We answer your comments in detail as follows:
>
> ----
>
> **Significance of the theory:**
>
> > “ in particular the constants $A_{\theta^0}$ and $B_{\theta^0}$ [...] it is not obvious that these constants are independent of the size of the model. Can the authors provide a more precise comment on these constants?”
>
> This is a valid point as these constants, apriori, do not obviously seem to be independent of the model size. What we find empirically is that, quite interestingly, the ratio of the minimum non-zero eigenvalue of $\mathbf{C}^S_f$ at the optimum to that at initialization (which is what governs these two constants) is largely $\mathcal{O}(1)$. In particular, we find this observation to hold across the model sizes, in both under- and over-parameterized regimes. This point is detailed in the **Appendix C.8**, and specifically, Figure 10 and Table 1. To put it briefly, this ratio has a mean $3.754$, minimum $0.366$, maximum $11.955$, and thereby justifies the existence of such universal constants — independent of the model size. This therefore forms the basis for our assumption.
>
> > “However, growing a model by increasing its depth or its width results in different training dynamics.”
>
> You are completely right. This, in fact, touches upon some of the ongoing recent work in the area of double descent [1], where it is posited that growing the model by increasing its depth does not lead to double descent, but the usual U-shaped risk curve. As a result, it is itself not clear whether double descent happens with increasing depth, and thus we chose to focus on the setting of increasing width.
>
>
> > “What happens if, for example, the difference in width between the hidden layers is larger than 20?”
>
> We carried out an experiment in **Appendix C.6** where the hidden layers sizes differ upto **120**. We find that the behaviour remains largely the same, as depicted in Figure 8.
>
> Also, we would like to comment that prior work [2, Appendix B] largely considers fixed and balanced architectural patterns, while increasing the width. So, in this regard, we were actually pleasantly surprised that not only does the double descent behaviour hold but so does the behaviour of the minimum non-zero eigenvalue of the Hessian at the optimum.
>
>
> > “Can their analysis be adapted to take into account the network’s architecture?
>
> In principle, the analysis can handle different architectures, since our influence functions based approach does not rely upon the architecture choice. It may be that one of the assumptions that we make while analyzing double descent (here, mainly for fully-connected networks), however, needs to be altered. But, it would be quite unlikely that the whole framework needs to be adjusted when faced with a new architecture. Further, our current analysis for the fully-connected case, especially the terms that appear in the bounds, could also be used to understand where double descent happens and where it does not.
>
> \
> &nbsp;
>
>
> [1] Nichani et al., 2020, Increasing Depth Leads to U-Shaped Test Risk in Over-parameterized Convolutional Networks, https://arxiv.org/abs/2010.09610
>
> [2] Nakkiran et al., 2020, Deep Double Descent: Where Bigger Models and More Data Hurt, https://openreview.net/pdf?id=B1g5sA4twr
>
> ----
>
> (continued in the next comment)

---

### Official Review · Reviewer_mefD · 2021-11-03

**Correctness:** 3
**Technical Novelty And Significance:** 4
**Empirical Novelty And Significance:** 3
**Recommendation:** 8
**Confidence:** 4

**Main Review:**

Review for "PHENOMENOLOGY OF DOUBLE DESCENT IN FINITE-WIDTH NEURAL NETWORKS"
===

Typos
---
- 2nd paragraph:networksand ==> networks and
- "And while the connection of (infinite-width) neural networks to Neural Tangent Kernel (Jacot et al., 2018) provides interesting parallels, but ultimately its applicability to finite-width neural networks — i.e., models of practical significance — is unclear." <== Please rephrase.
- Please choose between use of "double descent" and "double-descent".
- "Add-one-in procedure ." ==> "Add-one-in procedure." (remove stray space)

Other minor issues
---
- "Thanks to the influence function of the parameter estimate (Eq. 4 for D = D_n)" <== Eq. 4
is in the appendix. It should already appear as an equation in Propositoon 2 of the main
manuscript, then reference it.
- A1: $\ell(f_{\theta_\star}(x), y)$ is not a function, it is a number...
- A4: Doesn't look good. The justification given by the authors is not satisfactory either. Indeed, the authors claim that A4 can be met at the price of a $\lambda_\min(\Sigma)$ factor in the bounds. The issue is that lambda_min(Sigma) that as long as there isn't a sensible lower-bound on $\lambda_\min(\Sigma)$, say in the case of neural networks, then all bets are off. At the very least, the authors should formally / rigorously justify this step.
- In the appendix, results shouldn't be restarted under labels / names different from those used in the main paper. Makes the proof difficult / impossible to follow. For example, Corollary 5 in the paper is called Corollary 13 in the appendix, without any reference to one another; Theorem 3 has become Theorem 10 in the appendix (again without any reference to one another...).

Major issues
----
- Assumption. One of my main concerns with the paper are the assumptions.
  - For example, what is a croncrete nontrivial scenario where all those assumptions hold simultaneously ? This is in contrast to previous works, wherein the assumptions made on the model and data are quite transparent to the reader. For example, what do the results of the current paper say about a single layer neural network with random hidden weights, and output weights learnt with MSE loss (this was analyzed in Adlam and Pennington 2020 "The Neural Tangent Kernel in High Dimensions: Triple Descent and a Multi-Scale Theory of Generalization")?

- It's not clear what is ultimately being assumed in Corollary 5.

After authors' response
---
I'm changing my score from 6 to 8. I was convinced by the answers given to my concerns.


**Summary Of The Paper:**

The paper studies the double-descent phenomenon in a very general scenario: general loss functions which include but are not limited to MSE and cross-entropy, and general finite-width neural networks (i.e not limited to linearized -- NTK / RF-- regimes). The main tool here is a clever use of *influences* function (from classical robust statistics), which allows the authors to study (in principle) double-descent curves for the risk of estimators defined only implicitly via MLE optimization problems, say. The end result is that the authors manage to lower-bound the population risk by a term which depends on the trace of the inverse of the hessian of the loss function at the optimum. When then this hessian becomes singular, the aforementioned trace-of-inverse explodes to infinity, and a double-descent curve occurs. The location of this double-descent curve is accurately predicted by this paper.

**Summary Of The Review:**

This paper presents a completely new approach to understand the phenomenon of double-descent, for very general models / loss functions (including finite-width neural networks). I have a few concerns about the assumptions being used in the paper though (as explained further above).

---

> ### Author Response · Authors · 2021-11-22
> **Response to reviewer mefD**
>
> Thanks a lot for your constructive comments and suggestions, which have really helped to better shape our paper. Also, we are quite encouraged to hear that you recognize the wide generality of our analysis and the clever use of influence functions. Below are the detailed responses to your comments:
> \
> &nbsp;
>
> **Major issues:**
> \
> &nbsp;
>
> >  "Assumptions [....] what do the results of the current paper say about a single layer neural network with random hidden weights, and output weights learnt with MSE loss"
>
> In **Appendix A.4**, we discuss concrete examples of settings where our assumptions hold simultaneously — in particular, the one asked by you about “a single layer neural network with random hidden weights, and output weights learnt with MSE loss”. Essentially, in this scenario, all our assumptions hold comfortably, since:
> - (a) functional Hessian is always zero,
> - (b) the minimum eigenvalue assumption between the optimum and the initialization holds with equality,
> - (c) the sub-Gaussian assumption is satisfied as soon as the inputs are sub-Gaussian.
> We hope this serves to make the assumptions more transparent. Besides, this also indicates that the random features/NTK scenario can be recovered as a simple special case of our finite-width analysis.
> \
> &nbsp;
>
> > It's not clear what is ultimately being assumed in Corollary 5
>
> We assume that you are talking about the double descent result for MSE loss. Specifically, the assumptions A1-A4 (under the new indexing) are being assumed. The above answer and the appendix A.4 should also clarify what is being ultimately assumed here.  (But, please correct us if there is something else that you meant by this comment.)
> \
> &nbsp;
>
> **Minor issues:**
> \
> &nbsp;
>
> We have corrected the typos and resolved the other minor issues, including consistent labelling of the theoretical results (along with linked references to the corresponding one in the main text).
>
> *Regarding assumption A4 (isotropic):* As mentioned in the general overview, we were able to altogether remove this assumption and thus, we have expanded our proof to rigorously handle the non-isotropic case. In terms of the final bound, the difference is that we additionally get the population condition number (i.e., the condition number of the matrix, $\mathbf{C}^{\mathcal{D}}_f$).
>
> We also include a discussion in *Appendix C.7* about the effect of this new term on the overall behaviour of the lower bound. (The short answer is that the minimum non-zero eigenvalue of the Hessian at the optimum still dictates the trend).
>
> \
> &nbsp;
>
> *We hope that these responses have addressed the few concerns you had about the assumptions and that you will accordingly reconsider your evaluation.*

---

> > ### Comment · Reviewer_mefD · 2021-12-03
> > **Feedback to authors's after their feedback**
> >
> > Thanks for the response. I'm convinced with your handling of my concerns. I've raised my score from 6 to 8.
> >
> > Best,

---

### Official Review · Reviewer_hctT · 2021-11-05

**Correctness:** 4
**Technical Novelty And Significance:** 3
**Empirical Novelty And Significance:** 3
**Recommendation:** 8
**Confidence:** 4

**Main Review:**

The paper is very well written, very clear and it is well aware of the current state of research and the literature. The contribution is worth publication.

**Summary Of The Paper:**

The paper discusses the phenomenon of double descent in the framework of finite width neural networks trained with various loss functions. The analysis of the authors relies on the use of influence functions. They provide intuition on the dependence of double descent on the number of samples in the overparametrized regime and on the number of parameters (in the under parametrized regime) by highlighting the dependence of the population loss on the inverse of the smallest eigenvalue of the Hessian. By adding some additional assumption on the initialization and relying on random matrix theory, they are then able to express this smallest eigenvalue as proportional to the square of the difference between the number of training points and the number of parameters which matches the blow up observed numerically at the so-called interpolation threshold.


**Summary Of The Review:**

Introduction

- I would remove the sentence “as well as can cater to the idiosyncrasies that arise with the usage of neural networks”. It is so unclear that you have to explain it right after with the id est. Stick to  what you do, this is good enough: I would replace the entire paragraph with something like “We develop a theoretical analysis of double descent in the finite width framework and study how this phenomenon is impacted by changes in the loss function” or something similar
- “by utilizing the tool of influence functions” —> ”by relying on influence functions which are commonly used in robust statistics” or if you really want “by using the notion of influence functions from ..”
-  In the last paragraph, “starkly” might be a bit strong. I would replace with “clearly” or “which has a clear influence”

Section 2.1.

- When you use not only at the beginning of a paragraph, you should invert : “Not only they have useful properties” —> “Not only do they have useful properties”

Section 2.2.

- Why would you need regularity conditions to exchange the order of the expectation and the derivatives. This is not clear to me. Expand or remove.

Section 3

- The sentence “the change in loss and which we leverage” should be changed to “the change in loss which we leverage”
- The sentence “this quantity is inconsequential ” is not very clear. Why not replace the remark with a clear explanation of the form : “Although the first term on the RHS of (1)  is not the exact training loss, it only deviates from that loss by a negligible quantity related to the difference of the training losses for two distinct training sets. When training for sufficiently long, both losses vanish and the difference thus becomes negligible. ” or something similar


Section 4

- A reference on the decomposition of the Hessian into The outer product Hessian and the functional Hessain would be good
- I would replace the constant $A_{\theta_0}$ and $B_{\theta_0}$ with simpler symbols such as lower case letter or $c_0$ and $c_1$ or
- Do we really use the term external regularization to denote the absence of any regularization?

Section 4.2.

- The key result (to me) which formalizes the double descent phenomenon is really corollary 5. I think it would be good to have that result appear somewhere earlier. For example in an informal form in the introduction.

Section 4.3.

- You provide a lower bound on the population loss, however, from the plot you give in section 4.3. it seems that one could also derive an upper bound (although probably not at the interpolation threshold). I.e. Would such an upper bound more difficult to derive given your proof technique?
- page 7, “we can be thoroughly assure of the accuracy” should be replace by “thoroughly assured” or “we “can thoroughly assure”

Section 5

- The first plot in Figure 3 is unclear. What do you mean by Test error in this case, how does it vary with respect to the population loss as highlighted by the double descent curve shown in red ?
- The explanation of the binary cross entropy loss is not very clear (double check)
- What do you mean by “Note, this is not the case that networks were not trained long enough — rather, we run them for 40, 000 epochs” ? It would be more clear to replace the sentence with something like “Note that the networks were trained for long enough”

Section 6

- In the first paragraph, the sentence “which avoids the need to retrain training set size many models” is unclear. What do you mean ?
- In the statement of Theorem 7, I think you forgot the sentence “Consider the particular case of the ordinary least-squares ” which appears in the appendix. it would be more clear to add in section 6 as well.
- Generally speaking, it is not clear how section 6 relates to the rest of the paper. Is there a direct implication w.r.t double descent ? If I understand well we are not discussing the over parametrization regime as we consider inverting X^TX ? Although we still have a bound in the inverse of the Hessian. What prevents a bound similar to
- Last paragraph : “raises an important aspect”. You don’t really raise an aspect. I would replace with “which highlights an important downside of first order influence functions” or even simpler “which confirms the suboptimality of first order influence functions when used in the leave one out framework”.


A couple of additional comments and/or typos:

- In the introduction “frequently employed deep networksand” —> “networks and”
- In the introduction: “And while the connection … provides interesting parallels, but ultimately its applicability … is unclear.” —>  I would remove the but ultimately
- third paragraph of the introduction, I would remove the “strive” which is not really appropriate here and replace it with the simpler "we develop” (why complicate things when they can be simple?)
- I would the term “function output” this makes the statements less clear. I would stick with “the function $f$”
- Statement of Corollary 5: “For MSE loss” —> “For the MSE loss”

---

> ### Author Response · Authors · 2021-11-22
> **Response to reviewer hctT**
>
> Thank you so much for your detailed, section-wise comments and feedback, which have been very helpful in revising the paper. We were also quite happy to hear your positive words, and *hope the below responses even address some of the remaining concerns.*
>
> ----
> \
> &nbsp;
>
> *Comments on writing:* These make a lot of sense, and we have incorporated all of these into our revision.
>
> *Reference on Hessian decomposition:* This has been added.
>
> *Constants $A_{\theta_0}$ and $B_{\theta_0}$:* You are right, we will replace them with simpler symbols. (We did not do it yet so as to not bother the other reviewers with notational changes)
>
> *Upper bound:* Thanks for the remark. In the **Appendix A.2.1**, we discuss how to obtain an analogous upper bound to the population risk.
> \
> &nbsp;
>
> > "The first plot in Figure 3 is unclear. What do you mean by Test error in this case "
>
> Test error refers to the population error (0-1 loss) approximated over the unseen test set, while the population loss is based on cross-entropy approximated over the same unseen test set (as we lack the access to the entire distribution).
> \
> &nbsp;
>
> > "In the first paragraph, the sentence “which avoids the need to retrain training set size many models” is unclear. What do you mean ?"
>
> We mean that if leave-one-out loss is to be computed naively, then for every choice of the left-out sample, we will have to retrain a new model on the remaining $n-1$ training samples and then compute the loss on the respective left-out sample. Thus, a total of $n$ models have to be trained so as to get the resulting losses on left-out samples before we can average them. We have edited this paragraph and we will also try to further convey this point better.
> \
> &nbsp;
>
> > "how section 6 relates to the rest of the paper."
>
> We added the 'leave-one-out' discussion for completeness since we covered the `add-one-in' procedure, but also since we obtained an important result here that generalizes, in a principled manner,  the well-known LOO formula in linear regression to the case of finite-width neural networks.
>
> This can itself be of use in practice since it provides a proxy for measuring the generalization performance. Besides, concurrently, there seem to be explorations of using the LOO formula for kernel regression in explaining double descent — although in the infinite-width regime [1, https://openreview.net/pdf?id=7grkzyj89A_]. So, our generalized LOO formula can be potentially useful to connect this line of work [1] in the infinite-width regime with the case of finite-width networks.
> \
> &nbsp;
>
> > ​​"we are not discussing the over parametrization regime as we consider inverting X^TX"
>
> Yes, this is a valid point. We have included a remark in **Appendix A.6.1** that handles this extension in the over-parameterized case, and also formally relates it with the (empirical) Neural Tangent Matrix (Jacot et. al., 2018).

---

### Author Response · Authors · 2021-11-22
**General overview of the revision**

To begin, we are very grateful to all the reviewers for their fruitful comments, questions, and feedback — all of which has helped in further strengthening our paper. Also, we want to express that we were delighted to hear that the reviewers recognize the topic as being ‘very important’, featuring a ‘novel’, ‘clever’ use of influence functions that ‘provides valuable insights’ and the overall positive sentiment.

We have been eagerly awaiting the completion of the requested experiments before sharing the revision and posting our rebuttal, but finally, this little wait is over :) In accordance with the reviewer feedback, we have revised our paper, with the updates coloured in **red** and the main changes summarized as follows *(we will also follow-up individually with each of the reviewers, stay tuned!)*:
\
&nbsp;

**Theoretical side:**

- *Relaxed assumptions:* Firstly, we have been able to remove a couple of assumptions, namely assumption A1 about the loss and assumption A4 on isotropicity. As a result, the eventual lower bound for double descent now contains a 'population condition number’.

- *Cosmetic changes:* This has resulted in a cosmetic renumbering of the assumptions: A3 -> A1, A2 -> A2, A5 -> A3, A6 -> A4. We will refer to the new numbering henceforth.

- *Decoupling of Theorem 4 and 5:* Next, so as to precisely reflect the usage of assumptions and their role, Theorem 4 and Corollary 5 (now renamed as Theorem 5) are slightly decoupled. As a consequence, Theorem 4 itself only relies on Assumption A1, while the latter employs assumptions A1 - A4. *Therefore, this serves to illustrate the broader applicability of Theorem 4, which was not clear before.*

- *Support and extensions of theoretical analyses:*  These include,
    - (a) Appendix A.4 describes concrete examples of settings where our assumptions hold simultaneously, in particular, indicates that *random features scenario forms but a special case of our analysis*.
    - (b) Appendix A1.1 details how the first-order influences suffice.
    - (c) Appendix A.2.1 mentions an analogous upper bound to the population risk.
    - (d) Appendix A.6.1 includes a discussion about the extension of LOO formulas for the over-parameterized setting. \
&nbsp;





**Empirical side:**

- First, and foremost, the experiments on CIFAR10 and MNIST can be found in *Appendix C.5*.

- Next, we have run our main experiments for additional seeds and updated the respective plots (see *Figure 1*).

- Importantly, in *Appendix C.8* we establish the nature of constants $A_{\theta^0}$ and $B_{\theta^0}$ by showing that they are indeed $\mathcal{O}(1)$ and *thus implying the existence of universal constants.*

- Further, in *Appendix C.6* we present results for double descent for more non-uniform architectural patterns.

- Also, in *Appendix C.9*, we provide additional support of assumption A2 (on the functional Hessian) by showing that, at the optimum, spectrum of $\mathbf{H}_\mathcal{L}$ is closely – if not exactly – matched by the spectrum of $\mathbf{H}_o$.

\
&nbsp;

*We are hopeful that in light of the above revisions – thanks again to the constructive feedback of the reviewers – we have been able to properly address their concerns.*

Lastly, we are also happy to take any follow-up comments, suggestions, or questions that you might have!

---

### Author Response · Authors · 2021-11-30
**Brief summary (and reminder) to underline our contributions**

Dear reviewers and area chair,

Firstly, we are truly thankful for the time you have devoted to our paper. In particular, the reviewer feedback, as well as the ensuing discussion, has helped us to further refine and strengthen our work. Amidst all of this, we would still like to emphasize the significance of the particular contributions that our paper makes:
\
&nbsp;

1. We provide a theoretical analysis that provably shows the occurrence of double descent behaviour for **finite-width neural networks --- with any number of hidden layers (and all of them learnable), trained via usual gradient descent, applicable to almost any loss used in practice** --- *which is being done for the first time, to the best of our knowledge*.

    - We believe this in itself opens up many directions for future work, since almost all of the past analyses proceed via the NTK/NNGP regime --- relative to which we are miles away.
   - Furthermore, this also provides an interesting direction to study the generalization behaviour of neural networks, since the lower bound in the (now updated) Theorem 4 works with bare minimum assumptions.
   - Also, as we have argued in our response --- by providing further empirical and theoretical evidence --- that the assumptions required in Theorem 5 are indeed quite mild and form natural extensions for an analysis of double descent in the finite-width setting (see Appendices A.4, C.6, C.7, C.8, C.9).
\
&nbsp;

2. We also demonstrate empirically via a number of experiments, *for multiple hidden layers (with both balanced, imbalanced layer-widths settings), across three commonly used datasets (including CIFAR10)*, that indeed **the trend of the minimum non-zero eigenvalue matches our theoretical predictions and captures the peak in population loss arising due to double descent**.

   - Furthermore, in our latest reply to reviewer hRBt, we have shared figures https://imgur.com/a/MgQTeNp, https://imgur.com/a/NentHp8, https://imgur.com/a/wraCGh7 which clearly illustrates that the source of double descent does in fact stem from the lowest few eigenvalues (as indicated by our bounds).
   - Also, we have presented numerical evidence that lower bound is far from being 'crude', and is rather close to the population loss especially near the region of concern, i.e., near the interpolation threshold.
\
&nbsp;

3. Lastly, we provide a **natural generalization of the analytical leave-one-out (LOO) formula in the case of finite-width neural networks** and show that it has intimate connections to the (empirical) NTK at the optimum.
   - In hindsight, this is something we should have perhaps emphasized even more strongly. The reason being the LOO itself is an important object in regards to measuring the generalization performance (since it depends only on the training set), and thereby can find a multitude of applications in practice as well as for future theoretical analyses --- given the concise formula provided here.
\
&nbsp;

Therefore, as the discussion period comes to a close, *we sincerely hope that the reviewers and area chair will be mindful of the significance of contributions made in the paper* and, in accordance, rightfully adjudge the merit of the paper.
\
&nbsp;

Cheers,

Authors

---

### Decision · Program_Chairs · 2022-01-20

**Decision:**

Accept (Poster)

**Comment:**

This paper studies the important statistical phenomenon of double descent, a very timely topic, using influence functions, and thereby derives lower bounds for the population loss. The reviewers generally appreciated the conceptual as well as the technical contributions in the work, but argued that the set of assumptions taken by authors can potentially diminish the significance of the analysis. This, as well as additional issues regarding the empirical and the analytical support for the modeling assumptions (and the implied scope of applicability, i.e. lazy\kernel vs. rich regime) have generated considerable discussion between the reviewers and the authors. Along the process, major and minor concerns were addressed to the satisfaction of the reviewers, resulting in a substantial improvement of the overall evaluation. Thus, this AC recommends acceptance.